# Predicting Deep-Seated Landslide Displacement in Taiwan's Lushan Mountain through the Integration of Convolutional Neural Networks and an Age of Exploration-Inspired Optimizer

**Jui-Sheng Chou[1,*], Hoang-Minh Nguyen[1], Huy-Phuong Phan[1], Kuo-Lung Wang[2]**

[1] Department of Civil and Construction Engineering, National Taiwan University of Science and Technology, Taipei, Taiwan
[2] Department of Civil Engineering, National Chi Nan University, Nantou, Taiwan
(jschou@mail.ntust.edu.tw; d11005813@mail.ntust.edu.tw; huyphuong777@gmail.com; klwang@ncnu.edu.tw)
*Correspondence e-mail address: jschou@mail.ntust.edu.tw

## Abstract

Deep-seated landslides have caused substantial damage to both human life and infrastructure in the past. Developing an early warning system for this type of disaster is crucial to reduce its impact on society. This research contributes to developing predictive early warning systems for deep-seated landslide displacement by employing advanced computational models for environmental risk management. Our novel framework integrates machine learning, time series deep learning, and convolutional neural networks (CNN), enhanced by the Age of Exploration-Inspired Optimizer (AEIO) algorithm. Our approach demonstrates exceptional forecasting capabilities by utilizing eight years of comprehensive data—including displacement, groundwater levels, and meteorological information from the Lushan Mountain region in Taiwan. The AEIO-MobileNet model precisely predicts imminent deep-seated landslide displacement with a mean absolute percentage error (MAPE) of 2.81%. These advancements significantly enhance geohazard informatics by providing reliable and efficient landslide risk assessment and management tools. These safeguard road networks, construction projects, and infrastructure within vulnerable slope areas.

**Keywords:** deep-seated landslide; displacement forecasting; landslide risk assessment; early warning system; machine learning; time-series deep learning; convolutional neural network; metaheuristic optimization.

## 1. Introduction

Landslides are among the most devastating natural disasters (Huang and Fan, 2013), claiming an average of over 4,000 lives annually worldwide between 2004 and 2010 (Petley, 2012). Landslides represent a global hazard, particularly in developing countries, where rapid urbanization, population growth, and significant land use changes occur (Caleca et al., 2024). The identification, management, and monitoring of landslides are made difficult by the diversity of their types (shallow slides, deep-seated slides, rock falls, rock slides, debris flows) and the complexity of their categorization based on triggers, material composition, movement speed, and other characteristics (Das et al., 2022; Hungr et al., 2014). These issues are further exacerbated in countries with complex geological and climatic conditions.

A deep-seated landslide involves the gradual and persistent displacement of a substantial amount of soil and rock, which can escalate into a sudden and devastating event (Kilburn and Petley, 2003; Geertsema et al., 2006; Chigira, 2009). Unlike shallow landslides, which typically affect surface layers to a few meters, deep-seated landslides extend deeper, often exceeding 10 meters, and can involve the movement of underlying bedrock (Lin et al., 2013). Predicting these events is challenging and costly (Thai Pham et al., 2019). Therefore, extensive efforts have been made to predict such disasters throughout history (Corominas and Moya, 2008; David and Raymond, 1989; Aleotti and Chowdhury, 1999). One method that has been employed involves thoroughly examining the physical and geological characteristics of the mountainous areas at risk of landslides (Cotecchia et al., 2020). Furthermore, the level of groundwater has been shown by numerous studies in the past to influence the mechanisms behind landslide formation significantly (Miao and Wang, 2023; Preisig, 2020; Iverson and Major, 1987).

In pursuing a generalized approach to landslide forecasting, researchers have determined that the critical factors associated with slope instability exhibit temporal variability, necessitating using time series data (Chae et al., 2017). This approach combines slope deformation data collected through sensors drilled deep into the slope bed with data on the natural conditions of the monitoring area, which is collected simultaneously. Upon establishing that the data pertinent to landslide prediction falls within the category of time series data, a formidable challenge in research related to this type of disaster is devising a predictive model capable of forecasting the likelihood of such catastrophes based on related factors.

One of the most effective solutions for constructing models to predict time series data involves applying data-driven techniques. The advancement of computational capabilities has driven the widespread adoption of data-driven machine-learning models over physics-based models. This shift is based on the premise that the data used for slope monitoring originates from nonlinear systems (Zhou et al., 2018). However, a significant drawback of traditional machine learning models, such as Random Forest and Support Vector Machines, is their difficulty handling spatiotemporal data. These models need help to capture the sequential relationships necessary for landslide prediction, resulting in lower performance (Zhang et al., 2022a; Tehrani et al., 2022).

An increasing array of novel data-driven solutions is being developed to overcome the constraints of traditional machine-learning approaches. Among these data-driven solutions, convolutional neural networks (CNN) have emerged as one of the most effective methods. These CNN models, which excel at automated feature extraction, can enhance efficiency in analyzing complex datasets and improve the accuracy of prediction results (Alzubaidi et al., 2021).

Moreover, there is a noteworthy recent trend in employing metaheuristic optimization algorithms to fine-tune the hyperparameters of artificial intelligence (AI) models, thereby augmenting their efficiency. This approach has found application in geological and construction studies and other fields, showcasing

substantial effectiveness. Consequently, the fine-tuning of hyperparameters represents a potent avenue for elevating the efficiency of AI models in research focused on predicting deep-seated landslide displacement.

Leveraging the effective methodologies mentioned above, this study employs AI models optimized by an innovative metaheuristic optimization algorithm to predict deep-seated landslide displacement on the northern slope of Lushan Mountain in Ren'ai Township, Nantou County, Taiwan. The geological characteristics of this area have undergone extensive research (Wang et al., 2015; Lin et al., 2020). Previous studies have identified varying depths of the shear plane. Specifically, Lin et al. (2020) determined that the depth of the shear plane is 85m and 106m based on inclinometer data. This research paper is firmly grounded in empirical evidence meticulously collected over eight years from extensometers at depths of 70 and 40 meters. Our analysis also considers the cumulative impact of storms and heavy rainfall on groundwater levels, utilizing data from four stations measuring groundwater levels in the study area and other weather conditions that potentially trigger landslides. The objectives of our research were as follows:

1) To analyze the application of machine learning and deep learning methods to time series data to forecast short-term, deep-seated landslide displacement across the Lushan Mountain area.

2) To identify the optimal model and hyperparameters for accurately forecasting deep-seated landslide displacement in the study area.

3) To evaluate the role of metaheuristic optimization algorithms in fine-tuning the hyperparameters of AI models.

This study represents the first instance of AI models being utilized to predict deep-seated landslides in Lushan Mountain. Additionally, it marks the inaugural application of AEIO for fine-tuning AI models in landslide-related research. Our findings serve as a valuable resource for civil engineers, contractors, and inspectors involved in the planning and overseeing of construction projects in landslide-prone areas. Predicting the likelihood of landslide events can help minimize property loss, guide schedule adjustments, improve work safety, and ensure smooth traffic flow during critical periods. Additionally, understanding internal displacement provides engineers with precise data to evaluate the resilience of structures and infrastructure in vulnerable areas, enabling the issuance of prudent warnings.

**2. Literature Review**

**2.1 Groundwater Levels and the Forecasting of Deep-seated Landslide Displacement**

Landslide triggers can be attributed to loading, slope geometry, weather conditions, and hydrological conditions (Perkins et al., 2024; Van Natijne et al., 2023; Millán-Arancibia and Lavado-Casimiro, 2023; Jones et al., 2023). Among these, hydrological conditions, especially groundwater levels, have been one of the most critical elements considered in studies related to landslide prediction. Numerous studies have substantiated this point. For instance, research by Take et al. (2015) demonstrated that the

distance and velocity of landslides triggered under high-antecedent groundwater conditions are significantly more significant compared to scenarios with drier conditions. Another study has shown that water accumulation at a soil-bedrock contact can develop positive pore water pressures, causing landslides (Matsushi and Matsukura, 2007) (see Figure 1). Moreover, studies on past landslide events have also demonstrated similar findings. Examples of this research include the Tessina landslide in northeastern Italy, where groundwater conditions triggered movement (Petley et al., 2005). Additionally, the study by Keqiang et al. (2015) on water-induced landslides in the Three Gorges Reservoir project area highlights the significant impact of hydrological conditions on the likelihood of such disasters.

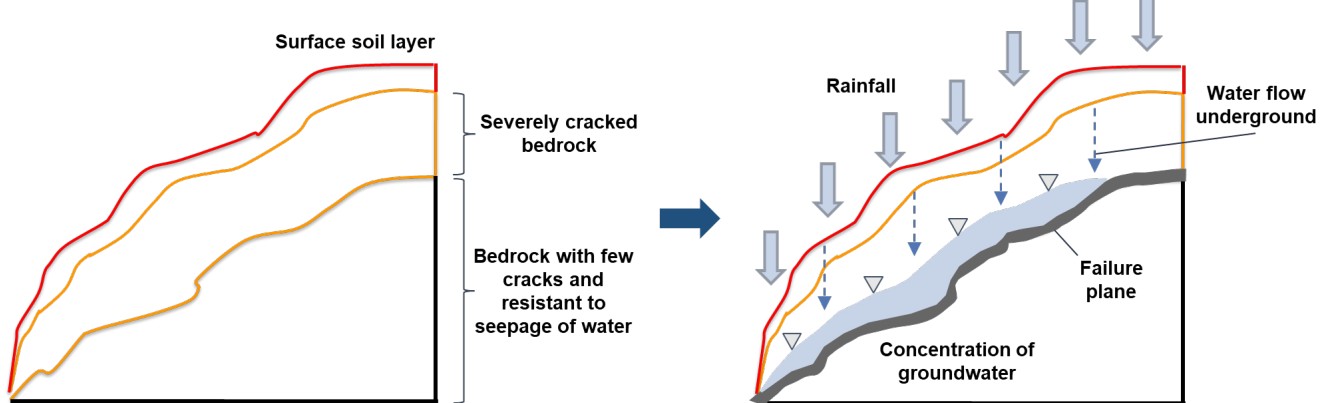

Figure 1. Schematic illustration showing the effects of groundwater on deep-seated slope failure.

Similarly, Preisig (2020) developed a groundwater prediction model for analyzing the stability of a compound slide in the Jura Mountains. Additionally, Srivastava et al. (2020) explored machine learning algorithms to forecast rainfall and established thresholds for landslide probabilities. Although the research by Srivastava et al. did not directly rely on groundwater levels to predict landslides, it is evident that rainfall, a crucial factor in their study for landslide prediction, also influences hydrological conditions. Therefore, their research further underscores the importance of considering groundwater levels in landslide prediction.

The northern slope in the Lushan area of central Taiwan, the region investigated in this study, exhibits significant gravitational slope deformation, making it prone to landslides during typhoons or heavy rainfall events. Lin et al. (2020) conducted in-depth studies on the mechanisms of landslide occurrence based on the geological conditions of the area. While successfully providing valuable insights into the evolution of deep-seated gravitational deformations, their study focuses exclusively on employing traditional analytical methods in geological research, such as analyzing data from geotechnical instruments and conducting geological borehole analysis.

Our research aims to adopt a novel approach compared to previous landslide studies at Lushan Mountain by utilizing AI models and metaheuristic optimization algorithms. This research will utilize

temperature, humidity, and groundwater levels as input data for AI models to predict deep-seated landslide
displacement, thus aiding in landslide forecasting in this region.

## 2.2 Forecasting Slope Displacements: Conventional Methods

Several conventional methods are commonly employed to predict deep slope displacement. These
methods primarily involve simulating factors affecting slope stability in landslide-prone areas using data
collected from ground-based monitoring devices. An early approach to predicting deep-seated slope
movements is geotechnical mapping. This technique characterizes rock and soil's strength, density, and
porosity.
For instance, Crosta and Agliardi (2003) analyzed the geology and rock mass behavior using
Voight's semi-empirical failure criterion, incorporating time-dependent factors to generate velocity curves
that indicate risk levels. Recently, Xu et al. (2018) utilized real-time remote monitoring systems to
measure internal stress, deep displacement, and surface strain. This data was used to formulate forecasting
models to assess slope stability, particularly in railway construction. However, a common challenge with
this method is the instability and frequent changes in the terrain and geology of landslide-prone areas.
This necessitates constant updates to the computational model, which can be time-consuming and labor-
intensive.
Moreover, physically based numerical and laboratory modeling methods are also gaining traction in
landslide research. These methods aim to maintain forecasts using various data types while reducing
human workload and ensuring high accuracy. For example, Mufundirwa et al. conducted a laboratory
study to examine the effectiveness of the inverse velocity model in predicting rock mass destruction
resulting from landslides at depths of 2m and 4m along the sliding plane. This study utilized historically
recorded data from Asamushi, Japan, and the Vaiont reservoir in Italy (Mufundirwa et al., 2010).
Meanwhile, Wu (2010) employed the numerical discontinuous deformation analysis method to
simulate a blocky assembly's post-failure behavior, incorporating earthquake seismic data. Another study
follows this trend by Jiang et al. (2011), who utilized the fluid-solid coupling theory to simulate
displacement and capture the interaction between fluid and solid materials. However, both numerical
models and laboratory modeling methods require substantial effort from researchers. These approaches
demand deep expertise and the development of complex models. More importantly, they rely heavily on
assumptions during the simulation process and may need to reflect real-world conditions, leading to
significant errors accurately.
Stability analysis is another commonly used method related to physics, which evaluates the forces
acting on slope behavior. Fu and Liao (2010) presented a technique for implementing the non-linear Hoek-
Brown shear strength reduction, determining the correlation between normal and shear stress based on the
Hoek-Brown criterion. Subsequently, the micro-units (microscopic components of the rock mass)
instantaneous friction angle and cohesive strength under specific stress conditions are calculated.
Although this approach effectively addresses cost and labor issues, it still heavily relies on the
researcher's assumptions and is limited by the ability to utilize only a small portion of data from the
research area. Additionally, there are several other limitations. For instance, Mebrahtu et al. (2022)
indicated that stability analyses become less reliable in seismic load scenarios. Safaei et al. (2011) also
noted that stability analysis necessitates a substantial amount of detailed input data obtained from
laboratory tests and field measurements, thereby limiting the areas that can be effectively assessed.
As previously mentioned, using conventional methods poses significant challenges, as their
application requires a deep understanding of both the physics involved and the complex behavior of soil.
In addition, traditional methods require specific types of input data, highlighting the rigidity and lack of
flexibility inherent in these approaches (Safaei et al., 2011). In contrast, AI models can overcome these
difficulties by automatically learning to identify mapping functions between input and output data,
eliminating users needing specialized knowledge of soil behavior and physics. Additionally, AI models
can be updated to incorporate new input variables, offering flexibility to leverage available data based on
real-world conditions. Therefore, AI models will be utilized in this research instead of conventional
methods.
**2.3 Forecasting Slope Displacements: Machine Learning and Deep Learning**
In studies employing machine learning and deep learning models for landslide research, a plethora
of research utilizes discrete data to train AI models to predict the probability of landslides or to construct
maps depicting landslide susceptibility. For instance, Pradhan and Lee (2010) used a Geographic
Information System (GIS), remote sensing, and a neural network model to analyze landslide susceptibility
in Cameron Highlands, Malaysia. Ten factors, including topographic slope and drainage distance, were
processed to generate a susceptibility map. The model achieved 83% accuracy in predicting landslide
locations. In a similar study, Pham et al. (2016) used multiple AI models, including support vector
machines (SVM), logistic regression (LR), Fisher's linear discriminant analysis (FLDA), Bayesian
network (BN), and naïve Bayes (NB), for landslide susceptibility assessment in a region within the
Uttarakhand state of India. The SVM model yielded the best prediction results among the models used.
In addition to discrete data, many landslide studies utilize time series data. When it comes to
technical forecasting using time series data, machine learning regression prediction models, such as
extreme learning machine (ELM) (Li et al., 2018),  least squares support vector machine (LSSVM) (Liu
et al., 2019), dynamic neural network (DNN) (Aggarwal et al., 2020), random forests (RFs) (Hu et al.,
2021),  SVM (Zhang et al., 2021), and Gaussian process regression (GPR) (Hu et al., 2019), have proven
highly effective at yielding reliable results. These models also provide scalability and the ability to handle

larger datasets. However, it is essential to note that machine learning models are sensitive to the white noise typical of time series features. This can pose challenges in capturing subtle behaviors and complex interrelationships, mainly when data availability is limited (Zhang et al., 2020). Finally, feature engineering (the process of selecting and transforming input variables to enhance the performance of the models) is computationally intensive and labor-intensive, limiting its applicability when rapid forecasting is required.

Alongside the machine learning models mentioned above, a range of neural network models, from simpler ones like Artificial Neural Networks (ANN) to more advanced approaches such as Deep Neural Networks (DNNs) and CNN, are also employed in research related to landslide (Kumar et al., 2017; Zheng et al., 2022). Notably, CNN models have become increasingly popular and are widely used in research related to this disaster. CNN models often yield superior predictive results than other models in landslide susceptibility assessment and displacement prediction (He et al., 2024).

Moreover, another research trend in landslide forecasting involves the use of time series deep learning models such as Recurrent Neural Networks (RNN), Long Short-Term Memory (LSTM), and Gated Recurrent Units (GRUs), which use previous information to generate current outputs and provide state feedback (Yang et al., 2019; Xu et al., 2022; Yang et al., 2022; Zhang et al., 2022b). These time-series deep learning models can effectively capture patterns of changes over time, making them highly suitable for time-series data in landslide-related studies. However, there has yet to be a comprehensive study that employs a combination of machine learning methods, time-series deep learning, and CNN models to compare and determine the most suitable model for predicting landslide displacement. Therefore, our research aims to address this gap.

Another noteworthy research trend involves using AI models to predict landslides based on spatial-temporal data. For instance, Dahal et al. (2024) utilized spatial-temporal data to pinpoint where landslides may occur and predict when they might happen and the expected landslide area density per mapping unit. The Ensemble Neural Network employed in this research yielded promising predictions, demonstrating its potential for forecasting landslides in Nepal's areas affected by the Gorkha Earthquake. However, our study only managed to gather temporal data. Consequently, the AI models developed in our research will be trained to learn and forecast time-series data.

**2.4 Hybrid metaheuristic optimization algorithm and AI models in landslide prediction**

In landslide-related research, numerous studies have employed hybrid models, wherein metaheuristic optimization algorithms optimize the hyperparameters of AI models. For example, Balogun et al. (2021) studied landslide susceptibility mapping in Western Serbia. This research collected 14 different condition factors to serve as input data for the Support Vector Regression (SVR) model to predict landslide occurrences. The study results indicate that SVR models, with hyperparameters fine-tuned by optimization

algorithms such as gray wolf optimization (GWO), bat algorithm (BA), and cuckoo optimization algorithm (COA), all yielded better prediction results compared to using a single model.

Hakim et al. (2022) conducted a study utilizing CNN models optimized by the GWO and imperialist competitive algorithm (ICA) for landslide susceptibility mapping from geo-environmental and topo-hydrological factors in Incheon, Korea. This research demonstrates that GWO and ICA effectively fine-tuned the CNN model, resulting in a highly accurate landslide susceptibility map.

Jaafari et al. (2022) employed an AI model known as the group method of data handling (GMDH) for classification purposes, optimizing it using the cuckoo search algorithm (CSA) and the whale optimization algorithm (WOA). In northwest Iran, they aimed to predict landslides based on various factors, including topographical, geomorphological, and other environmental factors. After training and testing, the GMDH-CSA model produced superior prediction results compared to the GMDH-WOA and the standalone GMDH model.

It is evident from numerous past studies on landslides that the application of metaheuristic optimization algorithms significantly enhances the predictive effectiveness of AI models. Therefore, this study also incorporates this approach to ensure the model's accuracy in landslide prediction. This study will employ a recently developed metaheuristic algorithm that includes a clustering technique, which shows promise in effectively fine-tuning hyperparameters for AI models.

## 3. Methodology

### 3.1 Machine Learning

In addition to the aforementioned deep learning models, as elucidated earlier, machine learning models will be employed to predict deep-seated landslide displacement in this research. The machine learning models utilized will encompass the following: linear regression (LR) (Stanton, 2001), ANN (Mcculloch and Pitts, 2021), SVR (Drucker et al., 1996), classification and regression tree (CART) (Breiman, 1984), radial basis function neural network (RBFNN) (Han et al., 2010), extreme gradient boosting (XGBoost) (Chen; and Guestrin). These machine learning models will be used to make predictions and will be compared with other deep learning models.

### 3.2 Deep Learning Models for Time Series Data

RNN was introduced by Elman in 1990 (Elman, 1990). This model makes predictions based on sequential data, crucial for language modeling, document classification, and time series analysis. The architecture of an RNN model can be found in Appendix A. In this study, advanced models of RNN, such as LSTM and GRU, are also utilized, and their effectiveness in predicting deep-seated landslides will be compared.

### 3.3 Convolutional Neural Networks

In 1998, LeCun introduced a novel type of DNN known as the CNN, specifically designed for processing data with a grid-like structure, such as images. The complex, layered system of CNN facilitates the automated extraction of features without extensive preprocessing, making it ideal for object recognition, image classification, and segmentation tasks. The detailed mechanism of the CNN model can be found in Appendix B.

This study will use various CNN models to predict deep-seated slope displacement. The CNN models employed in this research include VGG (Simonyan and Zisserman, 2014), ResNet (He et al., 2016), Inception (Szegedy et al., 2015), Xception (Chollet, 2016), MobileNet (Howard et al., 2017), DenseNet (Huang et al., 2017), and NASNet (Zoph et al., 2018). To clarify, the term "standard CNN models" will refer to models with structures that can be user-defined, while "retrained CNN models" will denote those with architectures that have been researched and developed by other scientists and have been proven to be highly effective.

CNN models are typically used for image processing tasks. However, the input data for this study is in numerical and vector form. Therefore, several transformation steps are required to convert this numerical and vector data into image data suitable for CNN input. Detailed information about these transformation steps can be found in the study by Chou and Nguyen 2023 (Chou and Nguyen, 2023).

## 3.4 Data Management and Performance Analysis

### 3.4.1 Data Splitting and Evaluation Strategy

To obtain reliable (i.e., generalizable) evaluation and validation results, it is crucial that the data used for testing does not include the data used for training. Therefore, a dataset must be divided into training, validation, and testing subsets before training the AI model. Training data is used to learn patterns; testing data is used to assess model performance and identify errors; and validation data is used to fine-tune the hyperparameters. In the current study, we opted to refrain from employing cross-validation, which tends to be time-consuming. Instead, we adopted the holdout approach to manage our large dataset with well-represented target variables (Figure 2). A 90:10 ratio is generally used to split datasets into learning and testing data (Di Nunno et al., 2023). When implementing the holdout method during hyperparameter optimization, 20% of the learning data is used for validation, and the remaining 80% is used for training.

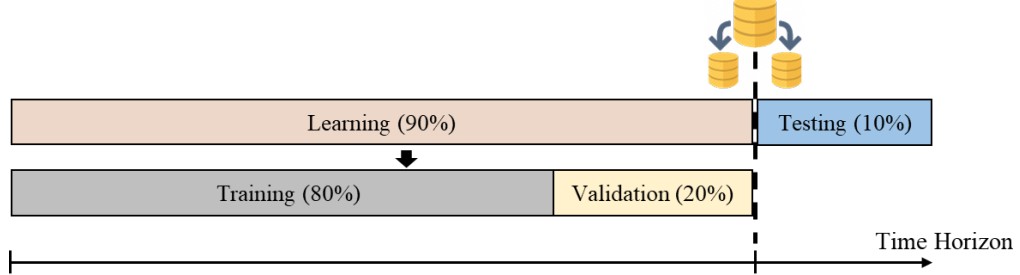

Figure 2. Data are splitting under the proposed Holdout scheme.

### 3.4.2 Performance Evaluation Metrics

This study utilized four widely recognized performance measures to assess the model's effectiveness
in prediction accuracy (Chou and Nguyen, 2023). The measures included mean absolute error (MAE),
mean absolute percentage error (MAPE), and root mean square error (RMSE).
MAE represents the mean of absolute errors, calculated as the average of the absolute differences
between actual and predicted values. Its advantage lies in its simplicity, which provides a straightforward
measure of average prediction error. However, a drawback of MAE is its insensitivity to more significant
errors, so it may not effectively highlight differences between models when significant errors are present.
It is defined as:
$MAE = \frac{1}{n}\sum_{i=1}^{n}|y_i - \hat{y}_i|$                                                                                      (1)
where $n$ is the number of predictions, $y_i$ is the $i^{\text{th}}$ forecasted value, and $\hat{y}_i$ is the corresponding $i^{\text{th}}$ actual
value.
MAPE quantifies the average absolute error ratio to the actual value derived from the differences
between actual and forecasted values. It provides a clear metric in percentage terms, facilitating
straightforward interpretation across various datasets. However, MAPE's limitation arises from its
sensitivity to zero values in the actual data, which can become undefined or impractical to compute,
limiting its utility in scenarios involving zero or near-zero actual values. The expression for MAPE is as
follows:
$MAPE = \frac{1}{n}\sum_{i=1}^{n}\left|\frac{y_i - \hat{y}_i}{y_i}\right|$                                                        (2)
where $n$ is the number of predictions, $y_i$ is the $i^{\text{th}}$ forecasted value, and $\hat{y}_i$ is the corresponding $i^{\text{th}}$ actual
value.
RMSE represents the square root of the average squared error between actual and forecasted values
and is widely used for its ability to indicate the dispersion of errors. This method captures the magnitude
and direction of errors, making it practical for assessing overall prediction accuracy. However, RMSE
tends to be more sensitive to outliers and significant errors than MAE due to its squaring of errors during
computation. This sensitivity can disproportionately affect its evaluation in datasets with extreme values.
The expression for RMSE is as follows:
$RMSE = \sqrt{\frac{1}{n}\sum_{i=1}^{n}(y_i - \hat{y}_i)^2}$                                                              (3)
where $n$ is the number of predictions, $y_i$ is the $i^{\text{th}}$ forecasted value, and $\hat{y}_i$ is the corresponding $i^{\text{th}}$ actual
value.
**3.5 Age of Exploration-Inspired Optimizer**
This study employs a range of AI models to forecast deep-seated landslide displacement in
mountainous regions. To enhance the prediction accuracy of these AI models, the study incorporates a

novel metaheuristic optimization algorithm known as the Age of Exploration-Inspired Optimizer (AEIO). Developed by Chou and Nguyen in 2024, this algorithm has demonstrated high effectiveness in fine-tuning the hyperparameters of AI models. This algorithm treats each particle in the search domain as an explorer. The movement of particles toward regions with higher fitness values parallels the exploratory activities of the Age of Exploration, where explorers sought ideal locations for establishing colonies. In this study, each particle represents a set of hyperparameters, with the ultimate goal of the search process being to identify the optimal particle or hyperparameter set that minimizes prediction error for AI models. Figure 3 illustrates the AEIO algorithm.

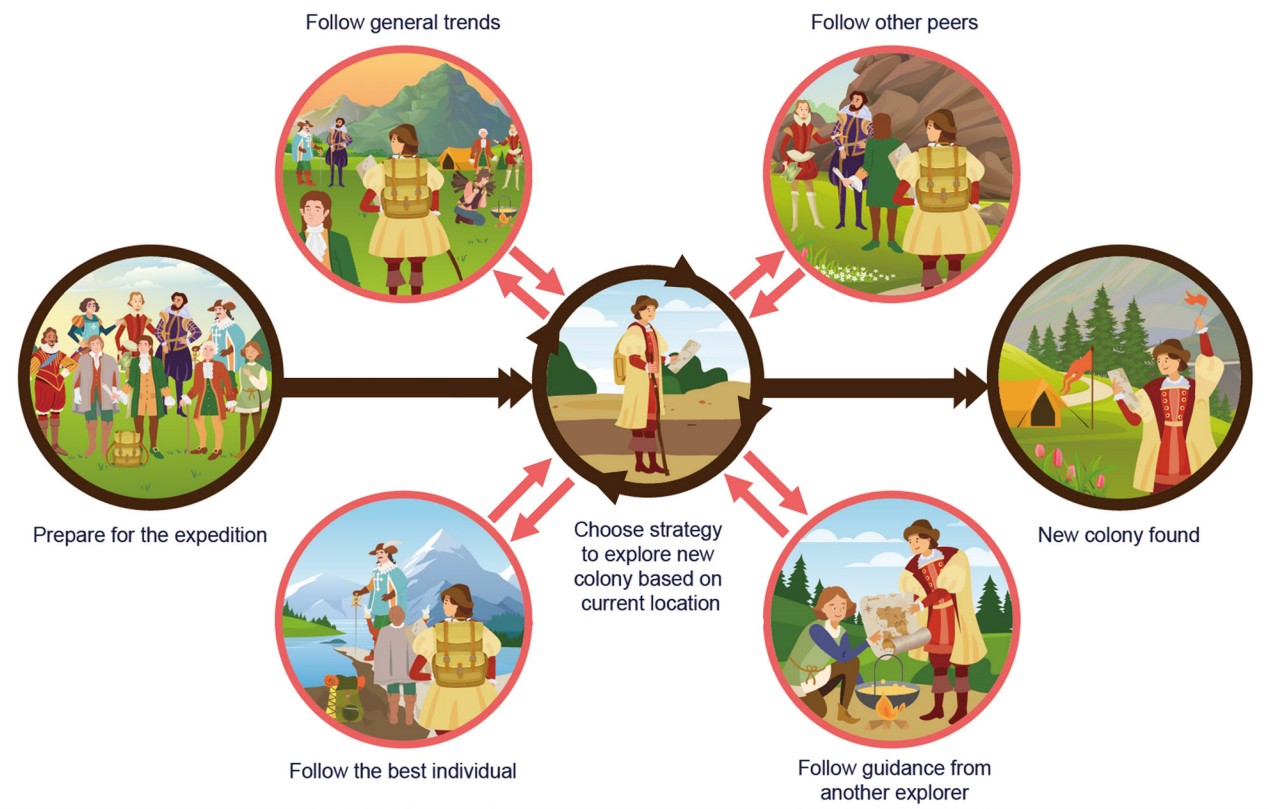

Figure 3. Illustration of Age of Exploration-Inspired Optimizer.

The strength of the AEIO algorithm lies in its ability to develop specific strategies for particles based on their positions, enabling faster convergence to the optimal point and using density-based spatial clustering of applications with noise (DBSCAN) for particle clustering. DBSCAN is an unsupervised clustering method that organizes data points by their spatial closeness in high-dimensional spaces (Ester et al., 1996). This algorithm is particularly effective at detecting clusters of different shapes and densities. It relies on two primary parameters: ε (the radius of the neighborhood) and MinPts (the minimum number of points required to form a dense area). Clusters are created by locating neighboring points with enough surrounding points, while those that do not belong to any cluster are classified as noise or outliers.

Using the DBSCAN algorithm, the AEIO determines whether particles are in favorable or unfavorable positions, reminiscent of explorers during the Age of Exploration. The proximity (within

clusters) allows explorers to gather information and move toward optimal locations, thereby enhancing
their ability to establish new colonies. In contrast, explorers far apart (outside clusters) adopt different
strategies, relying on limited peer guidance or general trends in their quest for new territories.
In each iteration, explorers forecast their next move. If it promises a better position, they relocate.
Otherwise, if the new spot is less favorable for colony establishment, they stay put and await the next
iteration. The algorithm employs specific mathematical formulas to calculate the movement step of
explorers or particles in the AEIO. The exploratory steps of an explorer in the AEIO algorithm will
continuously iterate until the stop condition is satisfied.
● **Explorers follow general trends**
The explorer choosing this movement type will calculate the distance from their location $x_{i,d}(t)$ to
the center of all other explorers ($Meanvl_d(t)$), then attempt to move towards that central point in the
hope of finding a better location with the potential to establish a new colony. The following formula
determines the explorer's position after the movement:
$$x_{i,d}(t+1) = x_{i,d}(t) + \alpha * \left(Meanvl_d(t) - x_{i,d}(t)\right) \times rand(0,1) \times R \qquad (4)$$
$$Meanvl_d(t) = \frac{x_{1,d}(t) + x_{2,d}(t) + \cdots + x_{n_{Pop},d}(t)}{n_{Pop}} \qquad (5)$$
where $d = 1,2,\dots D$; $D$ is the number of dimensions; $i = 1,2,\dots n_{Pop}$; $n_{Pop}$ is the total number of
explorers; $t = 1,2,\dots MaxIt$ is the number of iterations; $MaxIt$ is the maximum value of iteration; $\alpha$ is a
parameter for adjusting the particle's movement toward the centroid position (usually equals 3).
$Meanvl_d(t)$ is the centroid of all particles in dimension $d$. $rand(0,1)$ is the random number in the range
$[0,1]$. $R$: a number that equals 1 or 2 depending on the value of $rand(0, 1)$ per the equation. $R =$
$round(1 + rand(0,1) \times 1)$, $x_{i,d}(t)$ is the location of particle $i$ in iteration $t$, $x_{i,d}(t+1)$ is the location
of particle $i$ in iteration $(t+1)$.
● **Explorers follow three other peers**
Explorers employing this movement method will calculate the average position of three randomly
selected other explorers $\left(\frac{x_{1,d}(t) + x_{2,d}(t) + x_{3,d}(t)}{3}\right)$ and then move toward this newly calculated average
position. The explorer's new position is computed using the following formula:
$$x_{i,d}(t+1) = x_{i,d}(t) + \left(\frac{x_{1,d}(t) + x_{2,d}(t) + x_{3,d}(t)}{3} - x_{i,d}(t)\right) \times rand(0,1) \times R \qquad (6)$$
where: $x_{1,d}(t)$, $x_{2,d}(t)$ and $x_{3,d}(t)$ are three random explorers in dimension $d$ at iteration $t$, $d = 1,2,\dots D$;
$D$ is the number of dimensions; $i = 1,2,\dots n_{Pop}$; $n_{Pop}$ is the total number of explorers; $t = 1,2,\dots MaxIt$
is the number of iterations; $MaxIt$ is the maximum value of iteration.
● **Explorers follow the best one**
According to this strategy, the explorer ($x_{i,d}(t)$) will move closer to the position of another explorer
currently holding the best position ($Best_d(t)$), as determined by the following formula:
$$x_{i,d}(t+1) = x_{i,d}(t) + \left(Best_d(t) - x_{i,d}(t)\right) \times rand(0,1) \times R \qquad (7)$$
where: $Best_d(t)$ represents the position of the particle with the best fitness in dimension d at iteration t,
the parameters $d$ and $t$ hold the same significance as defined in Equation 6.
● **Explorers follow guidance from another one**
Explorers in favorable positions with access to information can execute this movement strategy. In
this scenario, explorers ($x_{i,d}(t)$) will consult with another explorer. The consulted explorer will compare
their direction and distance to the best individual, who holds the most favorable position ($Best_d(t)$) and
guide the inquirer. This algorithm assumes that the inquirer can be any explorer, i.e., a random explorer
($x_{1,d}(t)$). The following formula describes how to calculate the new position of the explorer following
this strategy:
$$x_{i,d}(t+1) = x_{i,d}(t) + \left(Best_d(t) - x_{1,d}(t)\right) \times rand(0,1) \times R \qquad (8)$$
where: $x_{1,d}(t)$ is a random explorer in dimension $d$ at iteration $t$. the parameters $d$ and $t$ hold the same
significance as defined in Equation 6.
● **Crowd control mechanism**
To enhance the efficiency of AEIO in transitioning between exploration and exploitation, a
mechanism is employed to adjust the parameters of DBSCAN throughout each cycle, according to the
following formula:
$$\varepsilon_d = \left(0.1 + \frac{t}{MaxIt}\right) \times (Meanvl_d(t) - Best_d(t)) \qquad (9)$$
$$MinPts = round\left(1 + \frac{t}{MaxIt} \times 10\right) \qquad (10)$$
The exploratory steps in the AEIO algorithm begin by classifying positions using the DBSCAN
algorithm. Subsequently, the explorers update the crowd control mechanism according to equations (9)
and (10), and move according to various strategies defined by equations (4), (6), (7), and (8). This process
is conducted iteratively until the maximum number of iterations is reached.
To fine-tune the hyperparameters of AI models, the AEIO algorithm treats each hyperparameter as
a variable. Furthermore, the objective function of the AEIO algorithm seeks to minimize the prediction
error of AI models, which is quantified by an evaluation metric (MAPE). Figure 4 presents a flowchart
illustrating the process by which the AEIO algorithm aids in fine-tuning hyperparameters for AI models.

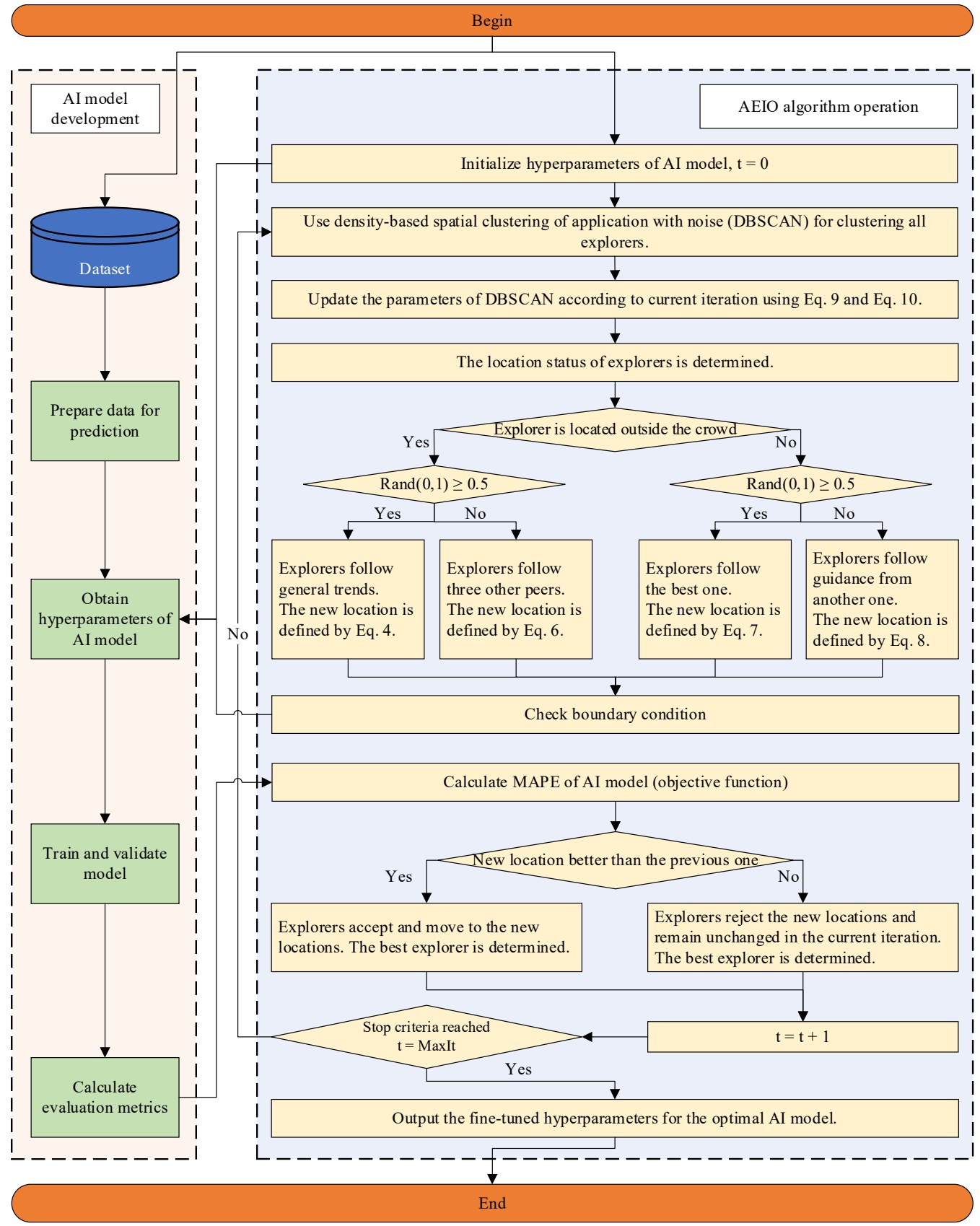

Figure 4. Flowchart of the fine-tuning process of AI models by the AEIO algorithm.

## 4. Lushan Hot Springs: Geography and Geology

### 4.1 Research Area

The current study focuses on the northern slope of the Lushan hot spring in Ren'ai Township, Nantou County, Taiwan (Figure 5), with Nenggao Mountain to the east, Hehuan Peaks to the north, Zhuoshe Mountain to the south, Puli Basins to the west. The terrain features rugged mountain ranges, incipient valleys, and notable river erosion (Lee and Chi, 2011). Lushan Hot Springs is located below the hill, and the main access roads for nearby settlements and hot spring sites include Provincial Highway 14 and County Highway 87.

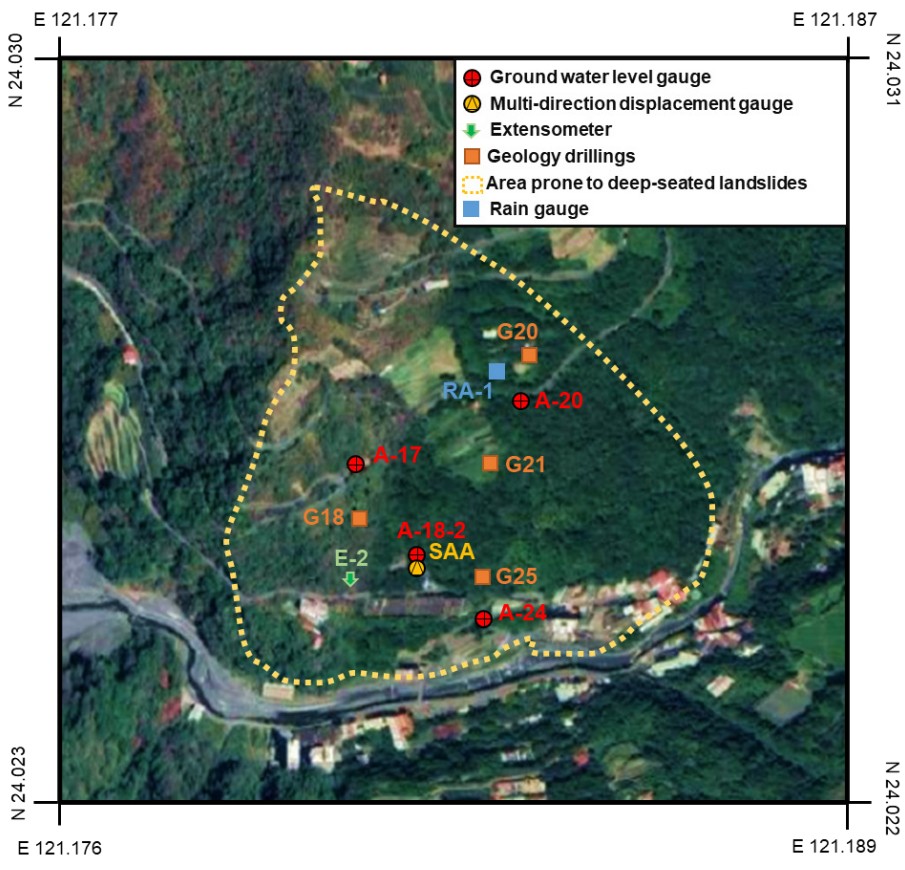

Figure 5. Locations of measurement devices (Image source: Imagery ©2022 CNES/Airbus, Maxar Technologies, Map data ©2022 Google).

In an early study of deep landslides in this area, Lin et al. (2020) reported that the Lushan slope exhibits large-scale deep-seated gravitational slope deformation, characterized by a steep scarp, a gently inclined head, and a curving river at its base. Figure 6 illustrates the geological details of the research area and shows the distribution of four survey boreholes (G20, G21, G18, and G25) along the slope. Regolith, slate, and meta-sandstone are three distinct lithological units revealed through drilling. Additionally, the study by Lin et al. identified the depths of failure planes in these survey boreholes. Specifically, boreholes G18 and G25 did not record any failure planes, while boreholes G20 and G21 recorded failure planes at depths of 85 meters and 106 meters, respectively. These failure planes were identified based on inclinometer data from the corresponding study (Lin et al., 2020).

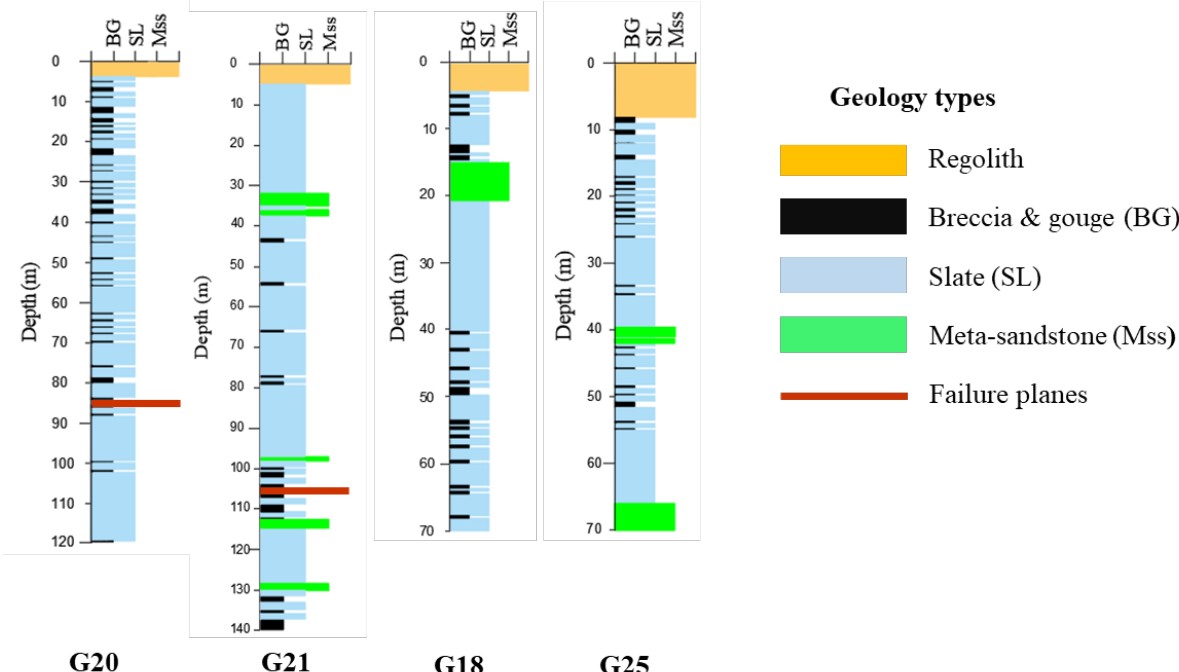

Figure 6. Illustration of geological drilling survey.

Initially, the topmost regolith layer's thickness was less than 10 meters. Secondly, slate predominated, exhibiting a notable presence with sporadic evidence of weathering that resulted in brecciated patterns. This composition frequently broke into breccia and gouges, particularly along cleavage planes and thin shear zones, indicating its susceptibility to collapse. This geological layer is identified as the area's primary cause of landslide risk. Finally, meta-sandstone appeared intermittent compared to the more prevalent lithological units, characterized by its fragility and fractures and occurring less frequently in the drilled samples.

Previous research has detected signs of brittle deformation in the area. These indications include chevron folds within fractures, visible cracks, and intricate jigsaw puzzle-like patterns at the head of the rock formations. Overturned and flexural toppling fractures are prevalent toward the toe of the slope. Additionally, kink bands are observable on fractures recently undergoing flexural folding along the eastern boundary. Notably, horizontal fractures near the toe region also exhibit inter-fracture gouges. Further details on this geological information can be found in the study by Lin et al. (2020). These instances highlight the potential for significant geological changes and landslide risk in this region.

**4.2 Data Collection**

In this study, hourly data of deep-seated landslide displacement and groundwater level were collected by the Department of Civil Engineering, College of Science and Technology, at the National Chi Nan University research group over eight years from July 2009 to June 2017, yielding 68,317 data points. The installation time points and locations are presented in Table 1 and Figure 5, respectively.

The data used in this study were collected using an in-hole telescopic gauge (E-2), a multidirectional
shape acceleration array sensor (SAA) with an underground displacement gauge, and four groundwater
level gauges (A-17, A-18-2, A-20, and A-24). The transmission, storage, and processing of data are
described in detail in the research of Lau et al. (2023).
The operation of the in-hole extensometer entailed the installation of a borehole through the sliding
surface. One end of a steel cable was anchored at the bottom, and a displacement gauge was placed at the
free end to measure deformations automatically. The fixed stops for E-2 and SAA were situated at depths
of 70 meters and 40 meters below the surface, respectively. In addition to groundwater level data,
information regarding significant rainfall events in this area was also measured and is presented in Table

456     2.

Table 1. Device installation time points.

| Year | 2008 | 2009 | 2010 | 2011 | 2012 | 2013 | 2014 | 2015 | 2016 | 2017 |
|---|---|---|---|---|---|---|---|---|---|---|
| **Groundwater level gauge** | A-17 | | | | | | | | | |
| | No data | | | | | | A-18-2 | | | |
| | No data | A-20 | | | | | | | | |
| | No data | A-24 | | | | | | | | |
| **Extensometer** | No data | E-2 | | | | | | | | |
| | No data | | | SAA | | | | | | |


Based on the collected data, analyses have examined the correlation between groundwater levels
and deep-seated landslide displacement at Lushan Mountain. To observe this correlation, graphs
illustrating the precipitation of recorded heavy rainfall (Figure 7A), variations in displacement (Figure 7B
and Figure 7C), and groundwater levels (Figure 7D) over time have been plotted.

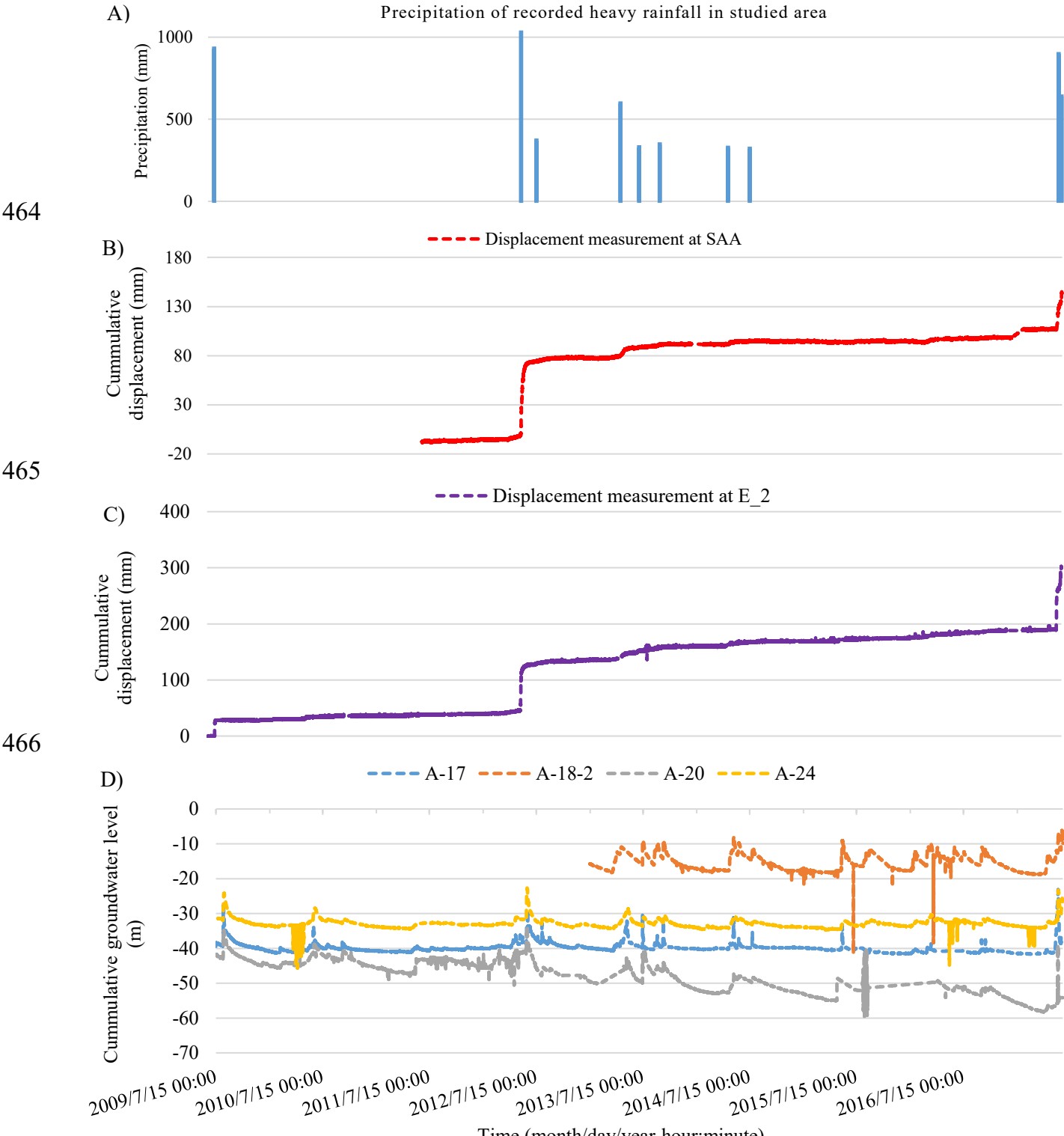





Figure 7. Unified timeline visualization of data in this study: A) Precipitation of recorded heavy rainfall in the studied area;

B) Measured displacements from extensometer SAA C) Measured displacements from extensometer E_2; D) Groundwater

levels at stations A-17, A-18-2, A-20, and A-24.

The graphs above show that the displacement values at both stations often exhibit significant increases coinciding with periods of pronounced fluctuations in groundwater levels. Specifically, in June 2012, there was a notable surge in groundwater levels attributed to heavy rainfall from June 8, 2012, to

June 17, 2012, totaling 1029 mm over 219 hours (as indicated in Table 2 and Figure 7A). The abnormal rise in groundwater levels led to increased pore water pressure, which triggered deep-seated landslide displacement at both stations, namely E_2 and SAA, as evidenced in Figure 7B and Figure 7C.

Table 2. Heavy rainfall events in the study area.

| No. | Rain onset (month/day/year hour: minute) | Rain end time (month/day/year hour: minute) | Accumulating rainfall (mm) | Drop rain hour (hr) | Event |
|---|---|---|---|---|---|
| 1 | 7/17/2008 14:00 | 7/19/2008 21:00 | 418 | 55 | Kameiji typhoon |
| 2 | 9/112008 16:00 | 9/15/2008 12:00 | 943.5 | 92 | Pungentmusc typhoon |
| 3 | 9/28/2008 1:00 | 9/30/2008 10:00 | 523.5 | 57 | Rose honey typhoon |
| 4 | 8/4/2009 3:00 | 8/12/2009 20:00 | 931 | 209 | Mopull typhoon |
| 5 | 6/8/2012 13:00 | 6/17/2012 16:00 | 1029 | 219 | Torrential rain |
| 6 | 7/30/2012 7:00 | 8/3/2012 11:00 | 370 | 100 | Supull typhoon |
| 7 | 5/10/2013 16:00 | 5/25/2013 1:00 | 597 | 345 | Torrential rain |
| 8 | 7/12/2013 19:00 | 7/15/2013 23:00 | 330 | 76 | Suprofit typhoon |
| 9 | 9/20/2013 22:00 | 9/23/2013 18:00 | 347 | 68 | Usagi typhoon |
| 10 | 5/9/2014 5:00 | 5/22/2014 3:00 | 326.5 | 310 | Torrential rain |
| 11 | 7/22/2014 14:00 | 7/24/2014 0:00 | 321.5 | 34 | Madham typhoon |
| 12 | 6/1/2017 11:00 | 6/4/2017 21:00 | 897 | 82 | Torrential rain |
| 13 | 6/11/2017 17:00 | 6/19/2017 3:00 | 638.5 | 178 | Torrential rain |

Similar events occurred in November 2017. Heavy rainfall totaling 638.5 mm over 178 hours during this period also caused a sudden alteration in groundwater levels, resulting in significant deep-seated landslide displacement. Through comparison, it is apparent that there were up to 13 instances of anomalous heavy rainfall during the study period. However, not every example of heavy rain resulted in significant fluctuations in groundwater levels, leading to substantial displacement. Hence, data regarding groundwater level elevation will be used to predict deep-seated landslides rather than rainfall data.

In addition to groundwater level data, weather factors such as temperature and humidity are also utilized as input data for the prediction model. This study includes temperature as an input variable for AI models to predict deep-seated landslide displacement due to its impact on soil structure. Elevated temperatures can cause thermal expansion of soil particles, which can increase pore water pressure and reduce effective frictional resistance forces (Pinyol et al., 2018). Additionally, previous research has shown a relationship between temperature and the likelihood of landslides in clay-rich soils, which are also present in the geological composition of Lushan Mountain (Shibasaki et al., 2017; Loche and Scaringi, 2023).

This study collected groundwater level and displacement data on-site using sensors. Furthermore,
temperature and humidity data were obtained from the website https://power.larc.nasa.gov. This dataset
is part of the Prediction of Worldwide Energy Resource (POWER) project, developed by the National
Aeronautics and Space Administration (NASA) of the United States. The POWER solar data derives from
satellite observations, which are used to infer surface insolation values. Meteorological parameters are
sourced from the Modern-Era Retrospective analysis for Research and Applications, Version 2 (MERRA-
2) assimilation model. The primary solar data is available with a global resolution of 1° x 1°
latitude/longitude, while the meteorological data is provided at a finer resolution of ½° x ⅝°
latitude/longitude. Users can download the data hourly, daily, or monthly through this website.
Table 3 displays the input and output variables for AI models to predict deep-seated landslide
displacement at Lushan Mountain. Two datasets will be generated: one for predicting displacement at the
E_2 station and another for indicating displacement at the SAA station. Table 4 outlines the number of
data points for each dataset and illustrates how the data is divided into training and testing sets.
Table 3. Input and output variables of a model predicting deep-seated landslide displacement.

| | Attributes group | Attributes | Variable ID | Dataset of E_2 station | Dataset of SAA station |
|---|---|---|---|---|---|
| Output variables | Deep-seated landslide displacement measures | Displacement extensometer at station E_2 (mm) | Y1 | ✓ | - |
| | | Displacement extensometer at station SAA (mm) | Y2 | - | ✓ |
| Input variables | Groundwater level data | Groundwater level at station A-17 (m) | X1 | ✓ | ✓ |
| | | Groundwater level at station A-18-2 (m) | X2 | ✓ | ✓ |
| | | Groundwater level at station A-20 (m) | X3 | ✓ | ✓ |
| | | Groundwater level at station A-24 (m) | X4 | ✓ | ✓ |
| | Weather data | Temperature at 2 meters (ºC) | X5 | ✓ | ✓ |
| | | Specific humidity at 2 meters (g/kg) | X6 | ✓ | ✓ |

Table 4. Number of data points.

| Quantity of data points | Dataset of the E-2 station | Dataset of SAA station |
|---|---|---|
| Total data samples | 68312 | 51679 |
| Count of training samples | 61477 | 46523 |
| (90% of the total sample) | (2009/07/15-2016/09/07) | (2011/07/13 – 2016/11/16) |
| Count of testing samples | 6835 | 5156 |
| (10% of the total sample) | (2016/09/07-2017/06/20) | (2016/11/16-2017/06/20) |

**4.3 Data Preprocessing**

Firstly, the data in this study will undergo a normalization process to scale all features to a consistent

range (typically between 0 and 1). This step is essential to ensure that the model considers the importance
of each feature, thereby enhancing overall prediction accuracy (Han et al., 2006).

In the current study, the sliding window technique is implemented after data normalization to

organize data according to a specific time frame. This involves using historical data from previous steps
to predict the output for subsequent steps (Chou and Ngo, 2016). The forecasting horizon refers to the
length of time into the future for which output forecasts are made.

The basic process of the sliding window technique is illustrated in Figure 8. To train AI models, this

study opts for a window size of one week (equivalent to 168 hours). This fixed window size is utilized
exclusively for single AI models. Subsequently, the hybrid model's AEIO algorithm and other
hyperparameters will fine-tune the window size to determine the most suitable settings.

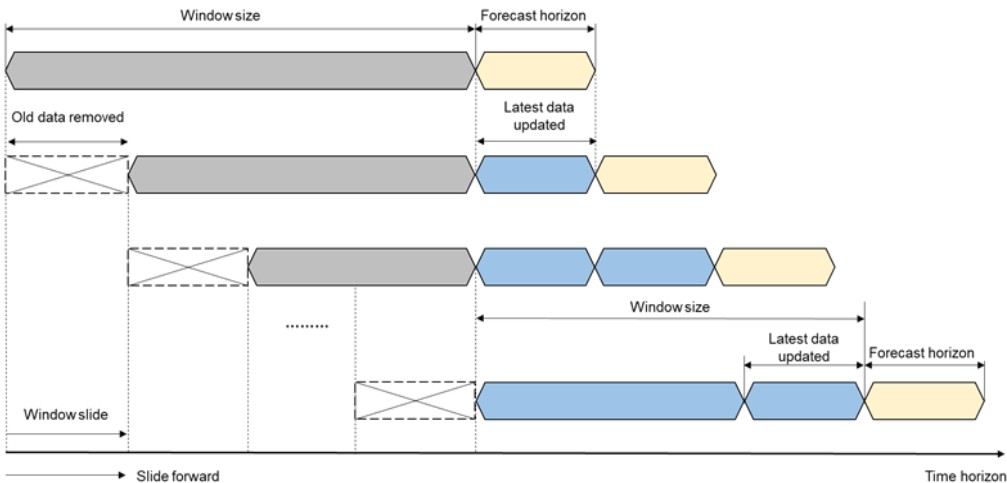


Figure 8. Sliding window technique.

This study focuses on predicting deep displacement values at two distinct time intervals: 1 day ahead

(+24 hours) and seven days ahead (+168 hours). These forecast horizons are strategically chosen to
provide timely information, enabling management departments to make accurate decisions regarding
evacuating people and assets from areas prone to landslides.
Specifically, for valuable assets and machinery that require time for relocation from landslide-prone
areas, having advance knowledge of the landslide event one week ahead of relocation is crucial.
Furthermore, for humans, animals, or other assets that can be evacuated more swiftly, predicting the
landslide one day in advance is sufficient to ensure safety.
The predicted outputs are quantified in mm/day, facilitating decision-making for administrators
according to the TGS-SLOPEM106 standard (Ruitang et al., 2017). Table 5 outlines suggested actions
corresponding to different degrees of deep displacement as per the TGS-SLOPEM106 standard issued by
the Taiwan government.
Table 5. Recommendations are taken from TGS-SLOPEM106 for addressing displacement values in the
early stages of deep sliding.

| Classification of the displacement value | Attention value | Warning value | Action value |
|---|---|---|---|
| Corresponding displacement value | 2 mm/month | 0.5 mm/day | 10 mm/day |
| Condition of slopes | The slope started to slip or slowly move | The hill is undergoing constant-velocity descent. | The rate of slope movement is increasing, elevating the risk of collapse. |
| Recommendations on monitoring activities | - Inspect the monitoring system for any irregularities and consider increasing the frequency of visual inspections | - Enhance the frequency of the automated monitoring system | - Implement a rigorous monitoring system frequency |
| Countermeasures | - Conduct a slope stability investigation and assessment - Develop a reinforcement and improvement plan to enhance slope stability | - Execute emergency slope reinforcement procedures - Develop an emergency response plan for individuals and vehicles within the landslide area | - Evacuate people and vehicles from the landslide area |

**5. Model Development and Analysis Results**
**5.1 Model Development**
Predicting deep-seated landslide displacement at Lushan Mountain is undoubtedly highly
challenging, given that such landslides depend on numerous factors. Therefore, multiple methods will be
employed simultaneously to identify the optimal AI model for prediction. These methods include single
machine learning, time series deep learning, CNN, and hybrid models.
This study will conduct a testing process to systematically identify the optimal model capable of
accurately predicting deep-seated landslides. An illustration of this process can be found in Figure 9.
Initially, the study will sequentially employ various single numerical AI models, such as machine learning
models (LR, ANN, SVR, CART, RBFNN, XGBoost) and time series deep learning models (RNN, R-
RNN, LSTM, R-LSTM, GRU, R-GRU), to forecast displacement.

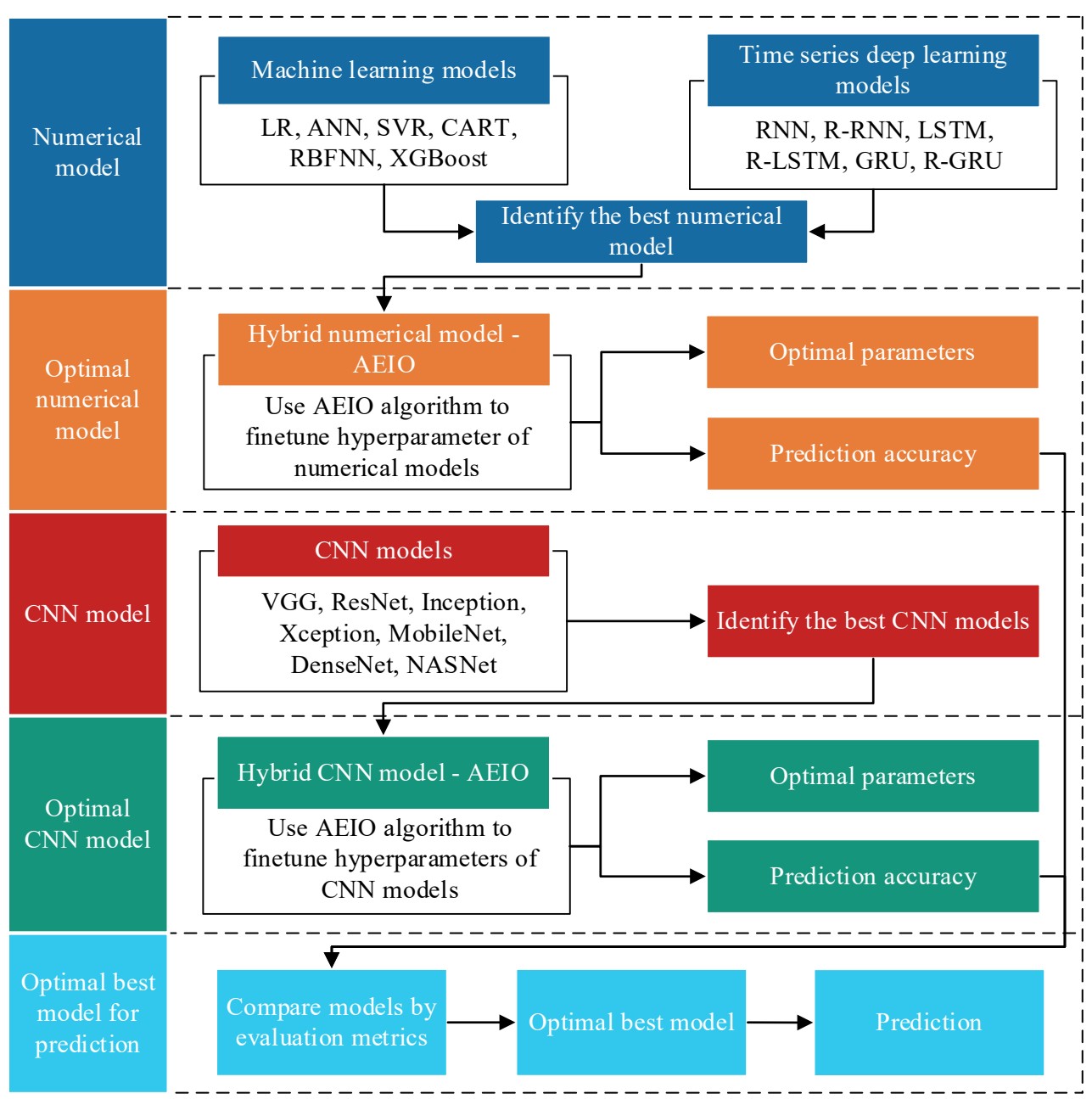


Figure 9. Diagram illustrating the steps for selecting the optimal AI model to predict deep-seated

landslide displacement.

Subsequently, the model with the highest prediction accuracy will be selected for integration with
the AEIO algorithm, forming a hybrid model. In this hybrid model, the hyperparameters of the best
numerical AI model will be fine-tuned by the AEIO algorithm to enhance prediction accuracy.

In addition to the numerical AI models, this study employs individual CNN models for predicting deep-seated landslide displacement. Subsequently, similar to the approach above, the best CNN model with the highest displacement prediction capability will be fine-tuned by the AEIO algorithm within a hybrid model. In the final step, a comparison process between the two hybrid models— one comprising the best numerical model and the other involving the best CNN model fine-tuned by AEIO— will be conducted to select the optimal model for this study.

**5.2 Analysis Results**

This section will present the experimental results of the steps outlined in Figure 9, along with relevant metrics and analysis.

**5.2.1 Numerical Models**

**a. Machine Learning Models**

Initially, single machine learning models will predict deep-seated landslide displacement. In this phase, machine learning models will utilize default hyperparameters, as detailed in the research of Chou and Nguyen (2023). The prediction results of these models at both E-2 and SAA stations are displayed in Table 6. These results show that most machine learning models demonstrate a relatively good predictive capability for displacement, particularly the XGBoost model, which exhibits MAPE values ranging from 8.14% to 9.58%. Following closely, CART also produces favorable prediction results, with MAPE ranging from 8.53% to 9.76%. Regarding prediction accuracy, XGBoost and CART models outperform LR, ANN, SVR, and RBFNN models.

Table 6. Performance results of machine learning models for predicting deep-seated landslide displacement.

| Model | MAPE (%) | | MAE (mm) | | RMSE (mm) | | Time (s) | |
|---|---|---|---|---|---|---|---|---|
| | 1-day-ahead | 7-day-ahead | 1-day-ahead | 7-day-ahead | 1-day-ahead | 7-day-ahead | 1-day-ahead | 7-day-ahead |
| E-2-station | | | | | | | | |
| LR | 10.70 | 11.22 | 22.61 | 21.32 | 28.17 | 31.96 | 0.0001 | 0.003 |
| ANN | 12.31 | 13.31 | 22.19 | 24.92 | 26.56 | 32.54 | 129.80 | 212.83 |
| SVR | 12.46 | 12.47 | 21.98 | 22.56 | 26.27 | 28.05 | 162.55 | 174.44 |
| CART | 8.53 | 8.67 | 15.67 | 16.87 | 25.16 | 27.81 | 1.50 | 2.57 |
| RBFNN | 15.13 | 15.19 | 23.81 | 22.56 | 28.42 | 31.96 | 2.32 | 4.10 |
| XGBoost | **8.14** | **8.36** | 14.80 | 14.68 | 23.07 | 23.92 | 1.58 | 3.28 |
| SAA-station | | | | | | | | |
| LR | 11.18 | 12.11 | 11.51 | 11.64 | 17.26 | 16.07 | 0.01 | 0.01 |
| ANN | 10.91 | 10.93 | 9.43 | 10.45 | 16.55 | 15.92 | 116.78 | 190.69 |

| Model | MAPE (%) | | MAE (mm) | | RMSE (mm) | | Time (s) | |
|---|---|---|---|---|---|---|---|---|
| | 1-day-ahead | 7-day-ahead | 1-day-ahead | 7-day-ahead | 1-day-ahead | 7-day-ahead | 1-day-ahead | 7-day-ahead |
| SVR | 10.55 | 10.94 | 10.87 | 9.18 | 15.64 | 13.42 | 136.01 | 346.30 |
| CART | 10.57 | 10.76 | 7.11 | 7.30 | 13.51 | 10.63 | 0.91 | 1.59 |
| RBFNN | 14.51 | 14.95 | 11.38 | 12.68 | 17.13 | 19.06 | 4.20 | 8.76 |
| XGBoost | **9.17** | **9.58** | 8.43 | 7.83 | 16.36 | 16.97 | 1.12 | 2.29 |

Moreover, the results in Table 6 also indicate that there is not a significant difference in the prediction
errors of the machine learning models at both E-2 and SAA stations, as the error values for both stations
are nearly equal across all machine learning models. Regarding the running time, the LR model
demonstrates the shortest duration, ranging from 0.001 to 0.1 seconds for all runs. However, the prediction
accuracy of this model could be higher, as mentioned earlier. In this case, the machine learning model
with the longest running time is SVR, ranging from 136.01 to 346.3 seconds. This, combined with the low
MAPE score, indicates that the SVR model operates inefficiently with the dataset in this study. After
reviewing the results of the machine learning models in this section, it is observed that XGBoost is the
most suitable machine learning model for predicting deep-seated landslides, exhibiting both high
prediction accuracy and a short running time.
**b. Time series deep learning models**
Similar to the machine learning models, in this section, the time series deep learning models will
also be trained with default hyperparameters, as found in the research of Chou and Nguyen (2023). The
performance results of these models are shown in Table 7. Overall, akin to the machine learning models,
the time series deep learning models also demonstrate fairly good prediction accuracy, especially the best
model - R-GRU model, with MAPE ranging from 7.95 to 9.13%.
Table 7. Performance results of time series deep learning models for predicting deep-seated landslide
displacement.

| Model | MAPE (%) | | MAE (mm) | | RMSE (mm) | | Time (s) | |
|---|---|---|---|---|---|---|---|---|
| | 1-day-ahead | 7-day-ahead | 1-day-ahead | 7-day-ahead | 1-day-ahead | 7-day-ahead | 1-day-ahead | 7-day-ahead |
| **E-2-station** | | | | | | | | |
| RNN | 12.72 | 12.92 | 23.61 | 24.75 | 31.18 | 29.62 | 83.24 | 177.53 |
| R-RNN | 12.31 | 12.84 | 22.88 | 21.97 | 30.20 | 34.42 | 91.47 | 114.33 |
| LSTM | 8.42 | 8.57 | 17.87 | 16.31 | 21.41 | 22.98 | 123.10 | 151.91 |
| R-LSTM | 8.13 | 8.75 | 16.63 | 17.84 | 22.85 | 24.67 | 148.56 | 161.14 |

| Model | MAPE (%) | | MAE (mm) | | RMSE (mm) | | Time (s) | |
|---|---|---|---|---|---|---|---|---|
| | 1-day-ahead | 7-day-ahead | 1-day-ahead | 7-day-ahead | 1-day-ahead | 7-day-ahead | 1-day-ahead | 7-day-ahead |
| GRU | 8.43 | 10.15 | 16.06 | 19.38 | 22.46 | 26.75 | 141.50 | 164.26 |
| R-GRU | **7.90** | **8.16** | 15.09 | 15.69 | 20.84 | 23.32 | 156.97 | 172.96 |
| **SAA-station** | | | | | | | | |
| RNN | 11.92 | 13.98 | 17.61 | 12.65 | 25.71 | 23.19 | 36.77 | 60.31 |
| R-RNN | 14.60 | 14.73 | 18.77 | 13.85 | 26.19 | 24.97 | 49.26 | 59.06 |
| LSTM | 10.64 | 10.94 | 12.73 | 12.25 | 29.21 | 29.57 | 62.84 | 113.76 |
| R-LSTM | 10.14 | 10.35 | 11.77 | 11.60 | 26.10 | 27.48 | 70.94 | 87.48 |
| GRU | 9.32 | 9.28 | 18.05 | 18.11 | 25.26 | 22.41 | 69.56 | 211.77 |
| R-GRU | **8.03** | **9.13** | 18.84 | 17.85 | 21.57 | 21.86 | 79.81 | 212.75 |

The performance of the R-GRU model surpasses that of the GRU model because the R-GRU model learns patterns from time series data in both forward and backward directions on the timeline, thereby capturing more patterns. Furthermore, the R-GRU model produces significantly better prediction results with a more complex learning mechanism than other time series deep learning models. However, due to its complex operational mechanism, the R-GRU model also requires more processing time than other time series deep learning models. From the results of Table 7, it is observed that the operating time of the R-GRU model ranges from 79.81 to 212.75 seconds.

From the conducted analyses, R-GRU has been identified as the best time series deep learning model, owing to its excellent prediction performance. Compared to the best machine learning model, XGBoost (with MAPE ranging from 8.14% to 9.58%), the R-GRU model (with MAPE ranging from 7.90 to 9.13%) demonstrates higher prediction accuracy. Therefore, the R-GRU model will be chosen as the best numerical AI model.

**5.2.2 Best Numerical Model Finetuned by AEIO Algorithm**

This section will focus on fine-tuning the hyperparameters of the numerical model to enhance its performance in predicting deep-seated landslide displacement. The AEIO algorithm will fine-tune the hyperparameters of the study's best numerical AI model, the R-GRU model. Details regarding the names and search ranges of the hyperparameters are outlined in Table 8. The objective function of the AEIO algorithm during the fine-tuning process is to minimize the MAPE value of the R-GRU model.

Table 8. Search ranges of the hyperparameters of the optimal hybrid numerical models (Chou and Nguyen, 2023).

| Hybrid model | Hyperparameter | Search range |
|---|---|---|
| AEIO-R-GRU | Window size | [1-720] |
| | Number of hidden units | [1-400] |
| | Learning rate | [0.0001, 0.5] |
| | Dropout | [0.00, 0.99] |
| | Number of epochs | [10, 120] |
| | Batch size | [32, 64] |

Table 9 illustrates the results of the fine-tuning process. From this table, it is observed that the AEIO algorithm has successfully identified the optimal hyperparameters of the R-GRU model, significantly improving the prediction accuracy of this model. For instance, the MAPE in predicting 1-day-ahead displacement of R-GRU before fine-tuning was 7.9%, but this number decreased to only 3.03% after fine-tuning.

Table 9. Performance results of hybrid time-series deep learning model with AEIO in deep-seated landslide displacement prediction.

| | Model | MAPE (%) | MAE (mm) | RMSE (mm) | Time (s) |
|---|---|---|---|---|---|
| **One-day-ahead displacement prediction** | **E-2-station** | | | | |
| | AEIO-R-GRU | 3.03 | 6.89 | 17.98 | 196 |
| | **SAA-station** | | | | |
| | AEIO-R-GRU | 3.94 | 4.16 | 11.20 | 184 |
| **Seven-day-ahead of displacement prediction** | **E-2-station** | | | | |
| | AEIO-R-GRU | 6.38 | 10.02 | 18.05 | 261 |
| | **SAA-station** | | | | |
| | AEIO-R-GRU | 7.96 | 12.49 | 7.82 | 248 |

Fine-tuning the R-GRU model using AEIO will maximize its potential, minimizing the prediction error to the lowest possible level. Therefore, the results obtained in this section reflect the actual quality of the dataset as well as the level of difficulty in prediction. Specifically, based on the results in Table 9, it is observed that the predictions for one-day ahead displacement (with MAPE of 3.03% and 3.94%) consistently outperform those for seven-days ahead displacement (with MAPE of 6.38% and 7.96%).

One-day-ahead predictions have a shorter time horizon, making them less affected by environmental fluctuations and making changes more accessible to predict. Conversely, in the case of seven-day-ahead

displacement prediction, this timeframe is long enough for various factors, such as weather conditions and
human interventions, to occur, increasing uncertainty and volatility in the predicted figures.
Additionally, Table 9 indicates that predictions from the dataset of the E-2 station consistently
outperform those of the SAA station. Specifically, the displacement prediction at the E-2 station is 3.03%
and 6.38%, better than the corresponding numbers for the SAA station, which are 3.94% and 7.96%,
respectively. This is attributed to the dataset collected by the E-2 station being more comprehensive and
gathered over a more extended period than the SAA station (as shown in Table 4).
Table 10 presents the optimal hyperparameters identified by the AEIO algorithm. Furthermore, in
terms of running time, most models, after fine-tuning, exhibit longer running times compared to the
original model. However, this increase is entirely acceptable since the additional running time is minimal,
and the benefits of fine-tuning are significant, as mentioned above, aiding in the model's more efficient
operation.
Table 10. Optimal hyperparameters of the time series deep learning model identified by the AEIO
algorithm.

| | Model | Window size | Number of hidden units | Dropout rate | Learning rate | Number of epochs | Batch size |
|---|---|---|---|---|---|---|---|
| **One-day-ahead displacement prediction** | **E-2-station** | | | | | | |
| | AEIO-R-GRU | 41 | 81 | 0.27 | 0.7 | 18 | 64 |
| | **SAA- station** | | | | | | |
| | AEIO-R-GRU | 54 | 145 | 0.19 | 0.46 | 32 | 32 |
| **Seven-day-ahead of displacement prediction** | **E-2- station** | | | | | | |
| | AEIO-R-GRU | 97 | 164 | 0.24 | 0.61 | 20 | 32 |
| | **SAA- station** | | | | | | |
| | AEIO-R-GRU | 69 | 147 | 0.28 | 0.31 | 17 | 32 |


**5.2.3 Image-Based CNN Models**
This section presents the results of utilizing CNN models, including VGG, ResNet, Inception,
Xception, DenseNet, and NASNet, to predict deep-seated landslide displacement. The CNN models in
this part use the default settings (Chou and Nguyen, 2023). Table 11 displays the prediction error results
of the CNN models for one-day-ahead and seven-day-ahead forecasts for both E-2 and SAA stations.
Table 11. Performance results of the CNN models for deep-seated landslide displacement prediction.

| Model | MAPE (%) | | MAE (mm) | | RMSE (mm) | | Time (hour) | |
|---|---|---|---|---|---|---|---|---|
| | 1-day-ahead | 7-day-ahead | 1-day-ahead | 7-day-ahead | 1-day-ahead | 7-day-ahead | 1-day-ahead | 7-day-ahead |
| E-2- station | | | | | | | | |
| VGG16 | 4.58 | 7.38 | 12.73 | 13.97 | 26.54 | 35.69 | 3.03 | 3.31 |
| VGG19 | 4.47 | 6.30 | 12.53 | 15.11 | 25.74 | 32.82 | 3.14 | 2.82 |
| ResNet50V2 | 4.87 | 7.68 | 15.28 | 12.52 | 31.82 | 27.19 | 2.99 | 3.44 |
| ResNet101V2 | 4.61 | 6.60 | 9.81 | 9.08 | 34.67 | 32.74 | 2.24 | 2.96 |
| ResNet152V2 | 4.71 | 6.46 | 7.26 | 12.60 | 21.13 | 19.08 | 2.94 | 2.05 |
| InceptionV3 | 4.99 | 7.30 | 11.18 | 11.65 | 32.97 | 34.92 | 2.43 | 3.27 |
| InceptionRestNetV2 | 13.32 | 15.78 | 22.51 | 27.08 | 76.75 | 61.11 | 3.22 | 3.08 |
| Xception | 5.27 | 7.34 | 11.60 | 10.20 | 35.86 | 30.68 | 2.94 | 3.29 |
| MobileNet | **4.11** | 8.92 | 12.22 | 13.62 | 47.43 | 31.72 | 1.21 | 1.44 |
| DenseNet121 | 11.15 | 11.13 | 16.30 | 21.49 | 37.68 | 46.51 | 3.32 | 3.99 |
| DenseNet169 | 4.74 | 7.86 | 11.44 | 12.20 | 17.09 | 36.28 | 3.02 | 3.52 |
| DenseNet201 | 4.66 | **5.30** | 8.11 | 7.44 | 21.82 | 10.39 | 2.09 | 2.29 |
| NASNetMobile | 13.82 | 15.91 | 31.00 | 19.52 | 46.07 | 55.65 | 2.53 | 3.13 |
| NASNetLarge | 13.20 | 34.23 | 20.46 | 61.81 | 61.52 | 75.39 | 3.89 | 3.93 |
| SAA- station | | | | | | | | |
| VGG16 | 5.76 | 7.90 | 6.07 | 12.76 | 9.48 | 8.95 | 3.14 | 3.36 |
| VGG19 | 5.95 | 7.32 | 9.14 | 13.45 | 11.68 | 7.03 | 3.55 | 3.20 |
| ResNet50V2 | 9.87 | 9.35 | 12.43 | 13.81 | 15.71 | 9.75 | 4.57 | 3.83 |
| ResNet101V2 | 8.48 | 17.68 | 10.56 | 19.36 | 11.47 | 21.94 | 3.54 | 3.40 |
| ResNet152V2 | 9.43 | 11.42 | 12.32 | 10.35 | 14.91 | 13.27 | 3.35 | 3.88 |
| InceptionV3 | 10.96 | 8.11 | 12.73 | 9.13 | 14.48 | 12.71 | 3.80 | 3.18 |
| InceptionRestNetV2 | 9.86 | 11.08 | 13.51 | 16.75 | 18.04 | 21.59 | 3.23 | 2.91 |
| Xception | 7.42 | 7.28 | 7.82 | 7.08 | 10.13 | 10.47 | 3.48 | 3.60 |
| MobileNet | 7.12 | **6.80** | 8.28 | 9.92 | 11.58 | 13.83 | 1.43 | 2.13 |
| DenseNet121 | 8.69 | 11.69 | 8.56 | 14.39 | 12.54 | 15.76 | 3.93 | 3.42 |
| DenseNet169 | 6.55 | 9.56 | 6.16 | 9.61 | 11.08 | 15.51 | 3.60 | 3.76 |
| DenseNet201 | **6.36** | 10.45 | 7.46 | 11.62 | 9.37 | 14.51 | 2.51 | 3.13 |
| NASNetMobile | 10.31 | 22.12 | 13.86 | 62.04 | 18.95 | 43.51 | 3.56 | 2.88 |

| Model | MAPE (%) | | MAE (mm) | | RMSE (mm) | | Time (hour) | |
|---|---|---|---|---|---|---|---|---|
| | 1-day-ahead | 7-day-ahead | 1-day-ahead | 7-day-ahead | 1-day-ahead | 7-day-ahead | 1-day-ahead | 7-day-ahead |
| NASNetLarge | 10.25 | 13.69 | 11.20 | 14.05 | 15.95 | 19.09 | 3.18 | 3.34 |

The prediction results demonstrate that most CNN models produce highly accurate predictions. Specifically, predictions made by VGG, ResNet, MobileNet, DenseNet, and Inception exhibit MAPE values below 5%. Among these, MobileNet and DenseNet201 emerge as the two models with the highest accuracy. For one-day-ahead prediction, the best model for predicting displacement at the E-2 station is MobileNet, with a MAPE of 4.11%, and the best model for predicting displacement at the SAA station is DenseNet201, with a MAPE of 6.36%. For seven-day-ahead prediction, the best model for predicting displacement at the E-2 station is DenseNet201, with a MAPE of 5.3%, and the best model for predicting displacement at the SAA station is MobileNet, with a MAPE of 6.8%. These models will be selected accordingly for fine-tuning in the subsequent section.

Regarding running time, the CNN models in this section exhibit significantly longer running times compared to the numerical models in the previous sections. For example, the running time of the best CNN model to predict one-day-ahead displacement at the E-2 station—MobileNet—is 1.21 hours. In contrast, the running time of the best single numerical model for predicting this index is 159.97 seconds.

While CNN models yield better prediction results, considering their extended running times, users need to weigh practical considerations before opting for this type of model. For instance, CNN models should be employed in cases requiring accurate predictions for research and measurement purposes. Conversely, numerical models like R-GRU are more suitable for real-time predictions and computations on low-performance devices.

**5.2.4 Best CNN Models Finetuned by AEIO Algorithm**

As analyzed in Section 5.2.3, the AEIO algorithm will sequentially fine-tune CNN models to enhance prediction accuracy. Table 12 illustrates the search range of hyperparameters for the CNN models to be fine-tuned. Table 12 presents the performance results of the CNN models after being fine-tuned.

Table 12. Search ranges of the hyperparameters of the optimal hybrid numerical models (Chou and Nguyen, 2023).

| Hybrid model | Hyperparameter | Search range |
|---|---|---|
| AEIO-CNN | Learning rate | [0.00, 0.1] |
| | Decay | [0.00, 0.1] |

| Hybrid model | Hyperparameter | Search range |
|---|---|---|
| | Momentum | [0.00, 0.99] |
| | Epsilon | [1.0e-7, 0.001] |
| | Dropout | [0.00, 0.99] |
| | Epochs | [10, 120] |
| | Batch size | [32, 64] |

Table 13. Performance results of best CNN models with AEIO in deep-seated landslide displacement prediction.

| | Model | MAPE (%) | MAE (mm) | RMSE (mm) | Time (hour) |
|---|---|---|---|---|---|
| **One-day-ahead displacement prediction** | E-2-station | | | | |
| | AEIO-MobileNet | 2.81 | 5.09 | 11.92 | 1.25 |
| | SAA-station | | | | |
| | AEIO-DenseNet201 | 3.30 | 6.32 | 15.65 | 3.48 |
| **Seven-day-ahead of displacement prediction** | E-2-station | | | | |
| | AEIO-DenseNet201 | 4.30 | 5.32 | 15.65 | 3.48 |
| | SAA-station | | | | |
| | AEIO-MobileNet | 5.63 | 9.35 | 14.27 | 3.39 |

However, a challenge in this section is that CNN models primarily analyze and learn from image data. Therefore, numerical data must be converted into image data before training. This poses a challenge because current computer hardware may need to be fully capable of efficiently converting numerical data into images for each computation. Hence, this study utilizes the optimal window sizes previously identified for fine-tuning numerical models (Table 10) for this scenario and employs these fixed window sizes for CNN models.

The results of the fine-tuning process demonstrate that the AEIO has successfully identified the optimal hyperparameters for the CNN models, enhancing their accuracy. For instance, in the case of the MobileNet model used for one-day-ahead prediction at the E-2 station, the fine-tuning process reduced the MAPE of this model from 4.11% to 2.81%. A similar trend is also observed in the remaining prediction scenarios.

Furthermore, similar to the case of AEIO-R-GRU, the CNN models exhibit the same trend, where one-day-ahead predictions are more accurate than seven-day-ahead predictions. Similarly, forecasts at the E-2 station demonstrate higher accuracy than predictions at the SAA station. The rationale for this has

been explained in Section 5.2.2. Lastly, the optimal hyperparameters of each CNN model, identified by
the AEIO algorithm, are presented in Table 14. CNN models with optimal hyperparameters are the most
effective models in this study for predicting deep-seated landslide displacement.

Table 14. Optimal hyperparameters of the CNN models identified by the AEIO algorithm.

| | Model | Learning rate | Decay | Momentum | Epsilon | Dropout | Epochs | Batch size |
|---|---|---|---|---|---|---|---|---|
| **One-day-ahead displacement prediction** | **E-2-station** | | | | | | | |
| | AEIO-MobileNet | 0.0011 | 0.00095 | 0.00001 | 3.0e-7 | 0.56 | 15 | 64 |
| | **SAA-station** | | | | | | | |
| | AEIO-DenseNet201 | 0.00012 | 0.0012 | 0.00011 | 1.0e-7 | 0.49 | 16 | 64 |
| **Seven-day-ahead of displacement prediction** | **E-2-station** | | | | | | | |
| | AEIO-DenseNet201 | 0.0012 | 0.0011 | 0.00022 | 1.0e-7 | 0.51 | 15 | 64 |
| | **SAA-station** | | | | | | | |
| | AEIO-MobileNet | 0.00014 | 0.00098 | 0.00011 | 2.0e-7 | 0.50 | 14 | 64 |

Figure 10 illustrates the differences between typical AI models' actual and predicted deep-seated
landslide displacement. Specifically, Figure 10a compares the performance of single models against the
predicted values, while Figure 10b does the same for hybrid models. The chart shows that hybrid models
demonstrate superior predictive capability for deep-seated landslides compared to single models. This is
evident from the displacement line of the hybrid models in Figure 10b, which closely aligns with the actual
deep-seated landslide displacement and significantly outperforms the single models depicted in Figure
10a.

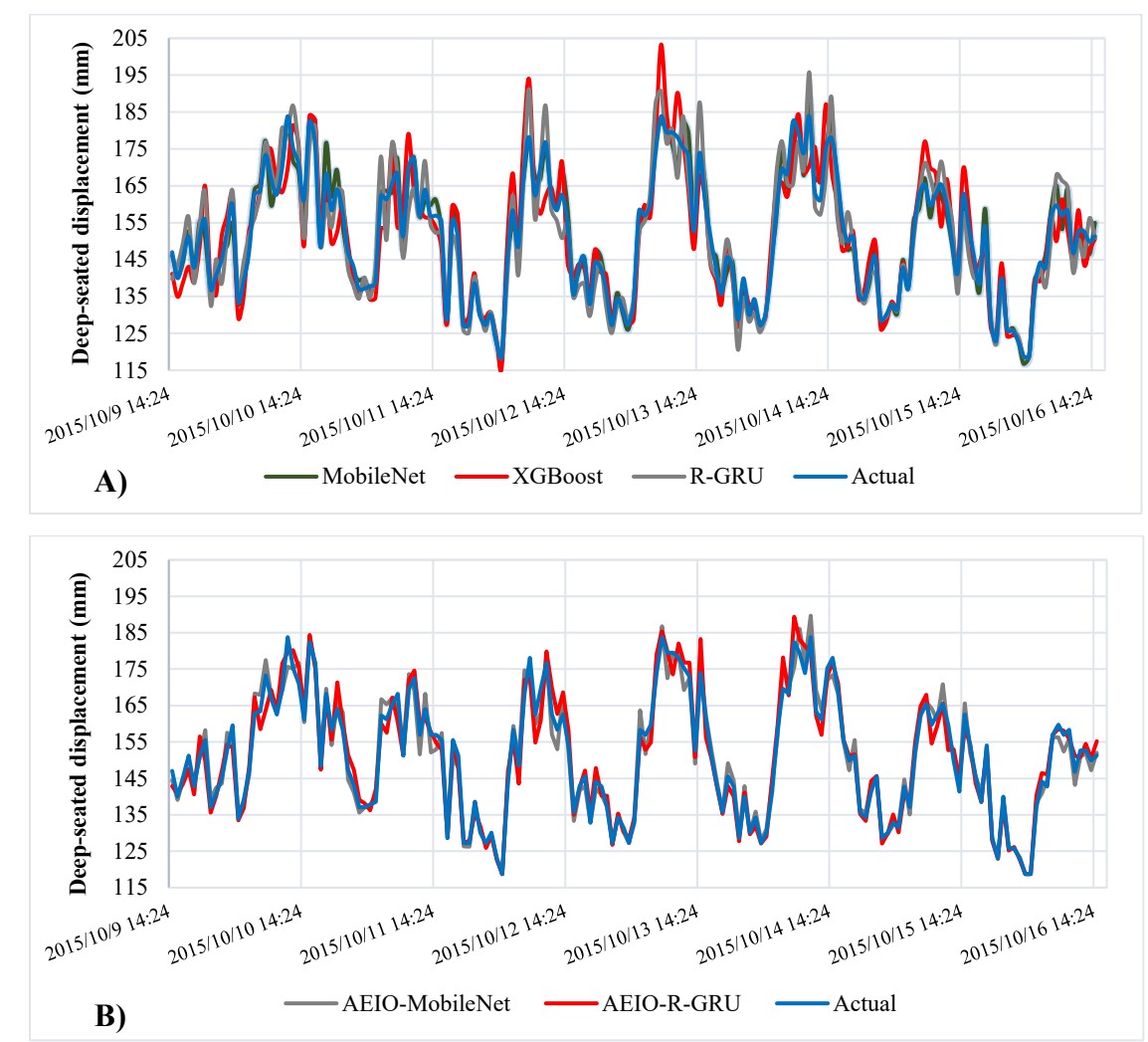

Figure 10. Graph comparing the real and predicted deep-seated landslide displacement: A) Prediction results of deep-seated landslide displacement by single AI models; B) Prediction results of deep-seated landslide displacement by AI models optimized using the AEIO algorithm.

**5.3 Discussion**

This study focuses on landslides in Lushan Mountain, Taiwan, intending to develop models to predict deep-seated landslide displacement for both 1-day and 7-day forecasts. These predictive models utilize input data such as the region's groundwater levels, temperature, and humidity. Accurately computing deep-seated landslide displacement offers several benefits. Firstly, it provides timely information for engineers to assess the resilience of structures and infrastructure in at-risk areas, facilitating the issuance of sensible warnings. Secondly, forecasting deep-seated landslide displacement offers insights into the severity of the disaster, aiding in effective evacuation and rescue planning.

Moreover, unlike AI models in previous studies (Balogun et al., 2021; Hakim et al., 2022; Jaafari et al., 2022), our research incorporates machine learning, time series deep learning, and CNN models, utilizing metaheuristic optimization algorithms to fine-tune their hyperparameters. However, the novelty

of our study lies in adopting pre-trained models, such as MobileNet, DenseNet, Inception, and VGG, rather than standard CNN models.

By employing various AI models, this study identifies the most effective model for predicting deep-seated landslides and offers a comprehensive overview of the performance of different AI models. Initially, machine learning models exhibited relatively high prediction errors, with MAPE ranging from 8.14% to 15.19%. This performance was generally lower than time-series deep learning models, which showed MAPEs ranging from 7.9% to 14.73%. The superior performance of the time series deep learning models is attributed to their ability to process sequential data and retain information from previous steps. This enables them to learn patterns from the dataset more effectively than traditional machine learning models.

Although time series deep learning models perform well, they fall short compared to CNN models. This disparity can be attributed to CNN's more advanced learning mechanism. The convolutional and pooling layers in CNN enable robust feature extraction from input data, with convolutional layers particularly effective at identifying complex patterns and subtle features in time series data, especially when spatial correlations are present. This capability allows CNNs to uncover critical features that other models may overlook.

The models developed in this study demonstrate predictive solid capabilities for deep-seated landslide displacement. Among them, the AEIO-MobileNet model is the most effective, achieving predictions with deficient error, indicated by a MAPE of 2.81%. However, these models' practical applicability in real-world scenarios must be improved due to the time-consuming processes involved in data collection, processing, and AI model operation, making timely predictions challenging. Meanwhile, there have been studies that successfully built real-time landslide detection systems (Wang et al., 2023; Das et al., 2020; C. et al., 2021). We acknowledge this limitation of our study. Therefore, future research endeavors will aim to address this issue.

The input data used for the AI models were selected because they significantly influence the likelihood of deep-seated landslides, as detailed in Section 4.2. However, a limitation of this study is that it needs to evaluate the relative importance of each input data type on prediction accuracy. Future research should explore the impact of different combinations of input data on AI model performance. This could help identify the significance of each input type and reveal the optimal combination of inputs to enhance prediction accuracy further.

**6. Conclusion**

This study addresses the persistent threat of large, slow-moving landslides, a primary concern due to their severe impact on lives and property. Employing various AI models, such as machine learning, time series deep learning, CNN models, and metaheuristic optimization algorithms, the research focuses on predicting deep-seated landslides at Lushan Mountain in Ren'ai Township, Nantou County. The study

aims to enhance early prediction accuracy by utilizing eight years of displacement and groundwater level data from Lushan Mountain and weather data from the POWER project. The predictions cover one-day and seven-day intervals, serving diverse purposes in landslide forecasting for timely evacuation. The research explores single and hybrid AI models to determine the most effective approach. The following conclusions are drawn from this research:

(a). CNN models optimized by the novel AEIO algorithm yield the best prediction results. In particular, AEIO-MobileNet predicts one-day-ahead displacement at the E-2 station with a MAPE score of only 2.81%, demonstrating high accuracy.

(b). While CNN models boast high prediction accuracy, their computational time is also considerable. Therefore, decisions regarding their usage should also consider real-world constraints.

(c). The AEIO-R-GRU model also yields reasonably good prediction results, although not on par with CNN models. The best result achieved by the AEIO-R-GRU model is a MAPE of 3.03% for one-day-ahead prediction at the E-2 station.

(d). The AEIO algorithm has successfully fine-tuned hyperparameters for AI models. Especially in the case of predicting one-day-ahead displacement, it has aided the MobileNet model in improving its predictive capability by 31.6%, enabling this model to provide more accurate predictions.

(e). The prediction results from the E-2 station consistently outperform those from the SAA station. This is attributed to the fact that data from the E-2 station has been collected over a longer and more comprehensive period.

(f). The study results demonstrate that AI models can accurately predict deep-seated landslide displacement, which can be implemented in real-world scenarios.

**Declare of Competing Interest**

The authors declare that there are no known conflicts of interest associated with this publication, and there has been no significant financial support for this work that could have influenced its outcome.

**Data Availability Statement**

The data and source codes supporting this study's findings are available at https://www.researchgate.net/profile/Jui-Sheng-Chou and from the corresponding author upon reasonable request.

**Acknowledgments**

The authors extend their gratitude to the National Science and Technology Council (NSTC), Taiwan, for financially supporting this research under NSTC grants 112-2221-E-011-033-MY3 and 111-2221-E-011-037-MY3. We also sincerely thank the Geological Survey and Mining Management Agency, Ministry of Economic Affairs, Taiwan, for providing favorable conditions for conducting this research.

**Author contribution**

Jui-Sheng Chou: conceptualization, methodology, supervision, manuscript writing, reviewing, and editing. Hoang-Minh Nguyen: data processing, coding, and manuscript writing. Huy-Phuong Phan: Data processing, coding, and manuscript writing. Kuo-Lung Wang: data preparation, supervision, and reviewing.

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

**Appendix A: Deep Learning Models for Time Series**

The architecture of an RNN includes an input layer, a hidden layer with a variable number of RNN cells, and an output layer designed for label identification based on future displacement values. Figure A1 illustrates the structure of simple RNNs.

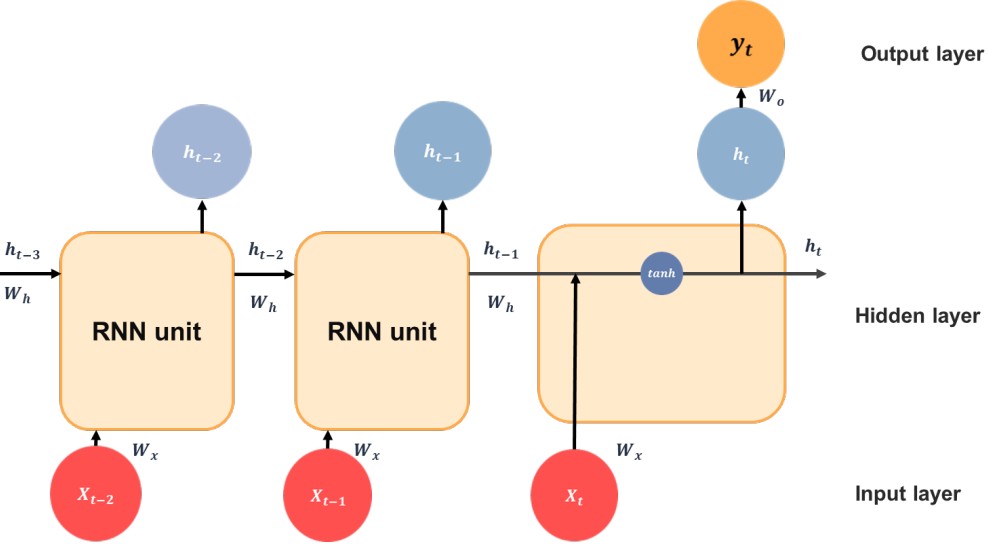

Figure A1. Structure of basic RNNs.

Each cell in an RNN acts as a memory cell, which is interconnected to enable the sequential transfer of time-dependent input information within a sliding window. This makes it possible to consider temporal correlations between events that may be widely separated in the time dimension. The following formula presents the hidden unit of standard RNNs at time t:

$$h_t = tanh(W_x * x_t + W_h * h_{t-1} + b) \tag{A1}$$

where $x_t$ is the input vector at time $t$; $h_t$ is the output vectors of hidden units for time $t$; $W_x$ and $W_h$ respectively indicate the input and interconnected weight matrices for the output of the hidden layer; $b$ is the bias term; and *tanh()* represents the hyperbolic tangent activation function, i.e., $tanh(x) = \frac{1-e^{2x}}{1+e^{2x}}$. The mechanism of learning over time steps, stored within cells, enables RNNs to effectively capture complex relationships between cells and time sequences. However, as the duration of dependencies increases, RNN models are susceptible to issues related to vanishing gradients. Therefore, RNNs are well-suited to learning time series involving short-term dependencies.

**Appendix B: Convolutional Neural Networks**

The architecture of a typical CNN, as illustrated in Figure B1, comprises an input layer (to receive image data), followed by hidden layers (including convolutional, pooling, and fully connected layers), and concludes with the output layers. As depicted in Figure B1, the complexity of CNN progressively increases from the convolutional layer to the fully connected (FC) layer. This design enables CNN to recognize relatively simple patterns (lines, curves, etc.) before progressing to capture more intricate

features (faces, objects, etc.), with the ultimate aim of extracting relevant information for accurate pattern
identification.

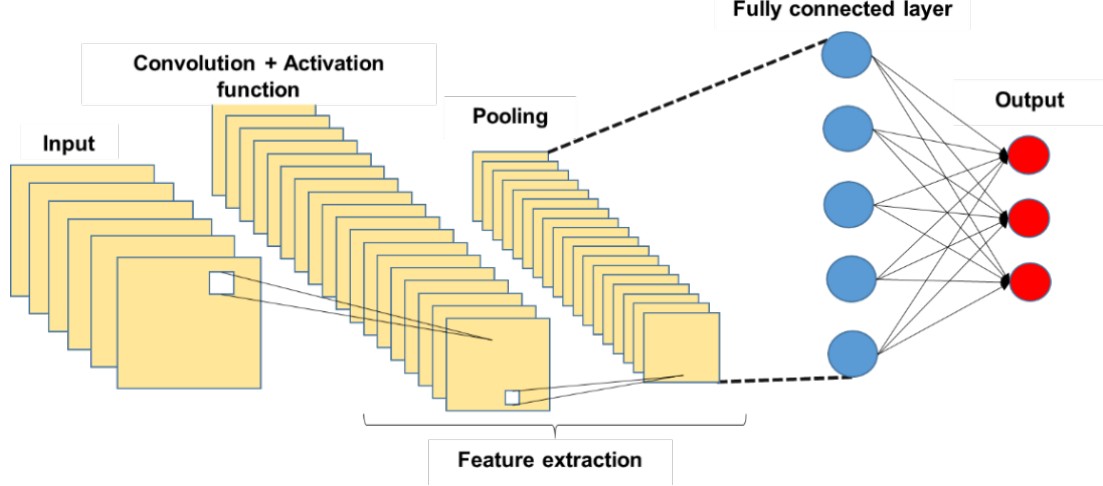


Figure B1. Structure of basic CNN.

As illustrated in Figure B2, the convolutional layer is responsible for most computations in the

network. This involves extracting local features from an image using a set of learnable filters known as
kernels. The behavior of the filter in the convolutional layer is influenced by two main factors: stride and
padding. Stride refers to the pixel shift of the filter across the image, while padding aims to preserve
information at the corners. In each iteration, a portion of the image is convolved with a filter to generate
a dot product of pixels within its receptive field. This process is replicated across the entire image to
produce a feature map. The convolution operation is defined as follows:
$C_i = b_i + \sum_{j=1}^{d_i} I_j * F_{ij} , \ i = 1 \dots d_c$                            (B1)
where $C_i$ is the output of the convolutional layer or feature map, $b_i$ is the bias, $d_i$ is the depth of input, $I_j$
is the input image, $F_{ij}$ is the filter, and $d_c$ is the depth of the convolutional layer.

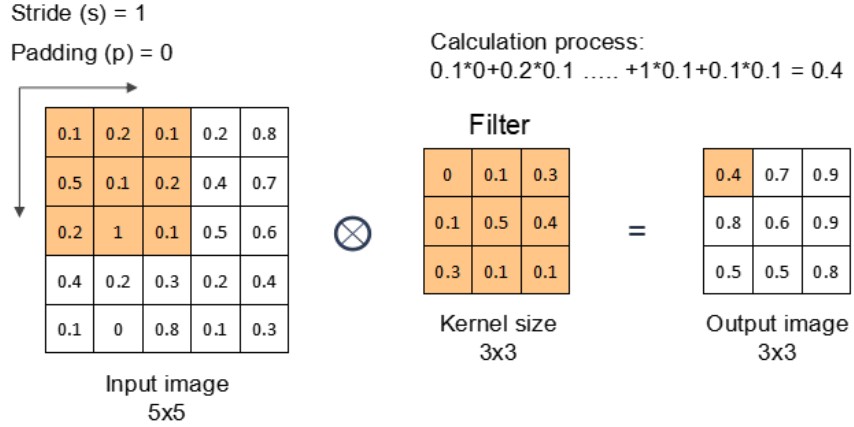


Figure B2. Processing flow in convolution layer.

The multiplicative operations are usually followed by an activation function (the final element in the
convolutional layer), which introduces nonlinearity and creates intricate mappings between network
inputs and outputs. The activation function can be defined as follows:
$$Y_i = f(C_i) \tag{B2}$$
where, $Y_i$ is the output of the convolutional layer after the activation function, and $f$ is the activation
function.
A rectified linear unit ReLU is a nonlinear CNN function with output $f(x) = max(0, x)$. A ReLU
converts all negative values to zero or returns the original input values if the input exceeds zero. ReLU is
only one of many activation functions; however, it has proven to be the most effective overall.
Pooling layers after the convolution layer can down-sample feature maps by summarizing features
within the coverage area of a 2-D filter to reduce sensitivity to feature location, thereby improving
resilience to changes in the position of features. Pooling layers also decrease the dimensions of the feature
map, reducing the number of parameters to be dealt with, thereby decreasing computational overhead.
Output dimensions from the pooling layer are computed as follows:
$$\frac{c_w - f_w + 1}{s} * \frac{c_h - f_h + 1}{s} * c_n \tag{B3}$$
where $c_n$ is the number of channels in the feature map and $f_w * f_h$ indicate the width and height of the
filter.
Max pooling and average pooling are commonly used in CNN. Max pooling accentuates salient
features by selecting the maximum value within the filter's coverage area. In contrast, average pooling
calculates the mean value within the exact location, providing a representative feature value. Illustrations
of max pooling and average pooling are presented in Figure B3.

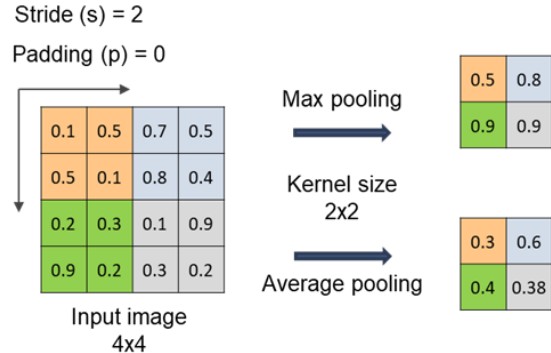


Figure B3. Max Pooling and Average Pooling.
The final stage of a CNN comprises a series of fully connected (FC) layers. After the convolution
and pooling operations, the feature map is flattened into a one-dimensional vector that connects to the FC
layers, resembling an ANN. FC layers identify specific features, each represented by a neuron. In
regression tasks, each neuron in the FC layer corresponds to a feature contributing to the final numerical
output. The value transmitted by each neuron indicates its significance toward the regression result. FC
layers are designed to predict the best continuous value for the target variable by combining and processing
these neuron outputs. Figure B4 illustrates the structure of an FC layer.

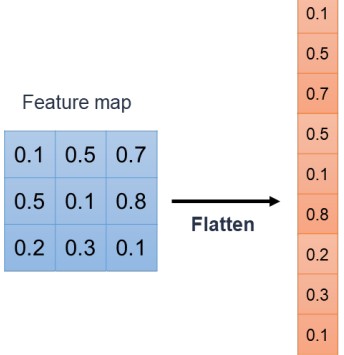


Figure B4. Structure of fully connected layer.