# Peer review of "Predicting Deep-Seated Landslide Displacement in Taiwan's Lushan Mountain through the"

_Natural Hazards and Earth System Sciences, 2024_

## Referee Comment (RC1)

**Review of Chou et al., Prediction Deep-Seated Landslide Displacements in Mountains through the Integration of Convolutional Neural Networks and Age of Exploration-Inspired Optimizer**

*Summary and Recommendations*

This paper by Chou et al. describes an effort to test the sensitivity of various machine learning models on forecasting deep-seated landslide displacement over single-day and weeklong timescales. The authors utilize two sets of extensometer data that record landslide displacement at Lushan Mountain in Taiwan over a period from 2009-2017, along with four records of groundwater well data and satellite-derived temperature and humidity data. Over this time, the extensometer data record multiple pulses of movement that appear to correspond to peaks in groundwater levels, suggesting a connection to pore-water pressure increases via rising water tables. The authors employ their record of time series data to train a bevy of various AI models, and then from the two top-performing models fine-tine their hyperparameters using a newly released optimization algorithm (the Age of Exploration-Inspired Optimizer, or AEIO). The authors find that: 1) many models perform well in forecasting landslide displacement although there are tradeoffs between accuracy and computation time (impressively low errors from ~4-7% in the best cases); and 2) the AEIO algorithm successfully reduces uncertainty in their top models.

Overall, the authors present a clear description of the AI models used in the analysis and show convincingly that for the study monitoring sites machine learning algorithms can indeed be used to accurately forecast landslide displacement, even at the multi-day time scale. Showing that these methods yield a ~5% error on a seven-day forecast of landslide displacement is highly impressive and has obvious societal relevance. The AEIO method (complete with a very fanciful Fig. 8) does appear to work well in reducing the prediction uncertainty for the top-performing models. Therefore, I think the paper succeeds in showing the practical utility of applying optimized AI-based methods to this type of extensometer data and the benefits of running an optimization scheme on improving model performance. As presented, however, the manuscript feels somewhat lopsided as there is comparatively little information about the landslide itself and any in-depth analysis on connections from the model(s) to the results. For example, how much does the choice of input parameters impact performance? Are four groundwater datasets necessary, or would one suffice? Does including humidity data actually help improve model results, or is it extraneous? These are the types of questions worth discussing that may help yield more insight and understanding that may expand the utility of these results beyond the authors' study data (and thus would be of increased relevance to the global NHESS readership). Beyond these primary concerns, there are a number of smaller line-by-line technical and editorial comments I provide below that warrant addressing by the authors. If the authors can address these comments, I think this manuscript will make a useful contribution to NHESS.

*Line-by-Line Comments*

1: I'm not sure the phrase "in Mountains" is necessary here.

2: I believe the word "an" should perceive "Age of Exploration-Inspired Optimizer"

9: Nothing is done in this manuscript to show that deep-seated landslides are becoming increasingly frequent due to changing climate patterns. Is there a reference the author can provide that shows this in order to justify its presence in the abstract? This is certainly a nuanced topic as projected climatic changes may impact different areas (and thus landslide-triggering potential) differently across the globe, and therefore it is difficult to make these blanket statements.

11: insert "by" after "displacements"

25: There are certainly more than 378 landslides recorded worldwide between 1997 and 2017. Is this a specific subset of slides from this study? If so, a little more context needs to be provided here on what this number represents.

35: The 10 m threshold for defining a deep-seated landslide seems arbitrary. Dou et al. (2015) use 10 m as an example in their example sketch (their Fig. 5), but they do not reference this as a specific genetic guideline. Please use a more appropriate definition here.

41-42: This sentence feels out of place here since the paragraph is just discussing background. It would fit better in the final paragraph of this section outlining the goals of the specific study (i.e., lines 63-76)

54: editorial suggestion – can remove "In contemporary times"

55: CNN has not been defined before the introduction of this acronym

64: The term "predict deep-seated landslides" sounds vague. Predicting incipient failure? Reactivation of an already-established failure? Please specify.

65-66: Please list references of pre-existing work that you are referencing here

67: Impressive! At what depth is the failure plane for each of these extensometers?

88-92: This section feels quite under-referenced, as there are numerous theoretical and observational examples of groundwater impacts on deep-seated landslide failure.

93-94: This is another purely editorial comment, but the citation style presented here could be more succinct. For example, "Similarly, Preisig (2020) developed…" rather than "Similarly, Presig developed a groundwater prediction… (Presig, 2020)." This same style is utilized throughout the manuscript

103-105: In what way did Lin et al. "somewhat overlook" the importance of hydrological conditions in landslide formation here? Please be specific.

110 (Figure 1). Where is the actual landslide here? Below the diagram? I find the arrow below the right diagram very confusing and vague. A schematic failure plane perhaps informed by the borehole data would be useful for clarifying what it is the authors are trying to illustrate here.

122: Numerical models can simulate many scales, not just the laboratory scale. Please fix.

125: Does this Mufundirwa et al. reference also utilize a numerical model? If not, this paragraph should perhaps speak to both laboratory and numerical studies.

130: editorial – can delete "Meanwhile," here

135-136: What are "micro-units" here?

140-142: The previous paragraphs have not demonstrated that these "conventional methods have shown limited success in handling big data…" More information needs to be provided in this or the previous paragraphs to provide justification for this argument.

154: Is there any discussion on why this model was the most successful?

163-164: Please define what the term "feature engineering" is here

166: these parameters (topographic slope and soil parameters) don't necessarily have to be one-dimensional. Topography can be 2-D and soil parameters can be 3-D (and perhaps even time-dependent).

168-169: from my limited understanding of AI-based models, most are black boxes and therefore disentangling physical processes can be difficult. I thought this was the domain of physics-informed neural networks?

184: "predicting landslide displacement" would be more specific here

Section 3.1 (Lines 218-277): This part confused me at first because CNN's deal with imagery and you are using time series vectors. It is later clarified in the paper that the time series data are converted to images for use with the models, but it would be worth stating something up front that vector data can also be utilized in this construct with the proper transformation.

250 (Fig. 3): the 3x3 kernel illustrated here is mislabeled as 2x2

292: It's not clear here why RNNS are well-suited to learning time series with short-term dependencies. Please clarify.

318-322 (Performance Metrics): If you are assigning a separate section to performance metrics, it would be good to describe what each one is and the benefits and drawbacks for each metric.

328: What exactly is a particle in this instance? Some context is needed here.

337 (Fig. 8): The red arcuate arrows that link the positional strategies appears to suggest that once one strategy is selected, the explorer goes from one strategy to the next when in fact they return to the middle after each time step (correct?). If that is the case, then the arrows should arc back down to the central location to reflect the decision-making process that occurs with each positional change.

361-372: These two steps need to be elaborated on a little bit more, as it is presented somewhat confusingly and the equations for (8) and (9) are identical.

388 (Fig. 10): Much more information is needed in the figure caption here, as the current captions are essentially vacant. Additionally, the map in (a) is missing crucial information such as latitude and longitude graticules, and contains extraneous information (e.g., random text and other symbols that are not defined). With regard to (b), was the landslide failure plane identified with these cores? Or is the failure plane depth only known in the extensometer boreholes? Please provide more information here or elsewhere in the manuscript.

407: Please cite the previous research here

407-412: I think the term "cleavage" is misused here. Do the authors mean "fracture"? Typically, cleavage refers to the tendency of a mineral to break along planes defined by crystal lattice structure and are typically not seen at the scale of an entire hillslope. Lastly, it would be worth putting these observation zones on the map of the landslide for reference.

426: How was the rainfall data measured? Via a local rain gauge? If so, can put it on the map as well.

438-442 (Figs. 12-14): It is very difficult to compare the time series data as all the axes are scaled differently. I strongly recommend making one three-panel figure that is aligned in the time dimension instead of three separate figures. I would also recommend putting the known storms from Table 2 as vertical bands on each subplot. This will really help unify the datasets and make it much easier for readers to discern how precipitation, groundwater levels, and landslide displacement are aligned.

446-447: Should be "June" instead of "August", otherwise the groundwater will be responding to a future event!

457-458: The groundwater levels that are apparently driving displacement here are 10s of meters below the ground surface (e.g., Fig. 14). Which impacts on soil structure by thermal processes are you referring to? Do thermal effects at this depth contribute to landsliding? Please provide context and references here to back up this statement or otherwise remove.

459-461: Please describe more the data used here? For example, is it daily data? What is the grid size? What is the measurement source?

488: Indeed! Having forecast data a week in advance would be extremely beneficial.

499: Specify process to be modeled (i.e., landslide displacement)

502: editorial comment – shouldn't end sentence with "…"

509 (Fig. 16): Very helpful flow chart

545: It would be helpful to have a figure showing a subset of the models plotted alongside the displacement data so readers could see how the differences in MAPE are actually reflected in the time series predictions

568: change "landslides" to "landslide displacement" or something similar

660-664: This is a nice motivating paragraph that belongs in the introduction and would help provide context for the study.

668-668: Are these models not considered "conventional"? If not, why not? Could also be specified earlier on in the manuscript.

678: Here again would be a great place to delve into the "why" a little bit more. Any thoughts why a certain class of models outperforms the others? This discussion section is quite short relative to the rest of this paper, and there are a lot of aspects to potentially discuss. Does withholding certain parameters (e.g., temperature, humidity, or both) impact the results substantially? If so, why might that be the case? Since so much work has been done to get to this stage of predictive success, a small amount of additional work may help elucidate the role of specific processes in aiding the predictability of landslide displacement in this context that could be useful for the broad readership of NHESS.

---

## Author Comment (AC1)

Summary of the Changes to Reviewer 1's Recommendations and Comments

**Journal: Natural Hazards and Earth System Sciences**

**Ref: NHESS-2024-86**

Title: Predicting Deep-Seated Landslide Displacements in Lushan Mountains through the Integration of Convolutional Neural Networks and an Age of Exploration-Inspired Optimizer

The authors appreciate the reviewer's valuable feedback. The summary of the changes based on the reviewer's recommendations & comments is listed below. All the revisions are TRACKED in the re-submitted WORD file along with marked RED COLOR for the ease of the reviewer's perusal. Our colleague, a native English speaker of BLUE COLOR, has corrected grammatical and writing style errors in the original version.

| Recommendations and Comments of Reviewer | Authors' Summary of the Changes |
|---|---|
| This paper by Chou et al. describes an effort to test the sensitivity of various machine learning models on forecasting deep-seated landslide displacement over single-day and weeklong timescales. The authors utilize two sets of extensometer data that record landslide displacement at Lushan Mountain in Taiwan over a period from 2009-2017, along with four records of groundwater well data and satellite-derived temperature and humidity data. Over this time, the extensometer data record multiple pulses of movement that appear to correspond to peaks in groundwater levels, suggesting a connection to pore-water pressure increases via rising water tables. The authors employ their record of time series data to train a bevy of various AI models, and then from the two top-performing models fine-tine their hyperparameters using a newly released optimization algorithm (the Age of Exploration-Inspired Optimizer, or AEIO). The authors find that: 1) many models perform well in forecasting landslide displacement although there are tradeoffs between accuracy and computation time (impressively low errors from ~4-7% in the best cases); and 2) the AEIO algorithm successfully reduces uncertainty in their top models. Overall, the authors present a clear description of the AI models used in the analysis and show convincingly that for the study monitoring sites machine learning algorithms can indeed be used to accurately forecast landslide displacement, even at the multi-day time scale. Showing that these | As authors, we wish to express our sincere gratitude to the reviewers for their time and effort in thoroughly evaluating our research. We are encouraged by the recognition that our study may contribute to NHESS. In response to the reviewers' insightful suggestions, we will revise our manuscript accordingly. The following sections will address each revision in detail. We hope that these updates will meet the reviewers' expectations and align with the high standards of NHESS for publication. |

methods yield a ~5% error on a seven-day forecast of landslide displacement is highly impressive and has obvious societal relevance. The AEIO method (complete with a very fanciful Fig. 8) does appear to work well in reducing the prediction uncertainty for the top-performing models. Therefore, I think the paper succeeds in showing the practical utility of applying optimized AI-based methods to this type of extensometer data and the benefits of running an optimization scheme on improving model performance. As presented, however, the manuscript feels somewhat lopsided as there is comparatively little information about the landslide itself and any in-depth analysis on connections from the model(s) to the results. For example, how much does the choice of input parameters impact performance? Are four groundwater datasets necessary, or would one suffice? Does including humidity data actually help improve model results, or is it extraneous? These are the types of questions worth discussing that may help yield more insight and understanding that may expand the utility of these results beyond the authors' study data (and thus would be of increased relevance to the global NHESS readership). Beyond these primary concerns, there are a number of smaller line-by-line technical and editorial comments I provide below that warrant addressing by the authors. If the authors can address these comments, I think this manuscript will make a useful contribution to NHESS.

| | |
|---|---|
| 1: I'm not sure the phrase "in Mountains" is necessary here.
2: I believe the word "an" should perceive "Age of Exploration-Inspired Optimizer". | We have identified inaccuracies in the title based on the reviewer's comments. We will replace the phrase "in Mountains" with "in Lushan Mountain" to provide readers with more precise information about the data collection and research location. Additionally, as suggested by the reviewer, we will add the word "an" before "Age of Exploration-Inspired Optimizer.".

**Predicting Deep-Seated Landslide Displacements in Lushan Mountains through the**
**Integration of Convolutional Neural Networks and an Age of Exploration-Inspired Optimizer**

Jui-Sheng Chou[1,*], Hoang-Minh Nguyen[1], Huy-Phuong Phan[1], Kuo-Lung Wang[2]
[1]Department of Civil and Construction Engineering, National Taiwan University of Science and Technology, Taipei, Taiwan
[2]Department of Civil Engineering, National Chi Nan University, Nantou, Taiwan
(jschou@mail.ntust.edu.tw; hoangminhkg1992@gmail.com; huyphuong777@gmail.com; klwang@ncnu.edu.tw)
*Correspondence e-mail address: jschou@mail.ntust.edu.tw |
| 9: Nothing is done in this manuscript to show that deep-seated landslides are becoming increasingly frequent due to changing climate patterns. Is there a reference the author can provide that shows this in | We greatly appreciate the reviewer's comment. The reviewer correctly pointed out that our study does not demonstrate the argument that deep-seated landslides are becoming more frequent due to |

| | |
|---|---|
| order to justify its presence in the abstract? This is certainly a nuanced topic as projected climatic changes may impact different areas (and thus landslide-triggering potential) differently across the globe, and therefore it is difficult to make these blanket statements.

11: insert "by" after "displacements" | changing climate patterns. As such, it is inappropriate to include this argument in the abstract, and we have revised the sentence accordingly. Additionally, we have added the word "by" after "displacement," as suggested by the reviewer.

**Abstract**
Landslides have caused substantial
damage to both human life and infrastructure in the past. Developing an early warning system for this type
of disaster is crucial to reduce its impact on society. This research contributes to developing predictive
early warning systems for deep-seated slope displacements by employing advanced computational models
for environmental risk management. Our novel framework integrates machine learning, time series deep |
| 25: There are certainly more than 378 landslides recorded worldwide between 1997 and 2017. Is this a specific subset of slides from this study? If so, a little more context needs to be provided here on what this number represents. | In this section, we aim to provide data to demonstrate that landslides have significant negative impacts on our lives. However, as suggested by the reviewer, it appears that the data used may not be accurate. Therefore, we have sought new data and revised this section accordingly.

**1. Introduction**
Landslides are among the most devastating natural disasters (Huang and Fan,
2013), claiming an average of over 4,000 lives annually worldwide between 2004 and 2010 (Petley, 2012).
Landslides represent a global hazard, particularly in developing countries, where rapid urbanization,
population growth, and significant land use changes occur (Caleca et al., 2024). The identification, |
| 35: The 10 m threshold for defining a deep-seated landslide seems arbitrary. Dou et al. (2015) use 10 m as an example in their example sketch (their Fig. 5), but they do not reference this as a specific genetic guideline. Please use a more appropriate definition here. | We fully agree with the reviewer that using the definition of "deep-seated landslide" from Dou et al. (2015) was inappropriate. Consequently, we have revised this paragraph to adopt the definition provided by Lin et al. (2013) and included the relevant references. We hope this revised definition offers greater clarity and accuracy, addressing the reviewer's concerns.

al., 2014). These issues are further exacerbated in countries with complex geological and climatic
conditions.
A deep-seated landslide involves the gradual and persistent displacement of a substantial amount of soil
and rock, which can escalate into a sudden and devastating event (Kilburn and Petley, 2003; Geertsema
et al., 2006; Chigira, 2009). Unlike shallow landslides, which typically affect surface layers to a depth of
a few meters, deep-seated landslides extend deeper, often exceeding 10 meters, and can involve the
movement of underlying bedrock (Lin et al., 2013). Predicting these events is challenging and costly (Thai
Pham et al., 2019). Therefore, extensive efforts have been made to predict such disasters throughout
history. One method that has been employed involves thoroughly examining the physical and geological |
| 41-42: This sentence feels out of place here since the paragraph is just discussing background. It would fit better in the final paragraph of this section outlining the goals of the specific study (i.e., lines 63-76) | We agree that the inclusion of this sentence in this paragraph is not appropriate as it only discusses the background of the study. Therefore, we will remove this sentence.

level of groundwater has been shown by numerous studies in the past to influence the mechanisms behind
landslide formation significantly (Miao and Wang, 2023; Preisig, 2020).
In pursuing a generalized approach to landslide forecasting, researchers have determined that the
critical factors associated with slope instability exhibit temporal variability, necessitating using time series |
| 54: editorial suggestion – can remove "In | The phrase "In contemporary times" has been |

| | |
|---|---|
| contemporary times" | removed according to the reviewer's suggestion. |
| | 58      One of the most effective solutions for constructing models to predict time series data involves
applying data-driven techniques. The advancement of computational capabilities has driven the
widespread adoption of data-driven machine-learning models over physics-based models. This shift is
based on the premise that the data used for slope monitoring originates from nonlinear systems (Zhou et
al., 2018).  An increasing array of novel data-driven solutions is being developed
to overcome the constraints of traditional machine-learning approaches. Among these data-driven
solutions, convolutional neural networks (CNNs) have emerged as one of the most effective methods.
These CNN models, which excel at automated feature extraction, can enhance efficiency in analyzing
complex datasets and improve the accuracy of prediction results (Alzubaidi et al., 2021). |
| 55: CNN has not been defined before the introduction of this acronym | We have added an additional sentence beforehand to clearly explain the abbreviation 'CNN' and to further elaborate on the paragraph's content.

One of the most effective solutions for constructing models to predict time series data involves
applying data-driven techniques. The advancement of computational capabilities has driven the
widespread adoption of data-driven machine-learning models over physics-based models. This shift is
based on the premise that the data used for slope monitoring originates from nonlinear systems (Zhou et
al., 2018).  An increasing array of novel data-driven solutions is being developed
to overcome the constraints of traditional machine-learning approaches. Among these data-driven
solutions, convolutional neural networks (CNNs) have emerged as one of the most effective methods.
These CNN models, which excel at automated feature extraction, can enhance efficiency in analyzing
complex datasets and improve the accuracy of prediction results (Alzubaidi et al., 2021). |
| 64: The term "predict deep-seated landslides" sounds vague. Predicting incipient failure? Reactivation of an already-established failure? Please specify. | We fully agree that the term "predict deep-seated landslides" is unclear. We will revise this term to "predict deep-seated displacement".

Leveraging the effective methodologies mentioned above, this study employs AI models optimized
by an innovative metaheuristic optimization algorithm to predict deep-seated  displacement on
the northern slope of Lushan Mountain in  Ren'ai Township, Nantou County. The geological |
| 65-66: Please list references of pre-existing work that you are referencing here | Thank you to the reviewer for this comment. It was an oversight on our part not to include the relevant references to support this point. We have now added the appropriate references, as shown below.

the northern slope of Lushan Mountain in  Ren'ai Township, Nantou County. The geological
characteristics of this area have undergone extensive research (Wang et al., 2015; Lin et al., 2020).
Previous studies have identified varying depths of the shear plane. Specifically, Wang et al. determined
the depth of the shear plane is 85m and 106m based on inclinometer data (Lin et al., 2020).This research
paper is firmly grounded in empirical evidence meticulously collected over eight years from |
| 67: Impressive! At what depth is the failure plane for each of these extensometers? | The geology and shear planes of the Lushan Mountain region have been studied previously, revealing shear planes at depths of 85m and 108m. We have incorporated this information into the manuscript as suggested by the reviewer.

the northern slope of Lushan Mountain in  Ren'ai Township, Nantou County. The geological
characteristics of this area have undergone extensive research (Wang et al., 2015; Lin et al., 2020).
Previous studies have identified varying depths of the shear plane. Specifically, Wang et al. determined
the depth of the shear plane is 85m and 106m based on inclinometer data (Lin et al., 2020).This research
paper is firmly grounded in empirical evidence meticulously collected over eight years from
research area and shows the distribution of four survey boreholes (G20, G21, G18, and G25) along the
slope. Regolith, slate, and meta-sandstone are three distinct lithological units revealed through drilling.
Additionally, the study by Lin et al. identified the depths of failure planes in these survey boreholes.
Specifically, boreholes G18 and G25 did not record any failure planes, while boreholes G20 and G21
recorded failure planes at depths of 85 meters and 106 meters, respectively. These failure planes were
identified based on inclinometer data from the corresponding study (Lin et al., 2020).
Initially, the thickness of the topmost regolith layer was found to be less than 10 meters. Secondly,
slate predominated, exhibiting a notable presence with sporadic evidence of weathering that resulted in |
| 88-92: This section feels quite under-referenced, as there are numerous theoretical and observational examples of groundwater impacts on deep-seated | We fully agree with the reviewer's comment. Accordingly, we have included examples from both theoretical and observational studies to clarify this |

| landslide failure. | point. |
|---|---|
| | 101  Casimiro, 2023; Jones et al., 2023). Among these, hydrological conditions, especially groundwater levels,
have been one of the most critical elements considered in studies related to landslide prediction. Numerous
studies have substantiated this point. For instance, research by Take et al. demonstrated that the distance
and velocity of landslides triggered under high antecedent groundwater conditions are significantly more
significant compared to scenarios with drier conditions (Take et al., 2015). Another study has shown that
the accumulation of water at a soil-bedrock contact can develop of positive pore water pressures, causing
landslides (Matsushi and Matsukura, 2007) (see Figure 1). Moreover, studies on past landslide events have
also demonstrated similar findings. example Examples of this research include the Tessina landslide in
northeastern Italy, where groundwater conditions triggered movement (Petley et al., 2005). Additionally,
the study by Keqiang et al. on water-induced landslides in the Three Gorges Reservoir project area
highlights the significant impact of hydrological conditions on the likelihood of such disasters (Keqiang
et al., 2015). |
| 93-94: This is another purely editorial comment, but the citation style presented here could be more succinct. For example, "Similarly, Preisig (2020) developed…" rather than "Similarly, Presig developed a groundwater prediction… (Presig, 2020)." This same style is utilized throughout the manuscript | We fully agree with the reviewer's suggestion and have revised the citation at this location to make it more concise.

Similarly, Preisig (2020) developed a groundwater prediction model for analyzing the stability of a
compound slide in the Jura Mountains (Preisig, 2020). Additionally, Srivastava et al. explored machine

In addition, we have used the citation style suggested by the reviewer for similar cases throughout the manuscript.

[revised manuscript text omitted]

| 103-105: In what way did Lin et al. "somewhat overlook" the importance of hydrological conditions in landslide formation here? Please be specific. | In fact, the research by Lin et al. has accounted for hydrological conditions in landslide formation. Therefore, we have revised the motivation section accordingly. Our research will incorporate the use of AI models to predict deep-seated displacement at Lushan Mountain, a task that has not yet been addressed by previous studies about landslides in this area. |
|---|---|
| | 122    heavy rainfall events. Lin et al. (2020) conducted in-depth studies on the mechanisms of landslide |
| | 123    occurrence based on the geological conditions of the area (Lin et al., 2020). While successfully providing |
| | 124    valuable insights into the evolution of deep-seated gravitational deformations, their research somewhat |
| | 125    overlooked the importance of hydrological conditions and groundwater levels in landslide formation, their |
| | 126    study focuses exclusively on employing traditional analytical methods in geological research, such as |
| | 127    analyzing data from geotechnical instruments and conducting geological borehole analysis. |
| | 128    Our research aims to adopt a novel approach compared to previous landslide studies at Lushan |
| | 129    Mountain by utilizing AI models and metaheuristic optimization algorithms. This research will utilize To |
| | 130    address the limitations of previous landslide research in the Lushan Mountain area, this study will explore |
| | 131    using hydrological weather conditions and groundwater levels as inputs for AI models to predict deep- |
| | 132    seated displacement, thus aiding in landslide forecasting in this region. |

| 110 (Figure 1). Where is the actual landslide here? Below the diagram? I find the arrow below the right diagram very confusing and vague. A schematic failure plane perhaps informed by the borehole data would be useful for clarifying what it is the authors are trying to illustrate here. | We have revised Figure 1 by removing the arrow and the text 'deep-seated slope failure,' and adding a label for the 'failure plane.' We hope these modifications meet the reviewer's expectations.
[Figure]
Figure 1. Schematic illustration showing the effects of groundwater on deep-seated slope failure |
|---|---|

| 122: Numerical models can simulate many scales, not just the laboratory scale. Please fix. | We have revised this section according to the reviewer's suggestion. |
|---|---|
| | 149    Moreover, physical-based numerical and laboratory modeling methods, which simulate phenomena |
| | 150    at a laboratory scale, are also gaining traction in landslide research. These methods aim to maintain |
| | 151    forecasts using various data types while reducing human workload and ensuring high accuracy. For |
| | 152    example, Mufundirwa et al. conducted a laboratory study to examine the effectiveness of the inverse |

| 125: Does this Mufundirwa et al. reference also utilize a numerical model? If not, this paragraph should perhaps speak to both laboratory and numerical studies. | We have revised this paragraph to include references to both laboratory and numerical studies, as suggested by the reviewer. |
|---|---|
| | 149    Moreover, physical-based numerical and laboratory modeling methods, which simulate phenomena |
| | 150    at a laboratory scale, are also gaining traction in landslide research. These methods aim to maintain |
| | 151    forecasts using various data types while reducing human workload and ensuring high accuracy. For |
| | 152    example, Mufundirwa et al. conducted a laboratory study to examine the effectiveness of the inverse |

| | |
|---|---|
| 130: editorial – can delete "Meanwhile," here | We have removed the term 'Meanwhile' and revised the sentence accordingly, as suggested by the reviewer.

reservoir in Italy (Mufundirwa et al., 2010). In another study, Wu (2010) employed the numerical
discontinuous deformation analysis method to simulate a blocky assembly's post-failure behavior,
incorporating earthquake seismic data  Another study follow this trend by Jiang
et al. (2011), who utilized fluid-solid coupling theory to simulate displacement and capture  the
interaction between fluid and solid materials . However, both numerical models and
laboratory modeling methods require substantial effort from researchers. These approaches demand deep |
| 135-136: What are "micro-units" here? | "micro-units" refer to microscopic components of the rock mass, a term delineated during the referenced study. We have added a concise explanation to clarify the meaning of this term in the manuscript:

acting on  slope behavior. Fu and Liao (2010) presented a technique for implementing the non-linear
Hoek-Brown shear strength reduction, determining the correlation between normal and shear stress based
on the Hoek-Brown criterion . Subsequently, the micro-units (microscopic
components of the rock mass) instantaneous friction angle and cohesive strength under specific stress
conditions are calculated. Although this approach effectively addresses cost and labor issues, it still |
| 140-142: The previous paragraphs have not demonstrated that these "conventional methods have shown limited success in handling big data…" More information needs to be provided in this or the previous paragraphs to provide justification for this argument. | The assertion that conventional methods show limited success in handling big data is not entirely complete or accurate. We have added more information in this section to explain the drawbacks of conventional methods and the necessity of using AI models in this research.

that indicate risk levels . Recently, Xu et al. (2018) utilized real-time remote
monitoring systems to measure internal stress, deep displacement, and surface strain. This data was used
to formulate forecasting models to assess slope stability, particularly in railway construction
. However, a common challenge with this method is the instability and frequent changes in the terrain
and geology of landslide-prone areas. This necessitates constant updates to the computational model,
which can be time-consuming and labor-intensive.
Moreover, physical-based numerical and laboratory modeling methods,
are also gaining traction in landslide research. These methods aim to maintain
reservoir in Italy (Mufundirwa et al., 2010). In another study, Wu (2010) employed the numerical
discontinuous deformation analysis method to simulate a blocky assembly's post-failure behavior,
incorporating earthquake seismic data  Another study follow this trend by Jiang
et al. (2011), who utilized fluid-solid coupling theory to simulate displacement and capture  the
interaction between fluid and solid materials . However, both numerical models and
laboratory modeling methods require substantial effort from researchers. These approaches demand deep
expertise and the development of complex models. More importantly, they rely heavily on assumptions
during the simulation process and may not accurately reflect real-world conditions, leading to significant
errors.
Stability analysis is another commonly used method related to physics, which evaluates the forces
acting on  slope behavior. Fu and Liao (2010) presented a technique for implementing the non-linear
Hoek-Brown shear strength reduction, determining the correlation between normal and shear stress based
on the Hoek-Brown criterion . Subsequently, the micro-units (microscopic
components of the rock mass) instantaneous friction angle and cohesive strength under specific stress
conditions are calculated. Although this approach effectively addresses cost and labor issues, it still
heavily relies on the  researcher's assumptions and is limited by the ability to utilize only a
small portion of data from the research area.
However, in landslide studies, monitoring data is constantly updated, generating large volumes daily
with a temporal relationship (Peternel et al., 2022; Corominas et al., 2014).
As previously mentioned,
using conventional methods in landslide research presents numerous challenges whenever data changes
or gets updated. In contrast, AI models can overcome these difficulties by automatically learning to
identify connections between input and output data. AI models can be updated to incorporate additional
input variables and handle increasing amounts of data flexibly in response to real-world conditions.
Therefore, AI models will be utilized in this research instead of conventional methods. |
| 154: Is there any discussion on why this model was | Pham et al. (2016) did not explain why the support |

| | |
|---|---|
| the most successful? | vector machine (SVM) model provided the most accurate predictions compared to other models. They simply noted that the superior performance of the SVM model was consistent with conclusions from numerous past studies. From our perspective, the study by Pham et al. did not employ methods to search for optimal hyperparameters to minimize the errors of the AI models (such as grid search or metaheuristic optimization algorithms). This oversight resulted in the models not operating under optimal conditions. Consequently, determining the truly effective model in their study remains challenging. Therefore, in the reference section of our current research, we can only mention the SVM as the most effective model according to their conclusions without further explanation due to the lack of optimization methods. We hope the reviewers understand this challenge we face. |
| 163-164: Please define what the term "feature engineering" is here | Adding further explanation for the term "feature engineering" will enhance readers' understanding of this study. We have included the requested annotation below as per the reviewer's suggestion.

interrelationships, mainly when data availability is limited (Zhang et al., 2020). Finally, feature
engineering (the process of selecting and transforming input variables to enhance the performance of the
models) is computationally intensive and labor-intensive, limiting its applicability when rapid forecasting
is required. |
| 166: these parameters (topographic slope and soil parameters) don't necessarily have to be one-dimensional. Topography can be 2-D and soil parameters can be 3-D (and perhaps even time-dependent).
168-169: from my limited understanding of AI-based models, most are black boxes and therefore disentangling physical processes can be difficult. I thought this was the domain of physics-informed neural networks? | We fully agree with the reviewer's comments regarding the inaccuracies in this paragraph. We have revised the paragraph as follows:

Alongside the
aforementioned machine learning models, a range of neural network models, from simpler ones like
Artificial Neural Networks (ANN) to more advanced approaches such as Deep Neural Networks (DNNs)
and CNN  are also  employed in research related to landslide (Kumar et al., 2017; Zheng et al., 2022)
Notably, CNN models have become increasingly popular and are widely used
in research related to this disaster. CNN models often yield superior predictive results than other models
in landslide susceptibility assessment and displacement prediction (He et al., 2024).
|
| 184: "predicting landslide displacement" would be more specific here | We have revised the term "landslide prediction" to "predicting landslide displacement" according to the reviewer's request.

employs a combination of machine learning methods, time-series deep learning, and CNN models to
compare and determine the most suitable model for predicting landslide displacement .
Therefore, our research aims to address this gap. |
| Section 3.1 (Lines 218-277): This part confused me at first because CNN's deal with imagery and you | We fully agree with the reviewer's comment and have added a paragraph to further elaborate on this |

| | |
|---|---|
| are using time series vectors. It is later clarified in the paper that the time series data are converted to images for use with the models, but it would be worth stating something up front that vector data can also be utilized in this construct with the prop | point, as detailed below.

CNN models are typically used for image processing tasks. However, the input data for this study is
in numerical and vector form. Therefore, several transformation steps are required to convert this
numerical and vector data into image data suitable for CNN input. Detailed information about these
transformation steps can be found in the study by Chou and Nguyen 2023 (Chou and Nguyen, 2023). |
| 250 (Fig. 3): the 3x3 kernel illustrated here is mislabeled as 2x2 | The incorrect annotation of the kernel has been corrected in the revised version of this figure.

[Figure]

Figure A-2. Processing flow in convolution layer. |
| 292: It's not clear here why RNNS are well-suited to learning time series with short-term dependencies. Please clarify. | We have provided additional reasoning as to why RNNs are well-suited for learning time series with short-term dependencies, as requested by the reviewer.

hidden layer; $b$ is the bias term; and _tanh()_ represents the hyperbolic tangent activation function, i.e., $tanh(x) = \frac{1-e^{2x}}{1+e^{2x}}$ . The mechanism of learning over time steps, stored within cells, enables RNNs to capture complex relationships between cells and time sequences effectively. However, as the duration of dependencies increases, RNN models are susceptible to issues related to vanishing gradients (Bengio et al., 1994). Therefore, RNNs are well-suited to learning time series involving short-term dependencies. |
| 318-322 (Performance Metrics): If you are assigning a separate section to performance metrics, it would be good to describe what each one is and the benefits and drawbacks for each metric. | We greatly appreciate this feedback from the reviewer. Performance metrics serve as evaluation criteria for AI models in this study. Providing comprehensive information about them will enhance readers' understanding of this research. Therefore, we have incorporated detailed information about these performance metrics in Section 3.4.2 as follows:

**3.4.2 Performance Metrics**
This study utilized four widely recognized performance measures to assess the  model's
effectiveness in prediction accuracy (Chou and Nguyen, 2023). The measures included mean absolute
error (MAE), mean absolute percentage error (MAPE), and root mean square error (RMSE).
MAE represents the mean of absolute errors, calculated as the average of the absolute
differences between actual and predicted values. Its advantage lies in its simplicity, which
provides a straightforward measure of average prediction error. However, a drawback of MAE is
its insensitivity to more significant errors, so it may not effectively highlight differences between
models when significant errors are present. It is defined as:
$MAE = \frac{1}{n}\sum_{i=1}^{n}|y_i - \hat{y}_i|$     (5)
where $n$ is the number of predictions, $y_i$ is the $i$th forecasted value, and $\hat{y}_i$ is the corresponding $i$th
actual value.
MAPE quantifies the average absolute error ratio—derived from the differences between
actual and forecasted values—to the actual value. It provides a clear metric in percentage terms,
facilitating straightforward interpretation across various datasets. However, MAPE's limitation
arises from its sensitivity to zero values in the actual data, which can become undefined or
impractical to compute, limiting its utility in scenarios involving zero or near-zero actual values.
The expression for MAPE is as follows: |

| | |
|---|---|
| | 393   $MAPE = \frac{1}{n}\sum_{i=1}^{n}\left|\frac{y_i - \hat{y}_i}{y_i}\right|$      (6) |
| | 394   where $n$ is the number of predictions, $y_i$ is the $i^{\text{th}}$ forecasted value, and $\hat{y}_i$ is the corresponding $i^{\text{th}}$ |
| | 395   actual value. |
| | 396     RMSE represents the square root of the average squared error between actual and forecasted |
| | 397   values and is widely used for its ability to indicate the dispersion of errors. This method captures |
| | 398   the magnitude and direction of errors, making it practical for assessing overall prediction |
| | 399   accuracy. However, RMSE tends to be more sensitive to outliers and significant errors than MAE |
| | 400   due to its squaring of errors during computation. This sensitivity can disproportionately affect its |
| | 401   evaluation in datasets with extreme values. The expression for RMSE is as follows: |
| | 402   $RMSE = \sqrt{\frac{1}{n}\sum_{i=1}^{n}(y_i - \hat{y}_i)^2}$      (7) |
| | 403   where $n$ is the number of predictions, $y_i$ is the $i^{\text{th}}$ forecasted value, and $\hat{y}_i$ is the corresponding $i^{\text{th}}$ |
| | 404   actual value. |
| 328: What exactly is a particle in this instance? Some context is needed here. | We fully agree with the reviewer's comment. Our manuscript lacked sufficient detail regarding the term 'particle.' We have now added an explanatory section on this term in Section 3.5.

**3.5 Age of Exploration-Inspired Optimizer**
This study employs a range of AI models to forecast deep-seated displacement in mountainous
regions. To enhance the prediction accuracy of these AI models, the study incorporates a novel
metaheuristic optimization algorithm known as the Age of Exploration-Inspired Optimizer (AEIO).
Developed by Chou and Nguyen in 2024, this algorithm has demonstrated high effectiveness in fine-
tuning the hyperparameters of AI models. This algorithm treats each particle in the search domain as an
explorer. The movement of particles toward regions with higher fitness values parallels the exploratory
activities of the Age of Exploration, where explorers sought ideal locations for establishing colonies. In
this study, each particle represents a set of hyperparameters, with the ultimate goal of the search process
being to identify the optimal particle or hyperparameter set that minimizes prediction error for AI models.
Figure 3 illustrates the AEIO algorithm. |
| 337 (Fig. 8): The red arcuate arrows that link the positional strategies appears to suggest that once one strategy is selected, the explorer goes from one strategy to the next when in fact they return to the middle after each time step (correct?). If that is the case, then the arrows should arc back down to the central location to reflect the decision-making process that occurs with each positional change. | We have revised the illustrative figure for the AEIO algorithm. Specifically, we removed the red arcuate arrows linking the positional strategies to prevent any misunderstanding for the reader. Additionally, we added bidirectional arrows from the action of choosing the strategy to each colony search action. Furthermore, we included arrows around the central image of the explorer-choosing a strategy, indicating that the search process repeats with each iteration.

[Figure]
 |
| 361-372: These two steps need to be elaborated on a little bit more, as it is presented somewhat confusingly and the equations for (8) and (9) are identical. | We acknowledge that the two equations mentioned above are quite similar. The only difference between them lies in the values $x_{i,d}(t)$ and $x_{1,d}(t)$. Despite this slight variation in the formula, the mechanisms of the two movements are fundamentally different. |

One equation guides the current particle towards the best particle, while the other directs the current particle in a direction based on the distance of a random particle from the best one. We have added annotations in the explanations of the formulas. These annotations clearly specify the mathematical notation for each type of particle in the explanations. We hope that this addition will make the movement mechanisms of the particles more comprehensible.

•   **Explorers follow general trends**

The explorer choosing this movement type will calculate the distance from their location $x_{i,d}(t)$ to the center of all other explorers $(Meanvl_d(t))$, then attempt to move towards that central point in the hope of finding a better location with the potential to establish a new colony. The following formula determines the explorer's position after the movement:

$x_{i,d}(t+1) = x_{i,d}(t) + \alpha * \left(Meanvl_d(t) - x_{i,d}(t)\right) \times rand(0,1) \times R$       (8)

$Meanvl_d(t) = \dfrac{x_{1,d}(t) + x_{2,d}(t) + \cdots + x_{n_{Pop},d}(t)}{n_{Pop}}$       (9)

where $d = 1,2,\ldots D$; $D$ is the number of dimensions; $i = 1,2,\ldots n_{pop}$; $n_{Pop}$ is the total number of explorers; $t = 1,2,\ldots MaxIt$ is the number of iterations; $MaxIt$ is the maximum value of iteration; $\alpha$ is a parameter for adjusting the particle's movement toward the centroid position (usually equals 3).

$Meanvl_d(t)$ is the centroid of all particles in dimension $d$. $rand(0,1)$ is the random number in the range

[0,1]. $R$: a number that equals 1 or 2 depending on the value of $\underline{rand}(0, 1)$ per the equation. $R =$

$round(1 + rand(0,1) \times 1)$, $x_{i,d}(t)$ is the location of particle $i$ in iteration $t$, $x_{i,d}(t+1)$ is the location of particle $i$ in iteration $(t+1)$.

•   **Explorers follow three other peers**

Explorers employing this movement method will calculate the average position of three randomly selected other explorers $\left(\dfrac{x_{1,d}(t) + x_{2,d}(t) + x_{3,d}(t)}{3}\right)$ and then move toward this newly calculated average position. The explorer's new position is computed using the following formula:

$x_{i,d}(t+1) = x_{i,d}(t) + \left(\dfrac{x_{1,d}(t) + x_{2,d}(t) + x_{3,d}(t)}{3} - x_{i,d}(t)\right) \times rand(0,1) \times R$       (10)

where: $x_{1,d}(t)$, $x_{2,d}(t)$ and $x_{3,d}(t)$ are three random explorers in dimension $d$ at iteration $t$, $d = 1,2,\ldots D$;

$D$ is the number of dimensions; $i = 1,2,\ldots n_{pop}$; $n_{Pop}$ is the total number of explorers; $t = 1,2,\ldots MaxIt$

is the number of iterations; $MaxIt$ is the maximum value of iteration.

•   **Explorers follow the best one**

According to this strategy, the explorer $(x_{i,d}(t))$ will move closer to the position of another explorer currently holding the best position $(Best_d(t))$, as determined by the following formula:

$x_{i,d}(t+1) = x_{i,d}(t) + \left(Best_d(t) - x_{i,d}(t)\right) \times rand(0,1) \times R$       (11)

where: $Best_d(t)$ represents the position of the particle with the best fitness in dimension d at iteration t, the parameters $d$ and $t$ hold the same significance as defined in Equation 10.

•   **Explorers follow guidance from another one**

Explorers in favorable positions with access to information can execute this movement strategy. In this scenario, explorers $(x_{i,d}(t))$ will consult with  another explorer. The consulted explorer will compare their direction and distance to the best individual, who holds the most favorable position

$(Best_d(t))$ and guide the inquirer. This algorithm assumes that the inquirer can be any explorer, i.e., a random explorer $(x_{1,d}(t))$. The following formula describes how to calculate the new position of the explorer following this strategy:

$x_{i,d}(t+1) = x_{i,d}(t) + \left(Best_d(t) - x_{1,d}(t)\right) \times rand(0,1) \times R$       (12)

where: $x_{1,d}(t)$ is a random explorer in dimension $d$ at iteration $t$. the parameters $d$ and $t$ hold the same significance as defined in Equation 10.
* * *
| | |
|---|---|
| 388 (Fig. 10): Much more information is needed in the figure caption here, as the current captions are essentially vacant. Additionally, the map in (a) is missing crucial information such as latitude and longitude graticules, and contains extraneous information (e.g., random text and other symbols that are not defined). With regard to (b), was the landslide failure plane identified with these cores? Or is the failure plane depth only known in the | We have made several adjustments to Figure 10 and Figure 11. Specifically, in Figure 10, we have added information on latitude and longitude. Additionally, we have removed unnecessary details (e.g., random text and undefined symbols) from Figure 10. |

| | |
|---|---|
| extensometer boreholes? Please provide more information here or elsewhere in the manuscript. |
[Figure]

Image source: Imagery ©2022 CNES/Airbus, Maxar Technologies, Map data ©2022 Google
Figure 5. Locations of measurement devices

[Figure]

Figure 6. Illustration of geological drilling survey

For Figure 11, we primarily use this image to provide readers with geological information about the area.

These survey boreholes utilize data inherited from the study by Lin et al. (2020), which provides a detailed account of the failure plane depths. We have included information on the depth of the failure planes for each survey borehole in the manuscript and added a citation to the previous study, allowing readers to seek further details on these surface plane determinations.

research area and shows the distribution of four survey boreholes (G20, G21, G18, and G25) along the
slope. Regolith, slate, and meta-sandstone are three distinct lithological units revealed through drilling.
Additionally, the study by Lin et al. identified the depths of failure planes in these survey boreholes.
Specifically, boreholes G18 and G25 did not record any failure planes, while boreholes G20 and G21
recorded failure planes at depths of 85 meters and 106 meters, respectively. These failure planes were
identified based on inclinometer data from the corresponding study (Lin et al., 2020).
Initially, the thickness of the topmost regolith layer was found to be less than 10 meters. Secondly, |
| 407: Please cite the previous research here | We have added additional citations in this paragraph as per the reviewer's request.

Previous research has detected signs of brittle deformation in the area. These indications include
chevron folds within cleavages, visible cracks, and intricate jigsaw puzzle-like patterns at the head of the
rock formations. Overturned and flexural toppling cleavages are prevalent towards the toe of the slope.
Additionally, kink bands are observable on  fractures  recently
undergoing flexural folding along the eastern boundary. Notably, horizontal cleavages near the toe region
also exhibit inter-cleavage gouges. Further details on this geological information can be found in the study
by Lin et al. (2020). These instances highlight the potential for significant geological changes and
landslide risk in this region. |

| | |
|---|---|
| 407-412: I think the term "cleavage" is misused here. Do the authors mean "fracture"? Typically, cleavage refers to the tendency of a mineral to break along planes defined by crystal lattice structure and are typically not seen at the scale of an entire hillslope. Lastly, it would be worth putting these observation zones on the map of the landslide for reference. | We greatly appreciate the reviewers for this suggestion; we fully agree that the term "fracture" is more appropriate than "cleavage" and we have made the corresponding change.

Previous research has detected signs of brittle deformation in the area. These indications include
chevron folds within cleavages, visible cracks, and intricate jigsaw puzzle-like patterns at the head of the
rock formations. Overturned and flexural toppling cleavages are prevalent towards the toe of the slope.
Additionally, kink bands are observable on  fractures  recently
undergoing flexural folding along the eastern boundary. Notably, horizontal cleavages near the toe region
also exhibit inter-cleavage gouges. Further details on this geological information can be found in the study
by Lin et al. (2020). These instances highlight the potential for significant geological changes and
landslide risk in this region.

In response to the suggestion to display observation zones on the map, we have included them in Figure 10. In addition to showing the locations of the boreholes and data collection sites, Figure 10 delineates the areas prone to deep-seated landslides, which represent the observation zones.

[Figure]

Image source: Imagery ©2022 CNES/Airbus, Maxar Technologies, Map data ©2022 Google
Figure 5. Locations of measurement devices |
| 426: How was the rainfall data measured? Via a local rain gauge? If so, can put it on the map as well. | Rainfall data for this study were collected using a rain gauge installed on-site. The location of the rain gauge has been annotated on the map in Figure 10.

[Figure]

Image source: Imagery ©2022 CNES/Airbus, Maxar Technologies, Map data ©2022 Google
Figure 5. Locations of measurement devices |
| 438-442 (Figs. 12-14): It is very difficult to compare the time series data as all the axes are scaled differently. I strongly recommend making one three- | We have revised Figures 12-14 as per the author's request. Specifically, we merged all three original figures into one and placed them on a single |

| | |
|---|---|
| panel figure that is aligned in the time dimension instead of three separate figures. I would also recommend putting the known storms from Table 2 as vertical bands on each subplot. This will really help unify the datasets and make it much easier for readers to discern how precipitation, groundwater levels, and landslide displacement are aligned. | timeline. Additionally, we added a graph to depict the precipitation of recorded heavy rainfall in the studied area.

Placing all graphs on the same timeline facilitates easier tracking of concurrent data variations for readers. Moreover, it highlights the relationships between different datasets.

[Figure]

Figure 7. Unified timeline visualization of data in this study.
A) Precipitation of recorded heavy rainfall in the studied area; B) Measured displacements from extensometer SAA C)
Measured displacements from extensometer E_2; D) Groundwater levels at stations A-17, A-18-2, A-20, and A-24. |
| 446-447: Should be "June" instead of "August", otherwise the groundwater will be responding to a future event! | We sincerely apologize for this confusion. The error is corrected as below.

increases coinciding with periods of pronounced fluctuations in groundwater levels. Specifically, in June
2012, there was a notable surge in groundwater levels attributed to heavy rainfall from June 8, 2012, to
June 17, 2012, totaling 1029 mm over 219 hours (as indicated in Table 2 and Figure 7A). The abnormal
rise in groundwater levels caused a structural alteration in the area's soil, consequently amplifying deep- |
| 457-458: The groundwater levels that are apparently driving displacement here are 10s of meters below the ground surface (e.g., Fig. 14). Which impacts on soil structure by thermal processes are you referring to? Do thermal effects at this depth contribute to landsliding? Please provide context and references here to back up this statement or otherwise remove. | In this study, we incorporated temperature as an input for AI models to predict deep-seated landslides, due to its significant impact on pore water pressure and effective frictional resistance forces, which in turn affects soil strength. We have included several citations from past research to substantiate this argument, as outlined below.

In addition to groundwater level data, weather factors such as temperature and humidity are also
utilized as input data for the prediction model. This study includes temperature as an input variable for AI
models to predict deep-seated displacement due to its impact on soil structure. Elevated temperatures can
cause thermal expansion of soil particles, which can increase pore water pressure and reduce effective
frictional resistance forces (Pinyol et al., 2018). Additionally, previous research has shown a relationship
between temperature and the likelihood of landslides in clay-rich soils, which are also present in the
geological composition of Lushan Mountain (Shibasaki et al., 2017; Loche and Scaringi, 2023).These
factors significantly impact the soil structure and can trigger substantial displacement or landslides. |

| | |
|---|---|
| 459-461: Please describe more the data used here? For example, is it daily data? What is the grid size? What is the measurement source? | We have provided additional information to help readers better understand the data collected from the website https://power.larc.nasa.gov.

This study collected groundwater level and displacement data on-site using sensors. Furthermore,
temperature and humidity data were obtained from the website https://power.larc.nasa.gov. This dataset
is part of the Prediction of Worldwide Energy Resource (POWER) project, developed by the National
Aeronautics and Space Administration (NASA) of the United States. The POWER solar data derives from
satellite observations, which are used to infer surface insolation values. Meteorological parameters are
sourced from the Modern-Era Retrospective analysis for Research and Applications, Version 2 (MERRA-
2) assimilation model. The primary solar data is available with a global resolution of 1° x 1°
latitude/longitude, while the meteorological data is provided at a finer resolution of ½° x ⅝°
latitude/longitude. Users can download the data hourly, daily, or monthly through this website. |
| 488: Indeed! Having forecast data a week in advance would be extremely beneficial. | We sincerely appreciate the reviewer's acknowledgment. We hope that these predictive results will contribute to the advancement of forecasting methods, ultimately aiding in the evacuation efforts prior to landslide disasters. |
| 499: Specify process to be modeled (i.e., landslide displacement) | We have corrected this sentence for accuracy, as the focus of the study is on predicting deep-seated displacement rather than deep-seated landslides.

**4.1 Model Establishment**
Predicting deep-seated displacement  at Lushan Mountain is undoubtedly highly
challenging, given that such landslides depend on numerous factors. Therefore, multiple methods will be |
| 502: editorial comment – shouldn't end sentence with "…" | We have removed the ellipsis at the end of this section as requested by the reviewer.

Predicting deep-seated displacement  at Lushan Mountain is undoubtedly highly
challenging, given that such landslides depend on numerous factors. Therefore, multiple methods will be
employed simultaneously to identify the optimal AI model for prediction. These methods include single
machine learning, time series deep learning, CNN, and hybrid models |
| 509 (Fig. 16): Very helpful flow chart | We sincerely appreciate the reviewer's praise. We consistently strive to use visuals and diagrams to convey our research, aiming to make it more accessible and comprehensible for readers. |
| 545: It would be helpful to have a figure showing a subset of the models plotted alongside the displacement data so readers could see how the differences in MAPE are actually reflected in the time series predictions | We fully agree with the reviewer's suggestion. Including a figure that illustrates the temporal variation in the predicted deep-seated displacements by different models will help readers clearly see how the differences in MAPE are actually reflected in the time series predictions. However, given the extensive number of AI models used in this study, displaying the prediction results of all models would increase the complexity of the charts, making it challenging to discern the differences in the models' performance. Therefore, we have chosen to display the displacement predictions of the most representative models, including the best machine learning model (XGBoost), the best time-series deep learning model (R-GRU), the best CNN model (MobileNet), and the best hybrid models (AEIO-MobileNet and AEIO-R-GRU). |

a) Prediction results of deep-seated displacement by single AI models.

b) Prediction results of deep-seated displacement by AI models optimized using the AEIO algorithm.

Figure 10. Graph comparing the real and predicted deep-seated displacement.

| 568: change "landslides" to "landslide displacement" or something similar | We have revised the term "deep-seated landslide" to "deep-seated displacement" following the reviewer's suggestion.

**4.2.2 Best AI Model Finetuned by AEIO Algorithm**
This section will focus on fine-tuning the hyperparameters of the numerical model to enhance its
performance in predicting deep-seated  displacement. The AEIO algorithm will fine-tune the
hyperparameters of the study's best numerical AI model, the R-GRU model. Details regarding the names
and search ranges of the hyperparameters are outlined in Table 8. The objective function of the AEIO |
| 660-664: This is a nice motivating paragraph that belongs in the introduction and would help provide context for the study. | We agree that including this information in the introduction will help clarify the context of our research and enable readers to better understand the benefits of these predictive models. Therefore, we have incorporated this information into the final paragraph of the introduction.

This study represents the first instance of AI models being utilized to predict deep-seated landslides
in Lushan Mountain. Additionally, it marks the inaugural application of AEIO for fine-tuning AI models
in landslide-related research. Our findings provide a valuable resource for civil engineers, contractors, and
inspectors involved in the planning and monitoring of construction projects in landslide-prone areas.
Predicting the likelihood of landslide events can help minimize property loss, guide schedule adjustments,
improve work safety, and ensure smooth traffic flow during critical periods. Additionally, understanding
internal displacements provides engineers with precise data to evaluate the resilience of structures and
infrastructure in vulnerable areas, enabling the issuance of prudent warnings. |
| 668-668: Are these models not considered "conventional"? If not, why not? Could also be specified earlier on in the manuscript. | We sincerely apologize for this oversight; the term "conventional" should not be used for CNN models for the following reasons:
- "Conventional models" refer to traditional, simple machine learning models such as regression, decision trees, support vector machines, etc. In contrast, CNNs are not traditional methods and have recently become widely used.
- CNNs have been shown in numerous studies to yield superior performance compared to other models. Labeling CNNs as conventional models |

| | may diminish their value and advanced nature, potentially leading to misunderstandings about their applicability.

Therefore, we will use the term "standard CNN models" to refer to models other than retrained CNN models. We have added a section to explain this terminology to prevent any confusion for the readers.

This study will use various CNN models to predict deep-seated slope displacement. The CNN models
employed in this research include VGG (Simonyan and Zisserman, 2014), ResNet (He et al., 2016),
Inception (Szegedy et al., 2015), Xception (Chollet, 2016), MobileNet (Howard et al., 2017), DenseNet
(Huang et al., 2017), and NASNet (Zoph et al., 2018). To clarify, the term "standard CNN models" will
refer to models with structures that can be user-defined, while "retrained CNN models" will denote those
with architectures that have been researched and developed by other scientists and have been proven to
be highly effective. |
|---|---|
| 678: Here again would be a great place to delve into the "why" a little bit more. Any thoughts why a certain class of models outperforms the others? This discussion section is quite short relative to the rest of this paper, and there are a lot of aspects to potentially discuss. Does withholding certain parameters (e.g., temperature, humidity, or both) impact the results substantially? If so, why might that be the case? Since so much work has been done to get to this stage of predictive success, a small amount of additional work may help elucidate the role of specific processes in aiding the predictability of landslide displacement in this context that could be useful for the broad readership of NHESS. | We have expanded the discussion section to provide a more comprehensive explanation of the study's results. Specifically, we have added reasons to explain why CNN models performed better than both machine learning and time-series deep learning models. Additionally, this discussion highlights a limitation of the study: the lack of analysis on the relative importance of each type of input data for the predictive capabilities of the AI models. This limitation underscores the need for further research to clarify these aspects.

of our study lies in adopting pre-trained models, such as MobileNet, DenseNet, Inception, and VGG,
rather than  standard CNN models.
By employing various AI models, this study identifies the most effective model for predicting deep-
seated landslides and offers a comprehensive overview of the performance of different AI models. Initially,
machine learning models exhibited relatively high prediction errors, with MAPE ranging from 8.14% to
15.19%. This performance was generally lower than time-series deep learning models, which showed
MAPEs ranging from 7.9% to 14.73%. The superior performance of the time series deep learning models
is attributed to their ability to process sequential data and retain information from previous steps. This
enables them to learn patterns from the dataset more effectively than traditional machine learning models.
However, compared to CNN models, the results of the time series deep learning models are not as
strong. This disparity is attributed to the superior learning mechanism of CNNs. The convolutional and
pooling layers in CNNs enable robust feature extraction from the input data. Convolutional layers are
particularly effective at identifying complex patterns and subtle features within time series data, primarily
when spatial correlations exist. This capability allows CNNs to uncover essential features that other
models might overlook.
The input data used for the AI models were selected because they significantly influence the
likelihood of deep-seated landslides, as detailed in Section 3.6. However, a limitation of this study is that
it does not evaluate the relative importance of each input data type on prediction accuracy. Future research
should explore the impact of different combinations of input data on AI model performance. This could
help identify the significance of each input type and potentially reveal the optimal combination of inputs
to enhance prediction accuracy further. |

---

## Author Comment (AC2)

**Summary of the Changes to Reviewer 2's Recommendations and Comments**

**Journal: Natural Hazards and Earth System Sciences**

**Ref: NHESS-2024-86**

Title: Predicting Deep-Seated Landslide Displacements in Lushan Mountains through the Integration of Convolutional Neural Networks and an Age of Exploration-Inspired Optimizer

The authors appreciate the reviewer's valuable feedback. The summary of the changes based on the reviewer's recommendations & comments is listed below. All the revisions are TRACKED in the re-submitted WORD file along with marked RED COLOR for the ease of the reviewer's perusal. Our colleague, a native English speaker of BLUE COLOR, has corrected grammatical and writing style errors in the original version.

| Recommendations and Comments of Reviewer | Authors' Summary of the Changes |
|---|---|
| The manuscript can be an interesting contribution for the methodology of use and interpretation of data for the prediction of deep landslide movements. However, it requires a substantial review in the text, in the figures and in the production of additional figures to show the final results. The list presented below are the specific comments: | We are pleased to receive positive feedback from the reviewer on this study. We also sincerely appreciate the reviewer's detailed comments, which have identified the limitations of our research. We have endeavored to revise the manuscript in response to each of the reviewer's comments. The details of these revisions are outlined below. |
| 1) Sections 3.1 and 3.2 should be in the text in more synthetic form, placing much of the content in an appendix | We completely agree with the reviewer's suggestion. Excessive focus on the operational mechanisms of the AI models could distract readers from the primary objective of the study. Therefore, we have moved this content to the appendix. |

**3.1 Convolutional Neural Networks**
In 1998, LeCun introduced a novel type of DNN known as the CNN, specifically designed for
processing data with a grid-like structure, such as images. The complex, layered system of CNN facilitates
the automated extraction of features without extensive preprocessing, making it ideal for object
recognition, image classification, and segmentation tasks. The detailed mechanism of the CNN model can
be found in appendix A. The architecture of a typical CNN, as illustrated in Figure 2, comprises an input
layer (to receive image data), followed by hidden layers (including convolutional, pooling, and fully
connected layers), and concludes with the output layers. As depicted in Figure 2, the complexity of CNN
**3.2 Deep Learning Models for Time Series**
RNN was introduced by Elman in 1990 (Elman, 1990). This model makes predictions based on
sequential data, crucial for language modeling, document classification, and time series analysis. The
architecture of an RNN model can be found in appendix B.
**APPENDIX**
**Appendix A. Convolutional Neural Networks**
The architecture of a typical CNN, as illustrated in Figure A-1, comprises an input layer (to receive
image data), followed by hidden layers (including convolutional, pooling, and fully connected layers), and
concludes with the output layers. As depicted in Figure A-1, the complexity of CNN progressively
increases from the convolutional layer to the fully connected (FC) layer. This design enables CNN to
recognize relatively simple patterns (lines, curves, etc.) before progressing to capture more intricate
features (faces, objects, etc.), with the ultimate aim of extracting relevant information for accurate pattern
identification.

Figure A-1. Structure of basic CNN.

| | |
|---|---|
| | 12      As illustrated in Figure A-2, the convolutional layer is responsible for most computations in the
network. This involves extracting local features from an image using a set of learnable filters known as
kernels. The behavior of the filter in the convolutional layer is influenced by two main factors: stride and
padding. Stride refers to the pixel shift of the filter across the image, while padding aims to preserve
information at the corners. In each iteration, a portion of the image is convolved with a filter to generate
a dot product of pixels within its receptive field. This process is replicated across the entire image to
produce a feature map. The convolution operation is defined as follows: |
| 2) In section 3.4.2 the equation of the MAPE, MAE and RSME objective function is not presented | We have revised Section 3.4.2, adding detailed explanations of the calculations and the significance of each evaluation metric. These explanations enable readers to better understand the objective function when these evaluation metrics are applied.

◢ **3.4.2 Performance Metrics**
This study utilized four widely recognized performance measures to assess the  model's
effectiveness in prediction accuracy (Chou and Nguyen, 2023). The measures included mean absolute
error (MAE), mean absolute percentage error (MAPE), and root mean square error (RMSE).
MAE represents the mean of absolute errors, calculated as the average of the absolute
differences between actual and predicted values. Its advantage lies in its simplicity, which
provides a straightforward measure of average prediction error. However, a drawback of MAE is
its insensitivity to more significant errors, so it may not effectively highlight differences between
models when significant errors are present. It is defined as:
$MAE = \frac{1}{n}\sum_{i=1}^{n}|y_i - \hat{y}_i|$                      (5)
where $n$ is the number of predictions, $y_i$ is the $i^{th}$ forecasted value, and $\hat{y}_i$ is the corresponding $i^{th}$
actual value.
MAPE quantifies the average absolute error ratio—derived from the differences between
actual and forecasted values—to the actual value. It provides a clear metric in percentage terms,
facilitating straightforward interpretation across various datasets. However, MAPE's limitation
arises from its sensitivity to zero values in the actual data, which can become undefined or
impractical to compute, limiting its utility in scenarios involving zero or near-zero actual values.
The expression for MAPE is as follows:
$MAPE = \frac{1}{n}\sum_{i=1}^{n}\left|\frac{y_i - \hat{y}_i}{y_i}\right|$                (6)
where $n$ is the number of predictions, $y_i$ is the $i^{th}$ forecasted value, and $\hat{y}_i$ is the corresponding $i^{th}$
actual value.
RMSE represents the square root of the average squared error between actual and forecasted
values and is widely used for its ability to indicate the dispersion of errors. This method captures
the magnitude and direction of errors, making it practical for assessing overall prediction
accuracy. However, RMSE tends to be more sensitive to outliers and significant errors than MAE
due to its squaring of errors during computation. This sensitivity can disproportionately affect its
evaluation in datasets with extreme values. The expression for RMSE is as follows:
$RMSE = \sqrt{\frac{1}{n}\sum_{i=1}^{n}(y_i - \hat{y}_i)^2}$           (7)
where $n$ is the number of predictions, $y_i$ is the $i^{th}$ forecasted value, and $\hat{y}_i$ is the corresponding $i^{th}$
actual value. |
| 3) Section 3.5 - Chou and Nguyen in 2024 article not present in the bibliography or not mentioned in the correct form | The AEIO algorithm employed in this study was developed in 2024. It has successfully undergone testing on small, average, and large-scale benchmark functions, as well as in optimizing the hyperparameters of AI models. However, since the algorithm is currently under review for publication in a separate journal, we are unable to include it as a reference in this manuscript. We kindly ask for the reviewers' understanding regarding this limitation. Although we have not added a citation for the AEIO algorithm, we have provided a highly detailed explanation of its usage to ensure that readers can easily understand and apply it, as outlined below. |

[revised manuscript text omitted]

Additionally, the AEIO algorithm demonstrated strong optimization capabilities for the hyperparameters of AI models in this study, highlighting its effectiveness.

| 4) Section 3.5 - EQ. 10 and 11 - The meaning of the Maxit and Mind parameters are not indicated | We acknowledge the error in our initial manuscript, as pointed out by the reviewer's suggestion. We have now added annotations for the parameters d, D, $n_{pop}$, t, and MaxIt in Equation (10). Additionally, we have clarified that these values hold the same meaning in Equations (11) and (12). |
| --- | --- |

- **Explorers follow three other peers**

Explorers employing this movement method will calculate the average position of three randomly selected other explorers $\left(\frac{x_{1,d}(t)+x_{2,d}(t)+x_{3,d}(t)}{3}\right)$ and then move toward this newly calculated average position. The explorer's new position is computed using the following formula:

$$x_{i,d}(t+1) = x_{i,d}(t) + \left(\frac{x_{1,d}(t)+x_{2,d}(t)+x_{3,d}(t)}{3} - x_{i,d}(t)\right) \times rand(0,1) \times R \tag{10}$$

where: $x_{1,d}(t)$, $x_{2,d}(t)$ and $x_{3,d}(t)$ are three random explorers in dimension $d$ at iteration $t$, $d = 1,2,\ldots D$; $D$ is the number of dimensions; $i = 1,2,\ldots n_{Pop}$; $n_{Pop}$ is the total number of explorers; $t = 1,2,\ldots MaxIt$ is the number of iterations; $MaxIt$ is the maximum value of iteration.

- **Explorers follow the best one**

According to this strategy, the explorer $(x_{i,d}(t))$ will move closer to the position of another explorer currently holding the best position $(Best_d(t))$, as determined by the following formula:

$$x_{i,d}(t+1) = x_{i,d}(t) + (Best_d(t) - x_{i,d}(t)) \times rand(0,1) \times R \tag{11}$$

where: $Best_d(t)$ represents the position of the particle with the best fitness in dimension d at iteration t, the parameters $d$ and $t$ hold the same significance as defined in Equation 10.

- **Explorers follow guidance from another one**

Explorers in favorable positions with access to information can execute this movement strategy. In this scenario, explorers $(x_{i,d}(t))$ will consult with  another explorer. The consulted explorer will

| | 462   compare their direction and distance to the best individual, who holds the most favorable position |
| | 463   $(Best_d(t))$ and guide the inquirer. This algorithm assumes that the inquirer can be any explorer, i.e., a |
| | 464   random explorer $(x_{1,d}(t))$. The following formula describes how to calculate the new position of the |
| | 465   explorer following this strategy: |
| | 466   $x_{i,d}(t + 1) = x_{i,d}(t) + \left(Best_d(t) - x_{1,d}(t)\right) \times rand(0,1) \times R$        (12) |
| | 467   where: $x_{1,d}(t)$ is a random explorer in dimension $d$ at iteration $t$. the parameters $d$ and $t$ hold the same |
| | 468   significance as defined in Equation 10. |

| | |
|---|---|
| 5) Section 3.6.0-In Figure 9, references are indicated to the 18-19-20-21 and 22 equations. But these equations do not exist and the text | We have revised the equation numbering in this flowchart to ensure consistency with the sequence of equations presented earlier.
[Figure]
 Figure 9. Flowchart of the fine-tuning process of AI models by the AEIO algorithm |
| 6) section 3.6.0 in Figure 9 and in the text the optimization stop criterion should be indicated. | We fully agree with the reviewer's suggestion and have added content to the manuscript to emphasize the stop criterion of the AEIO algorithm. |

| | 426      In each iteration, explorers forecast their next move. If it promises a better position, they relocate. |
| | 427   Otherwise, if the new spot is less favorable for colony establishment, they stay put and await the next |
| | 428   iteration. The algorithm employs specific mathematical formulas to calculate the movement step of |
| | 429   explorers or particles in the AEIO. The exploratory steps of explorer in the AEIO algorithm will |
| | 430   continuously iterate until the stop condition is satisfied. |
| | 475      The exploratory steps in the AEIO algorithm begin by classifying positions using the DBSCAN |
| | 476   algorithm. Subsequently, the explorers update the crowd control mechanism according to equations (13) |
| | 477   and (14), and move according to various strategies defined by equations (8), (10), (11), and (12). This |
| | 478   process is conducted iteratively, continuing until the maximum number of iterations is reached. |

We have also incorporated the stop criterion into the flowchart of the AEIO algorithm during the fine-tuning of the AI model's hyperparameters.

[Figure]

Figure 9. Flowchart of the fine-tuning process of AI models by the AEIO algorithm

| | |
|---|---|
| 7) Section 3.6.2. Figures 12 13 and 14 should be presented together in the same group with the same temporal axis. And an additional figure should be added to the group, with the temporal sequence of the rains | We have revised these figures by merging Figures 12, 13, and 14 into a single figure, presented along a unified timeline. Additionally, the new figure includes rainfall data from significant storms in the region to facilitate easier comparison for the readers.
[Figure]
 Figure 12. Unified timeline visualization of data in this study. A) Precipitation of recorded heavy rainfall in the studied area; B) Measured displacements from extensometer SAA C) Measured displacements from extensometer E_2; D) Groundwater levels at stations A-17, A-18-2, A-20, and A-24. |
| 8) in Section 4, the comparative result of the deformations observations (shown in figure 14) with | In response to the reviewer's suggestion, we have added a figure that displays the predicted deep- |

| | |
|---|---|
| the comparative predictions of the best model should be graphically presented. | seated displacement of the best machine learning model, the best time-series deep learning model, the best CNN model, and the best hybrid models. This allows readers to compare and assess the predictive capabilities of these models.

Figure 10 illustrates the differences between typical AI models' actual and predicted deep-seated
displacement. Specifically, Figure 10a compares the performance of single models against the predicted
values, while Figure 10b does the same for hybrid models. The chart shows that, hybrid models
demonstrate superior predictive capability for deep-seated landslides compared to single models. This is
evident from the displacement line of the hybrid models in Figure 10b which closely aligns with the actual
deep-seated displacement and significantly outperforms the single models depicted in Figure 10a.

[Figure]

a) Prediction results of deep-seated displacement by single AI models.
b) Prediction results of deep-seated displacement by AI models optimized using the AEIO algorithm.
Figure 10. Graph comparing the real and predicted deep-seated displacement. |
| 9) section 4.2 is too long and should be simplified and synthesized | We fully understand the reviewer's concern regarding the length of Section 4.2. However, it is important to note that much of the length is due to the inclusion of performance result tables for the models, which are essential and cannot be condensed.

 Additionally, we believe that the explanations and commentary on the models' performance are equally essential. These details not only enhance the manuscript's relevance to readers interested in landslide research but also appeal to those focused on the use of AI models for regression studies.

 Last but not least, while this section is lengthy, it is organized in a logical structure. As a result, readers will not be distracted by its length; instead, they can easily find information on the specific models they are interested in, corresponding to each subsection within Section 4.2.

 However, in response to the reviewer's valuable suggestion, we have revisited Section 4.2 and removed redundant content, retaining only the information that is most valuable to the readers. |

most suitable machine learning model for predicting deep-seated landslides, exhibiting both high prediction accuracy and a short running time.

Similar to the machine learning models, in this section, the time series deep learning models will also be trained with default hyperparameters, as found in research of

Chou and Nguyen (2023). The performance results of these models are shown in Table 7. Overall, akin to demonstrates higher prediction accuracy. Therefore, the R-GRU model will be chosen as the best numerical AI model.

displacement of R-GRU before fine-tuning was 7.9%, but this number decreased to only 3.03% after fine- tuning.

Additionally, Table 9 indicates that predictions from the dataset of the E-2 station consistently outperform those of the SAA station. Specifically, the displacement prediction at the E-2 station is 3.03%

and 6.38%, better than the corresponding numbers for the SAA station, which are 3.94% and 7.96%, respectively. This is attributed to the dataset collected by the E-2 station being more comprehensive and gathered over a more extended period than the SAA station (as shown in Table 4).

the AEIO algorithm, are presented in Table 14.  CNN models with optimal hyperparameters  exhibit the most minor errors, indicating that these are the most effective models in this study for predicting deep-seated displacement

Figure 10 illustrates the differences between typical AI models' actual and predicted deep-seated displacement. Specifically, Figure 10a compares the performance of single models against the predicted values, while Figure 10b does the same for hybrid models. The chart shows that, hybrid models demonstrate superior predictive capability for deep-seated landslides compared to single models. This is evident from the displacement line of the hybrid models in Figure 10b which closely aligns with the actual deep-seated displacement and significantly outperforms the single models depicted in Figure 10a.

---

## Referee Report (RR1)

**Review for Chou et al., "Predicting Deep-Seated Landslide Displacements in Lushan Mountain through the Integration of Convolutional Neural Networks and an Age of Exploration-Inspired Optimizer"**

I reviewed a previous version of this manuscript and suggested major revisions. The authors have taken care to address my suggestions point-by-point, and their revised manuscript reflects well the efforts of the authors to incorporate these suggestions. The results shown herein are impactful, and I appreciate the thorough investigation of models that can help guide future researchers who may undertake similar efforts. I therefore recommend publication of this work in *NHESS*; however, I include some additional line-by-line comments for the authors to address below:

1 (Title): It is a good idea to insert the country name here so people know where the Lushan mountains are located

38-44: Much improved with the added context here!

45: Should have references to support this

49: There are much older references than these, e.g., Iverson and Major (1985) and references therein

63: It is not mentioned what the constraints are of traditional machine-learning models

73: A term to use throughout the manuscript would be "deep-seated landslide displacement"

74: Would insert the country name here as well

131: Specify which atmospheric variables will be used instead of the term "weather conditions"

134 (Fig. 1): Why is the orange layer filled in on the second panel and not the first? Why not the upper layer too? Additionally, water tables typically include an inverted triangle denoting their position.

149: change to "physically based" from "physical-based"

164-171: There is a deep literature on this subject and I encourage the authors to include some more fundamental contributions to slope stability analysis here. It does not need to be a substantially longer paragraph as that is not the focus of this work. However, some more foundational work should be briefly referenced.

172-180: I'm not sure I understand this paragraph. Why are AI models better suited to incorporation of new data than, say, deterministic models? I think the advantage may be that

most deterministic modeling requires some knowledge of physics to predict displacement, which can be exceedingly complex in a large landslide, and these kinds of models rarely can achieve predictive success of a few percent.

184-186: There is somewhat of a disconnect here because the Margarint et al. paper does not appear to utilize AI, it just presents an analysis using a standard logistic regression model. The preceding sentence should therefore be changed, or a more appropriate example should be provided.

474-477: The DBSCAN algorithm is not mentioned previously to this point and thus it is confusing. Furthermore, Equations 13 and 14 do not exist in the manuscript. Some additional prior explanation is needed here.

490 (Fig. 6): It would be useful to have the approximate failure plane depths measured for G20 and G21 shown graphically here.

494: I think the term "youthful" is too colloquial here

514-521: I don't think my previous comment regarding the definition of "cleavages" was sufficiently addressed here. Please specify what this term means in this context, or utilize a different term throughout

546 (Fig. 7): This is much improved from the previous figure, although there is an issue now in that the timing does not appear to line up between the plots. For example, the large displacement in 2012 appears to come *before* the rise in water levels in (D).

554-556: Did a previous study show specifically that a structural alteration in soil took place? Also, the failure plane is well below the "soil" depth and the landslide displacement should be insensitive to the soil present at the landslide surface. I recommend re-writing to say that, based on the temporal association of rapid displacement with a rapid rise in groundwater levels, it could be inferred that enhanced pore water pressure lead to the onset of motion.

616: "Deep-seated landslide displacement"

776 (Fig. 10). Why are the descriptions at (a) and (b) above the introduction to Fig. 10? Second, in panel (a) there are a bunch of confusing floating dots that fall below the main plot and cover the legend. Third, the dots in general are distracting because it is difficult to see the subtle differences in each time series. I would remove the dots and just show lines for each model.

783: This is not entirely fair as there are a number of studies now that use AI to forecast landslide displacement as a function of environmental variables.

826: I would specify that this study addresses the persistent threat of large, slow-moving landslides.

**References**

IVERSON, R. M. and MAJOR, J. J.: Rainfall, ground-water flow, and seasonal movement at Minor Creek landslide, northwestern California: Physical interpretation of empirical relations, GSA Bulletin, 99, 579–594, https://doi.org/10.1130/0016-7606(1987)99<579:RGFASM>2.0.CO;2, 1987.

---

## Author Response (AR2)

**Summary of the Changes to Reviewer 1's Recommendations and Comments**

**Journal: Natural Hazards and Earth System Sciences**

**Ref: NHESS-2024-86R2**

Title: Predicting Deep-Seated Landslide Displacement in Taiwan's Lushan Mountain through the Integration of Convolutional Neural Networks and an Age of Exploration-Inspired Optimizer

The authors appreciate the reviewer's valuable feedback. The summary of the changes based on the reviewer's recommendations & comments is listed below. All the revisions are TRACKED in the re-submitted WORD file along with marked RED COLOR for the ease of the reviewer's perusal.

| Comments of the Reviewer | Authors' Summary of the Changes |
|---|---|
| I reviewed a previous version of this manuscript and suggested major revisions. The authors have taken care to address my suggestions point-by-point, and their revised manuscript reflects well the efforts of the authors to incorporate these suggestions. The results shown herein are impactful, and I appreciate the thorough investigation of models that can help guide future researchers who may undertake similar efforts. I therefore recommend publication of this work in NHESS; however, I include some additional line-by-line comments for the authors to address below: | We, the authors of this study, would like to express our sincere gratitude to the reviewer. The feedback provided in the previous round has guided our improvements. We strive to meet the expectations of both the reviewer and the NHESS journal. |
| 1 (Title): It is a good idea to insert the country name here so people know where the Lushan mountains are located | We fully agree with the reviewer's suggestion and have added the country name to the title of this manuscript to provide readers with more detailed information about the study location.

**Predicting Deep-Seated Landslide Displacement in Taiwan's Lushan**
**Mountain through the Integration of Convolutional Neural Networks and an Age of**
**Exploration-Inspired Optimizer** |
| 38-44: Much improved with the added context here! | We are glad our revisions have met the reviewer's expectations in this section. |
| 45: Should have references to support this | We have included additional references to support the assertion that 'critical factors associated with slope instability exhibit temporal variability' as requested by the reviewer.

a few meters, deep-seated landslides extend deeper, often exceeding 10 meters, and can involve the
movement of underlying bedrock (Lin et al., 2013). Predicting these events is challenging and costly (Thai
Pham et al., 2019). Therefore, extensive efforts have been made to predict such disasters throughout
history (Corominas and Moya, 2008; David and Raymond, 1989; Aleotti and Chowdhury, 1999). One
method that has been employed involves thoroughly examining the physical and geological characteristics
of the mountainous areas at risk of landslides (Cotecchia et al., 2020). Furthermore, the level of |
| 49: There are much older references than these, e.g., Iverson and Major (1985) and references therein | We acknowledge the value of the reference recommended by the reviewer, which provides an excellent explanation for the argument in question. |

| | |
|---|---|
| | Notably, this reference was conducted some time ago, indicating that the argument has been widely accepted within the academic community for quite some time. Consequently, we have incorporated this reference as a citation in the relevant section.

method that has been employed involves thoroughly examining the physical and geological characteristics
of the mountainous areas at risk of landslides (Cotecchia et al., 2020). Furthermore, the level of
groundwater has been shown by numerous studies in the past to influence the mechanisms behind
landslide formation significantly (Miao and Wang, 2023; Preisig, 2020; Iverson and Major, 1987). |
| **63: It is not mentioned what the constraints are of traditional machine-learning models** | We have added a discussion on the limitations of machine learning in this section, as suggested by the reviewer.

One of the most effective solutions for constructing models to predict time series data involves
applying data-driven techniques. The advancement of computational capabilities has driven the
widespread adoption of data-driven machine-learning models over physics-based models. This shift is
based on the premise that the data used for slope monitoring originates from nonlinear systems (Zhou et
al., 2018). However, a significant drawback of traditional machine learning models
, such as Random Forest and Support Vector Machines, is their
difficulty  handling spatiotemporal data. These models struggle to capture the sequential relationships
necessary for landslide prediction, resulting in lower performance (Zhang et al., 2022a; Tehrani et al.,
2022). |
| **73: A term to use throughout the manuscript would be "deep-seated landslide displacement"** | We fully concur with the reviewer's insight and have consistently utilized the term 'deep-seated landslide displacement' throughout this study.

[revised manuscript text omitted]

| | |
|---|---|
| 74: Would insert the country name here as well | We have included information regarding the country Taiwan to enhance the reader's understanding of the study's geographical context.

by an innovative metaheuristic optimization algorithm to predict deep-seated
landslide displacement on the northern slope of Lushan Mountain in Ren'ai Township, Nantou County, Taiwan. The geological characteristics of this area have undergone extensive research (Wang et al., 2015;
Lin et al., 2020). Previous studies have identified varying depths of the shear plane. Specifically, Lin et |
| 131: Specify which atmospheric variables will be used instead of the term "weather conditions" | Specifying the variables related to weather conditions undoubtedly enriches the information presented and enhances the clarity and comprehensibility of this study for readers. Accordingly, we have incorporated this information following the reviewer's suggestion.

Our research aims to adopt a novel approach compared to previous landslide studies at Lushan
Mountain by utilizing AI models and metaheuristic optimization algorithms. This research will utilize
temperature, humidity, and groundwater levels as input datas for AI models to predict
deep-seated landslide displacement, thus aiding in landslide forecasting in this
region. |
| 134 (Fig. 1): Why is the orange layer filled in on the second panel and not the first? Why not the upper layer too? Additionally, water tables typically include an inverted triangle denoting their position. | We have revised Figure 1 to ensure color consistency between the images on the left and right. In the right image, only the water layer is filled with color, while the soil and rock layers remain uncolored. Additionally, we have added an inverted triangle symbol to mark the location of the groundwater. |

| | |
|---|---|
| |
[Figure]

Figure 1. Schematic illustration showing the effects of groundwater on deep-seated slope failure. |
| 149: change to "physically based" from "physical-based" | We have made the changes as per the reviewer's suggestion.
Moreover, physically based numerical and laboratory modeling methods are also gaining traction
in landslide research. These methods aim to maintain forecasts using various data types while reducing |
| 164-171: There is a deep literature on this subject and I encourage the authors to include some more fundamental contributions to slope stability analysis here. It does not need to be a substantially longer paragraph as that is not the focus of this work. However, some more foundational work should be briefly referenced. | In response to the reviewer's request, we have included additional citations of studies that employ stability analysis in landslide assessments.
Stability analysis is another commonly used method related to physics, which evaluates the forces
acting on a slope behavior. Fu and Liao (2010) presented a technique for implementing the non-linear
Hoek-Brown shear strength reduction, determining the correlation between normal and shear stress based
on the Hoek-Brown criterion. Subsequently, the micro-units (microscopic components of the rock mass)
instantaneous friction angle and cohesive strength under specific stress conditions are calculated.
Although this approach effectively addresses cost and labor issues, it still heavily relies on the researcher's
assumptions and is limited by the ability to utilize only a small portion of data from the research area.
Additionally, there are several other limitations. For instance, Mebrahtu et al. (2022) indicated that
stability analyses become less reliable in seismic load scenarios. Safaei
et al. (2011) also noted that stability analysis necessitates a substantial amount of detailed input data
obtained from laboratory tests and field measurements, thereby limiting the areas that can be effectively
assessed. |
| 172-180: I'm not sure I understand this paragraph. Why are AI models better suited to incorporation of new data than, say, deterministic models? I think the advantage may be that most deterministic modeling requires some knowledge of physics to predict displacement, which can be exceedingly complex in a large landslide, and these kinds of models rarely can achieve predictive success of a few percent. | We greatly appreciate the reviewer's suggestion, which allowed us to revise this section for greater clarity. We have updated the passage to explain why conventional methods were not used, as they require users to have specialized knowledge in physics and demand specific types of input data, making them less flexible compared to AI models. Therefore, given the advantages of AI models, they will be utilized in this study.
As previously mentioned,
using conventional methods poses significant challenges, as their application requires a deep
understanding of both the physics involved and the complex behavior of soil
In addition, traditional methods
require specific types of input data, highlighting the rigidity and lack of flexibility inherent in these
approaches (Safaei et al., 2011). In contrast, AI models can overcome these difficulties by automatically
learning to identify  mapping functions between input and output data, eliminating
users needing specialized knowledge of soil behavior and physics. Additionally, AI
models can be updated to incorporate new input variables, offering flexibility to leverage available data
based on real-world conditions.
Therefore, AI models will
be utilized in this research instead of conventional methods. |
| 184-186: There is somewhat of a disconnect here because the Margarint et al. paper does not appear to utilize AI, it just presents an analysis using a standard logistic regression model. The preceding sentence should therefore be changed, or a more appropriate example should be provided. | Based on the reviewer's comment, we reference a different citation that employs AI models in landslide research to align better with the core content of this section. |

<table>
<tr>
<td></td>
<td>

In studies employing machine learning and deep learning models for landslide research, a plethora
of research utilizes discrete data to train AI models to predict the probability of landslides or to construct
maps depicting landslide susceptibility. For instance,
Pradhan and Lee (2010) used Geographic Information
System (GIS), remote sensing, and a neural network model to analyze landslide susceptibility in Cameron
Highlands, Malaysia. Ten factors, including topographic slope and drainage distance, were processed to
generate a susceptibility map. The model achieved 83% accuracy in predicting landslide locations.
In a similar study, Pham et al. (2016) used multiple AI models, including support vector machines (SVM),
logistic regression (LR), Fisher's linear discriminant analysis (FLDA), Bayesian network (BN), and naïve
Bayes (NB), for landslide susceptibility assessment in a region within the Uttarakhand state of India. The
SVM model yielded the best prediction results among the models used.

</td>
</tr>
<tr>
<td>

474-477: The DBSCAN algorithm is not mentioned previously to this point and thus it is confusing. Furthermore, Equations 13 and 14 do not exist in the manuscript. Some additional prior explanation is needed here.

</td>
<td>

To explain the AEIO algorithm in this study, we have added citations and a description of the DBSCAN algorithm. We hope this addition will enhance the reader's understanding of the algorithm.

The strength of the AEIO algorithm lies in its ability to develop specific strategies for particles based
on their positions, enabling faster convergence to the optimal point.  density-based spatial
clustering of applications with noise (DBSCAN) for particle clustering. DBSCAN is an unsupervised
clustering method that organizes data points by their spatial closeness in high-dimensional spaces (Ester
et al., 1996). This algorithm is particularly effective at detecting clusters of different shapes and densities.
It relies on two primary parameters: ε (the radius of the neighborhood) and MinPts (the minimum number
of points required to form a dense area). Clusters are created by locating neighboring points
with enough surrounding points, while those that do not
belong to any cluster are classified as noise or outliers.
Using the DBSCAN algorithm,  the AEIO determines whether particles are in favorable or
unfavorable positions, reminiscent of explorers during the Age of Exploration. The proximity (within
clusters) allows explorers to gather information and move toward optimal locations, thereby enhancing

Additionally, we have rechecked the numbering of the equations and made necessary corrections to ensure their accuracy.

The exploratory steps in the AEIO algorithm begin by classifying positions using the DBSCAN
algorithm. Subsequently, the explorers update the crowd control mechanism according to equations (9)
and (10), and move according to various strategies defined by equations (4), (6), (7), and (8).
This process is conducted iteratively until the maximum number of iterations is reached.

We have also updated the numbering of the equations in the flowchart to facilitate easier tracking for the readers.

</td>
</tr>
</table>

[Figure]

     Figure 4. Flowchart of the fine-tuning process of AI models by the AEIO algorithm.

      Figure 4. Flowchart of the fine-tuning process of AI models by the AEIO algorithm.

| | |
|---|---|
| 490 (Fig. 6): It would be useful to have the approximate failure plane depths measured for G20 and G21 shown graphically here. | Incorporating the location of the failure plane is undoubtedly essential, as it provides readers with a clearer understanding of the geological conditions in the area. Therefore, we have included this information in Figure 6.
[Figure]
 Figure 6. Illustration of geological drilling survey. |
| 494: I think the term "youthful" is too colloquial here | We agree with the reviewer's suggestion and will replace 'youthful' with 'incipient.' This term is more academically appropriate and accurately reflects the geological conditions of the area, where many valleys are in the early stages of formation.

The current study focuses on the northern slope of the Lushan hot spring in Ren'ai Township, Nantou
County, Taiwan (Figure 5), with Nenggao Mountain to the east, Hehuan Peaks to the north, Zhuoshe
Mountain to the south, and Puli Basins to the west. The terrain features rugged mountain ranges,
incipient youthful valleys, and notable river erosion (Lee and Chi, 2011). Lushan Hot Springs is located
below the hill, and the main access roads for nearby settlements and hot spring sites include Provincial
Highway 14 and County Highway 87. |

| | |
|---|---|
| 514-521: I don't think my previous comment regarding the definition of "cleavages" was sufficiently addressed here. Please specify what this term means in this context, or utilize a different term throughout | This revision has replaced the term 'cleavage' with 'fracture.'

Previous research has detected signs of brittle deformation in the area. These indications include
chevron folds within fracturecleavages, visible cracks, and intricate jigsaw puzzle-like patterns at the head
of the rock formations. Overturned and flexural toppling fracturecleavages are prevalent towards the toe
of the slope. Additionally, kink bands are observable on fractures recently undergoing flexural folding
along the eastern boundary. Notably, horizontal fracturecleavages near the toe region also
exhibitexhibitsexhibit inter-fracturecleavage gouges. Further details on this geological information can be
found in the study by Lin et al. (2020). These instances highlight the potential for significant geological
changes and landslide risk in this region. |
| 546 (Fig. 7): This is much improved from the previous figure, although there is an issue now in that the timing does not appear to line up between the plots. For example, the large displacement in 2012 appears to come before the rise in water levels in (D). | The synchronization of events across all four charts is vital, highlighting the interrelationship within the dataset used in this study. This alignment forms a solid basis for selecting input variables for the AI models. We have carefully fine-tuned the data to ensure that the events in all four charts are precisely aligned.

[Figure]

Figure 7. Unified timeline visualization of data in this study;
A) Precipitation of recorded heavy rainfall in the studied area; B) Measured displacements from extensometer SAA C)
Measured displacements from extensometer E_2; D) Groundwater levels at stations A-17, A-18-2, A-20, and A-24. |
| 554-556: Did a previous study show specifically that a structural alteration in soil took place? Also, the failure plane is well below the "soil" depth and the landslide displacement should be insensitive to the soil present at the landslide surface. I recommend re-writing to say that, based on the temporal association of rapid displacement with a rapid rise in groundwater levels, it could be inferred that enhanced pore water pressure lead to the onset of motion. | In previous studies on the landslide in Lushan Mountain, Taiwan, other authors did not specifically demonstrate that a structural alteration in the soil occurred. Therefore, based on the reviewer's analysis in this comment, we have revised our explanation to state that enhanced pore water pressure led to motion onset.

The graphs above show that the displacement values at both stations often exhibit significant
increases coinciding with periods of pronounced fluctuations in groundwater levels. Specifically, in June
2012, there was a notable surge in groundwater levels attributed to heavy rainfall from June 8, 2012, to |

June 17, 2012, totaling 1029 mm over 219 hours (as indicated in Table 2 and Figure 7A). The abnormal
rise in groundwater levels led to caused a structural alteration in the area's soil increased pore water
pressure, consequently amplifying which triggered deep-seated displacement deep-seated landslide
displacement at both stations, namely E_2 and SAA, as evidenced in Figure 7B and Figure 7C.

| | |
|---|---|
| 616: "Deep-seated landslide displacement" | We have revised the terminology in this section and throughout the manuscript to 'deep-seated landslide displacement' per the reviewer's suggestion.

4.5.Model Establishment and Analysis Results
4.15.1 Model Establishment
Predicting deep-seated displacement deep-seated landslide displacement at Lushan Mountain is
undoubtedly highly challenging, given that such landslides depend on numerous factors. Therefore,
multiple methods will be employed simultaneously to identify the optimal AI model for prediction. These |
| 776 (Fig. 10). Why are the descriptions at (a) and (b) above the introduction to Fig. 10? Second, in panel (a) there are a bunch of confusing floating dots that fall below the main plot and cover the legend. Third, the dots in general are distracting because it is difficult to see the subtle differences in each time series. I would remove the dots and just show lines for each model. | For the issues identified in Figure 10, we have made several revisions per the reviewer's suggestions. These revisions include the following:
 - Move the descriptions of charts A and B below the introduction of Figure 10.
 - The floating dots appearing in the main plot and covering the legend are due to an error during the PDF export process. We will ensure this issue does not occur in the subsequent sections.
 - Remove the dots on each line to avoid confusion and simplify the plots.

A) MobileNet XGBoost R-GRU Actual
a) Prediction results of deep-seated displacement deep-seated landslide displacement by single AI
models.

B) AEIO-MobileNet AEIO-R-GRU Actual
b) Prediction results of deep-seated displacement deep-seated landslide displacement by AI models
optimized using the AEIO algorithm.
Figure 10. Graph comparing the real and predicted deep-seated displacement deep-seated landslide
displacement: A) Prediction results of deep-seated landslide displacement by single AI models;
B) Prediction results of deep-seated landslide displacement by AI models optimized using the AEIO
algorithm. |
| 783: This is not entirely fair as there are a number of studies now that use AI to forecast landslide displacement as a function of environmental variables. | Other studies have indeed employed AI models to forecast landslide displacement, and claiming this approach as entirely novel is inaccurate. Consequently, we have made several revisions in this part. At the beginning of Section 4.3 (Discussion), we concisely summarized the study's |

| | objectives and removed any misleading information to ensure clarity for the readers. |
|---|---|
| | 806    4.45.3 Discussion
This study focusescenters on landslides in Lushan Mountain, Taiwan, with the aim of
developingintending to develop models to predict deep-seated landslide displacement for both 1-day and
7-day forecasts. These predictive models utilize input data such as groundwater levels, temperature, and
humidity in the regionthe region's groundwater levels, temperature, and humidity. .Accuratelyadopting a
fundamentally different approach than previous research. While past studies primarily focused on
constructing AI models for classification, calculating the probability of landslide occurrences, and
generating landslide susceptibility maps (Balogun et al., 2021; Hakim et al., 2022; Jaafari et al., 2022),
our study is oriented towards predicting displacement to provide warnings about potential landslide
hazards.
As utilized in our calculations, computing deep-seated displacementdeep-seated landslide
displacement offers several benefits. Firstly, understanding internal displacementsit provides accurate
timely information for engineers to assess the resilience of structures and infrastructure in at-risk areas,
facilitating the issuance of sensible warnings. Secondly, forecasting deep-seated displacementdeep-seated
landslide displacement offers insights into the severity of the disaster, aiding in effective evacuation and
rescue planning. |
| 826: I would specify that this study addresses the persistent threat of large, slow-moving landslides. | We are very grateful to the reviewer for this suggestion, which helped clarify the type of landslide most relevant to our study. We have made the necessary revisions in line with the reviewer's recommendation.

5.6 Conclusion
This study addresses the persistent threat of large, slow-moving landslides, a primary concern due to
their severe impact on lives and property. Employing various AI models, such as machine learning, time |